# Evaluating Open-QA Evaluation

**Cunxiang Wang**[1, 2]\*, **Sirui Cheng**[3]\*, **Qipeng Guo**[4], **Yuanhao Yue**[5], **Bowen Ding**[2],
**Zhikun Xu**[5], **Yidong Wang**[2], **Xiangkun Hu**[4], **Zheng Zhang**[4], **and Yue Zhang**[2]†
[1]Zhejiang University, China; [2]School of Engineering, Westlake University, China
[3]Northeastern University, China; [4]Amazon AWS AI; [5]Fudan University, China
{wangcunxiang, zhangyue}@westlake.edu.cn

## Abstract

This study focuses on the evaluation of the Open Question Answering (Open-QA) task, which can directly estimate the factuality of large language models (LLMs). Current automatic evaluation methods have shown limitations, indicating that human evaluation still remains the most reliable approach. We introduce a new task, Evaluating QA Evaluation (QA-Eval) and the corresponding dataset EVOUNA, designed to assess the accuracy of AI-generated answers in relation to standard answers within Open-QA. Our evaluation of these methods utilizes human-annotated results to measure their performance. Specifically, the work investigates methods that show high correlation with human evaluations, deeming them more reliable. We also discuss the pitfalls of current methods and methods to improve LLM-based evaluators. We believe this new QA-Eval task and corresponding dataset EVOUNA will facilitate the development of more effective automatic evaluation tools and prove valuable for future research in this area. All resources are available at `https://github.com/wangcunxiang/QA-Eval` and it is under the Apache-2.0 License.

## 1 Introduction

Open Question Answering (Open-QA) [Chen et al., 2017, Yang et al., 2019], refers to the generation of precise responses to broad, open-ended queries. While Open-QA is useful for downstream applications, such as digital assistant and customer support, it also serves as an essential measure of a model's competency in dealing with factuality [Zhu et al., 2021]. With the advent of Large Language Models (LLMs), such as ChatGPT [OpenAI, 2022a] and BARD [Google, 2023], significant strides have been made in tackling NLP tasks [Qin et al., 2023, Kung et al., 2022]. However, evidence has indicated that LLMs can generate hallucinations or other contents that contradict reality [Bang et al., 2023, Ji et al., 2022, Wang et al., 2023a,b]. Consequently, ensuring factuality has become a prime concern, and Open-QA can be a valuable benchmark task for detecting hallucinations.

The Open-QA task is traditionally evaluated via the Exact Match (EM) score [Izacard and Grave, 2021, Lewis et al., 2020, Chen et al., 2017], which uses whether the model output and one of golden answers are an exact character match to determine whether it is correct. Hence, EM has certain limitations as it does not adequately account for the variation in expression of the answers. For instance, 'Lionel Messi' can also be expressed as 'Lionel Andrés Messi', 'Messi', or 'Leo Messi', none of which would produce an exact match. Moreover, the EM score is inapplicable for detailed responses from LLMs such as ChatGPT [OpenAI, 2022a] and GPT-3.5 [OpenAI, 2022b], where formulating a gold standard answer could excessively restrict model performance.

---

\* Equal contribution
† The corresponding author

To understand the influence of EM mismatch, we manually evaluate the output of five Open-QA models including the classic Dense Passage Retriever (DPR) [Karpukhin et al., 2020] + Fusion-in-Decoder (FiD) [Izacard and Grave, 2021], LLMs including GPT-3.5 [OpenAI, 2022b] and ChatGPT [OpenAI, 2022a], as well as the retrieval-assisted LLM BingChat [Microsoft, 2023] on the standard benchmarks Natural Questions (NQ) [Kwiatkowski et al., 2019] and TriviaQA (TQ) [Joshi et al., 2017] [3], following Karpukhin et al. [2020] and Izacard and Grave [2021]. We focus on aligning model responses with gold standard answers and assessing their factual correctness. Our results show that exact match underestimates model performances by a significant margin. In particular, on the NQ dataset, the DPR + FiD model gives a result of 59.2 by exact match, but 68.9 by human evaluation. In addition, the models' relative performance ranks change according to human evaluation. These show the importance of finding a correct evaluation metric for Open-QA.

Our human evaluation records can serve as a benchmark for investigating what evaluation methods are best for evaluating the performance of various models on Open-QA. As a result, we consider the novel task of **Evaluating Open-QA Evaluation (QA-Eval)** by making our human evaluation results as the QA-Eval dataset EVOUNA (**EV**aluating **O**pen q**U**estion a**N**swering ev**A**luation). The main idea is to calculate the correlation between evaluator results on the models on the dataset and the human annotation. If a model has a higher correlation with human annotation, then it can be regarded as being more reliable, and vice versa.

Using EVOUNA, we examine several commonly used automatic evaluation metrics, including Lexical Matching, Neural-Evaluation [Zhang* et al., 2020], and LLMs [OpenAI, 2022b, 2023]. The last two methods have been widely used in evaluating NLG tasks [Sellam et al., 2020, Zhao et al., 2020, Qin et al., 2023] but not yet for Open-QA. Results on EVOUNA show that, firstly, although these methods are somewhat effective, they still fall short compared to human evaluator.In addition, the LLM-evaluators tends to perform worse on long answers with much additional information. Notably, while other evaluators have moderate performance, GPT-4 could be a promising candidate, as it scores only slightly lower than human evaluators in two dataset.

In our study, we assess the strengths and limitations of different automatic evaluators, identifying shared and unique error types for each. We've manually categorized these errors, revealing that evaluators often exhibit over-strictness for varied reasons. Our findings provide key insights for optimizing Open-QA evaluation using these evaluators.

To our knowledge, we are the first in manually assessing the correctness of answers generated by models on Open-QA and create a benchmark to evaluate evaluators on Open-QA. All resources of our benchmark EVOUNA is released at `https://github.com/wangcunxiang/QA-Eval`.

## 2 Related Work

**Large Language Models (LLMs)** , such as the GPT-series [Radford et al., 2018, 2019, Brown et al., 2020], including GPT-3.5 [OpenAI, 2022b], ChatGPT-3.5 [OpenAI, 2022a], and Bing Chat [Microsoft, 2023], have been at the forefront of recent research due to their capability to generate coherent and contextually accurate textual output. GPT-3.5, an evolution of OpenAI's GPT-3 [Brown et al., 2020], houses an immense 175 billion parameters, enhancing its capabilities in understanding and navigating complex linguistic structures. ChatGPT, a conversational variant of GPT-3.5, is specifically tailored for dialog-based applications [OpenAI, 2022a]. Functioning as an interactive AI chatbot, it provides contextually aware and fluent responses to human inputs. Bing Chat, a development by Microsoft, integrates the language comprehension of ChatGPT and an extensive retrieval system to serve as a conversational search engine [Microsoft, 2023].In this study, we utilize GPT-3.5, ChatGPT, and Bing Chat for Open-QA and GPT-3.5 for QA-Eval.

**Open-QA Datasets:** Natural Questions (NQ) [Kwiatkowski et al., 2019] and TriviaQA (TQ) [Joshi et al., 2017] are widely used datasets in the Open-Domain Question Answering (Open-QA) community, offering unique advantages for training and testing models' capabilities. The Natural Questions (NQ) dataset, is designed to reflect real-world information-seeking questions and their answers. TriviaQA consists of questions from trivia and quiz-league websites. We choose to use both NQ and TQ in our work for several reasons. Firstly, these datasets provide a diverse range of questions and answers [Kwiatkowski et al., 2019, Joshi et al., 2017]. This diversity helps us to better

---

[3]We concentrate on objective question answering with relatively short answers in this work

understand the strengths and weaknesses of different models and evaluation methods across various question types. Furthermore, both NQ and TQ are well-known and commonly used in the research community [Petroni et al., 2021, Izacard and Grave, 2021, Ju et al., 2022, Wang et al., 2023c]. Last, they are both under the Apache-2.0 license.

**Evaluation on LLM.** Recently, numerous research have conducted on LLMs. For instance, some researchers conduct a systematic analysis of ChatGPT's zero-shot capabilities on representative NLP tasks, concluding that while ChatGPT excels in inferential tasks, it still struggles with specific tasks such as sequence labeling [Qin et al., 2023]. Some researchers carry out a comprehensive evaluation of ChatGPT's robustness from adversarial and out-of-distribution (OOD) perspectives with findings indicate that the model's absolute performance is far from ideal, suggesting that adversarial and OOD robustness remains a significant challenge for foundational models [Wang et al., 2023d]. In this paper, we assess the performance of GPT-3.5, ChatGPT, GPT-4, and Bing Chat by examining their response accuracy ans their ability of evaluating correct responses on the NQ and TQ datasets within the Open-QA task and QA-Evaluation.

# 3 Open Question Answering (Open-QA)

The Open-QA task mandates that for any given open-domain question $q$, the model $\mathcal{M}$ must generate a corresponding answer $\hat{a}$ while the golden answers denotes as $A$. For models utilizing a retriever-reader mechanism, a database $D$ consisting documents is also accessible for information retrieval.

## 3.1 Data

Following [Lewis et al., 2020, Izacard and Grave, 2021], we use the Natural Questions (NQ) [Kwiatkowski et al., 2019] and TriviaQA (TQ) [Joshi et al., 2017] for the Open-QA task. Specifically, we use the development set of NQ and a portion of the test set of TQ. There are 3610 and 2000 cases in the NQ split and TQ split, respectively.

Given that certain questions have answers that change over time, such as 'Who is the current US president?', and some question pairs have answers that are evidently incorrect, we filter out these instances from the dataset. The quantity of the remaining dataset is shown in Table 1.

It is noteworthy that some gold standard answers contain inaccuracies. Some may include factual errors, and we retain these answers and base our evaluations upon them. Conversely, when the errors are structure-related or format-related or other severe ones, we exclude these from our evaluation process. We list four examples with incorrect golden in Section Appendix C.3.1

Additionally, there are instances where the LLMs refuse to answer certain questions. We record these refusals as answers as well.

## 3.2 Open-QA Models

There are two methods currently being employed in this field.

The first approach involves using pretrained language models to generate answers, leveraging their internal knowledge learned from a vast amount of text data. These models generate answers directly, without the need for external databases or retrieval systems [Roberts et al., 2020, Wang et al., 2021, Ye et al., 2022].

The second approach comprises of a two-tiered system: a retriever and a reader [Izacard and Grave, 2021, Yu et al., 2022, Wang et al., 2023e]. The retriever module fetches pertinent information from a predefined database, while the reader uses this fetched data to generate an appropriate answer. Notable representatives of the retriever module include Dense Passage Retrieval (DPR) [Karpukhin et al., 2020] and BM25, and for the reader module include Fusion-in-Decoder (FiD) [Izacard and Grave, 2021, 2020, Raffel et al., 2020] and Retrieval-Augmented Generation (RAG) [Lewis et al., 2020]. In this study, we adopt both of these approaches, to increase the data amount and the variety of models, which have their own advantages

**Retriever-Reader Models** For this task, we adopt the DPR model [Karpukhin et al., 2020] as the retriever model and the FiD [Izacard and Grave, 2021] model as the reader model. We make a detailed

Table 1: Statistics of data for Open-QA and QA-Eval. We annotate the results of Open-QA as the EVOUNA dataset for QA-Eval. In each cell, the right is the amount of original samples while the left the remaining amount after the filtration. We only annotate the remained data.

| | NaturalQuestions | | TriviaQA | |
|---|---|---|---|---|
| | remained | original | remained | original |
| DPR + FiD | 3020 | 3610 | 1938 | 2000 |
| GPT-3.5 | 3020 | 3610 | 1938 | 2000 |
| ChatGPT-3.5 | 3020 | 3610 | 1938 | 2000 |
| GPT-4 | 3020 | 3610 | 1938 | 2000 |
| Bing Chat | 3020 | 3610 | 1938 | 2000 |

introduction to the architectures of DPR and FiD in the Appendix Section D.1. For the DPR+FiD model, we first use a publicly available DPR checkpoint[4] to retrieve 100 passages for each question. Subsequently, we train a FiD model from a T5-large model [Raffel et al., 2020] with the retrieved 100 passages for each question using the FiD source code[5] and the suggested hyper-parameters.

**Large Language Models**    We directly use the question as a prompt, feeding it into the model to generate an answer $\hat{a}$:

$$\hat{a} = \mathcal{M}_{llm}(q) \qquad (1)$$

Where $\mathcal{M}_{llm}$ could be GPT-3.5 (text-davinci-003), ChatGPT-3.5 (gpt-3.5-turbo), ChatGPT-4, or Bing Chat. We obtain the answers using either the API or the webpage. For instance, for GPT-3.5 and ChatGPT-3.5, we utilize the OpenAI text-davinci-003 and gpt-3.5-turbo APIs, respectively, and we set the temperature to 0 to ensure consistent outputs. For ChatGPT-4 and Bing Chat, we use their respective webpages. The full Open-QA experiments for GPT-3.5 and ChatGPT-3.5 were conducted from April 15 to April 17, 2023. For ChatGPT-4 and Bing Chat, the Open-QA experiments were primarily conducted in April 2023 due to their daily access limit.

### 3.3 Evaluation Methods

We employ three representative methods for evaluating the correctness of Open-QA systems' responses: the lexical matching, the Neural-evaluation, and the large language model.

**Lexical Matching**    We utilize the traditional lexical match method as it is popular in the Open-QA evaluation [Chen et al., 2017, Izacard and Grave, 2021, Lewis et al., 2020]. We follow [Izacard and Grave, 2021, Lewis et al., 2020] to use the Exact Match method for answers generated by the DPR+FiD model. If the generated answer $\hat{a}$ exactly matches one golden answer $a \in A$, we classify it as correct, otherwise as incorrect. Since LLM-generated answers are typically long, the Exact Match is not applicable, so if at least one golden answer $a \in A$ appears in the AI-generated answer $\hat{a}$, we classify it as correct, otherwise as incorrect.

**Large Language Models**    Large Language Models (LLMs) have exhibited impressive linguistic capabilities, indicating significant ability in the assessment of QA results. Consequently, we adopt them into this task. We design a prompt filled with a question $q$, an AI-generated answer $\hat{a}$, and a list of golden answers $A$, and then feed it into the LLM to obtain the prediction $\hat{y}$:

$$\hat{y} = \mathcal{M}_{llm}(prompt) \qquad (2)$$

where $prompt$ = "Here is a question, a set of golden answers (split with /), an AI-generated answer. Can you judge whether the AI-generated answer is correct according to the question and golden answers, simply answer Yes or No." + 'Question: '+$q$+'; ' + 'Golden Answers: ' + $A$+'; '+ 'AI-generated answer: '+$\hat{a}$+'; '+'A:". In this work, we utilize GPT-3.5 (text-davinci-003) as $\mathcal{M}_{llm}$.

We obtain the judgement using the OpenAI text-davinci-003 API for GPT-3.5, and we set the temperature to 0 to ensure consistent outputs. The experiments is conducted on April 24, 2023.

---

[4]$dl.fbaipublicfiles.com/FiD/data/nq\_passages.tar.gz$
[5]$github.com/facebookresearch/FiD$

Table 2: Inter-annotator agreement for different subsets of EVOUNA. The left is for NaturalQuestions (NQ) subsets while the right is for TriviaQA (TQ) subsets.

| NQ subsets | Cohen's Kappa score | TQ subsets | Cohen's Kappa score |
|---|---|---|---|
| NQ-Fid | 91.1 | TQ-FiD | 100 |
| NQ-GPT35 | 92.4 | TQ-GPT35 | 98.4 |
| NQ-ChatGPT35 | 90.4 | TQ-ChatGPT35 | 96.6 |
| NQ-ChatGPT4 | 88.8 | TQ-ChatGPT4 | 99.1 |
| NQ-BingChat | 86.4 | TQ-BingChat | 99.2 |

Table 3: An illustrative example from the EVOUNA dataset. This example includes responses from FiD, GPT-3.5, ChatGPT-3.5, ChatGPT-4 and BingChat models (We omit some details due to space constraints). The question in focus is *"where do the greasers live in the outsiders?"*, with the correct answer being *"Tulsa, Oklahoma"*. The table presents the responses generated by different models and the corresponding human judgments on the accuracy of these responses.

| Models | Generated Answer | Human Judgement |
|---|---|---|
| FiD | Tulsa, Oklahoma | correct |
| GPT35 | The Greasers live in a poor part of town called the East Side. They live in rundown houses, abandoned buildings, and alleys. | incorrect |
| ChatGPT35 | The greasers live in the East Side of town in The Outsiders... | incorrect |
| ChatGPT4 | In the novel "The Outsiders" by S.E. Hinton... | correct |
| BingChat | According to , The Outsiders takes place in Tulsa, Oklahoma in the 1960s. The greasers live on the poorer East Side of town... | correct |

**Neural Evaluation Methods** play a pivotal role in gauging the efficacy of NLG tasks. One such model deployed is BERT-score [Zhang* et al., 2020]. It has been applied for evaluating several machine generation tasks, including machine translation [Zhang* et al., 2020], dialogue [Sellam et al., 2020] and summarization [Zhao et al., 2020]. In the context of QA-Eval, we use BERT-score as our neural-evaluation mechanism. We have also considered BART-Score [Yuan et al., 2021] and GPT-Score [Fu et al., 2023], but they are inapplicable since they provide a continuous score that measures the similarity between the generated text and the reference text. It doesn't explicitly differentiate between correct and incorrect answers in a binary fashion.

BERT-Score evaluates the similarity between two text sequences, typically between a reference and a hypothesis. We take the reference as the concatenation of a question ($q$) and the golden answer ($A$), and the hypothesis is the concatenation of the same question ($q$) and the AI-generated answer ($\hat{a}$). The BERT-score is computed using contextualized word embeddings from a pre-trained BERT model. We introduce the detailed description of the BERT-score algorithm as Sec D.2 For the BERT-Score approach, we set the threshold $\tau$ at 0.5, considering it as the most natural choice.

## 4 The EVOUNA Dataset

The section gives detailed information about the EVOUNA dataset for the QA-Eval task. The EVOUNA dataset is constructed from the results of different Open-QA models, including FiD, GPT-3.5, ChatGPT-3.5/4 and BingChat, on Natural Questions (NQ) and TriviaQA (TQ) with their human annotations. This dataset consists of various components that are divided based on the original dataset and the generator model used, and they are well-detailed in Table 1.

Formally, in the QA-Eval task, an evaluating model $\mathcal{M}$ is presented with an open-domain question $q$, an AI-generated answer $\hat{a}$, and a collection of gold standard answers $A$. The task asks the model to evaluate the correctness of the AI-produced answer in relation to the gold standard responses. The prediction $\hat{y}$ should be either positive (indicating correctness) or negative (indicating incorrectness). The task of QA-Eval in this context is seen as a binary classification task, and the performance of evaluators is quantified using two metrics: accuracy and F1 score. Table 3 presents a representative example of NQ subsets of the EVOUNA dataset. This example illustrates the process where different

Table 4: Human evaluated accuracy and Lexical Matching score of AI-models on NaturalQuestions (NQ) and TriviaQA (TQ). In each cell, the left is the human evaluated accuracy while the right the the lexical match score.

| | NaturalQuestions | | TriviaQA | |
| --- | --- | --- | --- | --- |
| | Human | LexicalMatch | Human | LexicalMatch |
| DPR + FiD | 68.9 | 59.2 | 81.5 | 73.5 |
| GPT-3.5 | 65.5 | 50.7 | 78.4 | 71.0 |
| ChatGPT-3.5 | 73.0 | 57.9 | 84.5 | 76.7 |
| ChatGPT-4 | 78.8 | 61.8 | 90.2 | 82.1 |
| Bing Chat | 79.9 | 65.4 | 89.6 | 81.6 |

Table 5: Performance of Eval-Models on the EVOUNA. In each cell, the left is the accuracy while the right is the Macro-F1.

| | NQ-FiD | NQ-GPT35 | NQ-ChatGPT35 | NQ-ChatGPT4 | NQ-BingChat |
| --- | --- | --- | --- | --- | --- |
| Lexical Matching | 89.7/92.0 | 84.8/86.9 | 80.3/84.9 | 82.5/87.6 | 82.3/87.8 |
| BERT-Score | 75.1/83.5 | 69.5/77.6 | 72.8/81.2 | 76.0/84.3 | 67.5/77.6 |
| GPT-3.5 | 93.6/95.3 | 84.1/87.2 | 82.2/86.9 | 80.9/86.9 | 69.5/77.2 |
| GPT-4 | 94.5/96.0 | 91.0/93.2 | 90.6/93.7 | 92.0/95.1 | 91.4/94.7 |
| Another Human | **96.3/97.4** | **96.8/97.8** | **95.6/96.5** | **96.6/97.9** | **95.5/97.2** |

on EVOUNA-NaturalQuestions

| | TQ-FiD | TQ-GPT35 | TQ-ChatGPT35 | TQ-ChatGPT4 | TQ-BingChat |
| --- | --- | --- | --- | --- | --- |
| Lexical Matching | 91.8/94.7 | 92.3/94.8 | 92.3/95.2 | 91.1/94.8 | 89.8/94.1 |
| BERT-Score | 65.5/75.1 | 75.7/84.1 | 80.8/88.4 | 93.5/90.5 | 80.4/88.3 |
| GPT-3.5 | 95.7/97.3 | 91.2/94.2 | 92.5/95.5 | 92.4/95.7 | 80.9/88.2 |
| GPT-4 | 97.3/98.3 | 97.5/98.4 | 97.5/98.5 | 97.8/98.8 | 96.5/98.1 |
| Another Human | **100/100** | **99.4/99.6** | **98.8/99.2** | **99.8/99.2** | **99.8/99.9** |

on EVOUNA-TriviaQA

models provide answers to the same question, and then a human judge determines the accuracy of each generated answer.

In addition, we normalize answers produced by BingChat. Firstly, we remove their special symbols and referential sources to avoid any potential influence on the performance of evaluating models. Secondly, some Bing Chat answers contain an extra question at the end, such as 'Do you want to know more about xx?', which might induce the evaluating models to answer the question instead of providing judgments. Hence, we also remove these end questions. An example of processing is shown in Section C.1

**Human Annotation**  The human annotation process for our study was conducted by the authors themselves, eliminating the need for external paid services. The process included the careful removal of inappropriate questions and thorough evaluation of the models' generated responses' correctness. To ensure consistency and precision in the annotation process, we established detailed guidelines, a portion of which can be found in Appendix C.2.

Additionally, we evaluate the inter-annotator agreement on 500 samples from each subset of EVOUNA. The Cohen's Kappa scores [Cohen, 1960] representing inter-annotator agreement for these evaluations are presented in Table 2. With all scores above 80, this indicates strong alignment and agreement among our annotations.

## 5   Experiments

### 5.1   Human Evaluation of Open-QA

The Open-QA results are displayed in Table 4. It's observed that results of the commonly-used lexical match metric on each model's outputs do not align with the accuracy assessed by human evaluators. Moreover, the relative ranking between models also diverge significantly between the

Table 6: Evaluation scores assigned by various evaluation models to different QA models on EVOUNA. In each cell, the left represents the score given by the evaluator (row) to the QA model's performance (column) on the respective dataset, while the value in parentheses is the relative ranks among the five models.

| | NQ-FiD | NQ-GPT35 | NQ-ChatGPT35 | NQ-GPT4 | NQ-BingChat |
|---|---|---|---|---|---|
| Lexical Matching | 59.2 (3) | 50.7 (5) | 57.9 (4) | 61.8 (2) | 65.4 (1) |
| BERT-Score | 82.5 (1) | 70.9 (4) | 71.9 (3) | 74.5 (2) | 64.8 (5) |
| GPT-3.5 | 67.3 (1) | 58.8 (4) | 63.0 (3) | 66.2 (2) | 54.1 (5) |
| GPT-4 | 68.2 (4) | 67.3 (5) | 75.8 (3) | 83.0 (2) | 83.3 (1) |
| Human | 69.7 (4) | 70.3 (3) | 63.0 (5) | 80.3 (1) | 78.2 (2) |

on EVOUNA-NaturalQuestions

| | TQ-FiD | TQ-GPT35 | TQ-ChatGPT35 | TQ-GPT4 | TQ-BingChat |
|---|---|---|---|---|---|
| Lexical Matching | 73.5 (4) | 71.0 (5) | 76.7 (3) | 82.1 (1) | 81.6 (2) |
| BERT-Score | 56.8 (5) | 73.9 (4) | 82.1 (2) | 84.4 (2) | 78.0 (3) |
| GPT-3.5 | 79.6 (3) | 72.2 (5) | 81.0 (2) | 85.8 (1) | 72.8 (4) |
| GPT-4 | 81.2 (4) | 78.0 (5) | 84.4 (3) | 90.7 (1) | 89.3 (2) |
| Human | 74.0 (4) | 72.6 (5) | 76.7 (3) | 86.8 (1) | 85.2 (2) |

on EVOUNA-TriviaQA

lexical match and humans, implying that lexical match metric is not suitable for evaluating Open-QA results, especially those generated by LLMs.

Furthermore, ChatGPT-4 and BingChat models show superior performance compared to the other three models on both the Natural Questions (NQ) and TriviaQA (TQ) datasets. However, even these top-performing models, ChatGPT-4 and BingChat, do not achieve perfect accuracy, with scores approximately 80% on NQ and 86% on TQ. This indicates that certain questions continue to present challenges. DPR+FiD, GPT-3.5 and ChatGPT-3.5 demonstrate comparable performance on both datasets. Additional analysis of Open-QA results can be found in Appendix Section E.1.

## 5.2 Evaluating QA Evaluators on EVOUNA

Table 5 shows the evaluation performance of different models, namely Lexical Matching, BERT-Score, GPT-3.5, and Another Human (used as a reference), on different subsets of the EVOUNA datasets. These subsets are identified by the generator model used to create them, including NQ-FiD, NQ-GPT35, NQ-ChatGPT35, NQ-ChatGPT4, NQ-BingChat, and their TQ equivalents. Besides, we also present the precision and recall performance in the Appendix Table 8.

A few key observations can be made from the data presented in the table 5:

**BERT-Score Analysis:** The performance of the BERT-Score model is generally lower compared to other models, more noticeably on the TQ datasets. This could imply that the BERT-Score methodology, which utilizes pre-trained language models for embedding comparisons, might struggle to adequately capture the intricate details and sophistication of answer quality, specifically when the AI-generated answers provide more information than the gold standard.

**GPT-3.5 Performance:** The performance of the GPT-3.5 model is reasonably good across all datasets. However, there is notable variation in its performance depending on the dataset, which underscores the impact of the specific dataset on the evaluation capability of this model.

**GPT-4 Performance Evaluation:** In comparative assessments across two distinct datasets, GPT-4 demonstrates superior performance relative to other automated evaluators. Its effectiveness marginally trails that of human evaluators, underscoring its robust capability in accurately assessing the correctness of AI-generated responses.

**Human Evaluation:** As we mentioned in Section 4, we also conduct an inter-annotator agreement analysis. This secondary evaluation yielded an accuracy and F1 score exceeding 95%, demonstrating superior consistency over all employed AI methods. This result is in line with expectations, as human evaluators, with their innate understanding and assessment capabilities, tend to outperform AI models

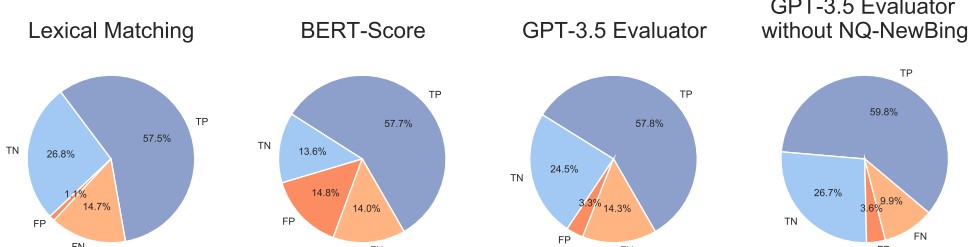

Figure 1: Distribution of outcomes for three evaluators on EVOUNA-NQ. Each pie chart aggregates results across EVOUNA-NQ subsets, showing proportions of TP, TN, FP, and FN for each evaluator.

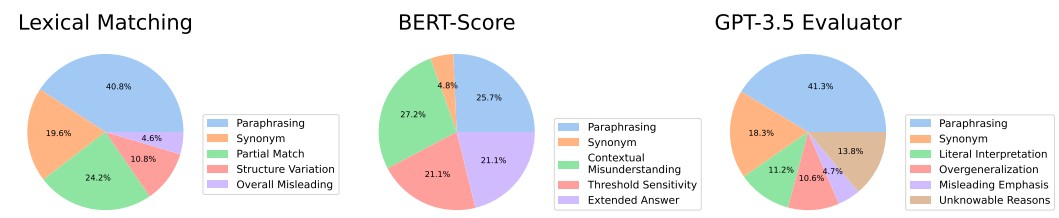

Figure 2: Distribution of error types for the three evaluators on the EVOUNA-NQ dataset, based on manual classification. Each pie chart segment represents the proportion of a specific error type for that evaluator.

in accurately gauging the quality of answers. Thus, human evaluators provide an essential benchmark against which the performance of these AI models can be measured.

**Assigned Scores:** The table 6 displays the evaluation scores that different evaluators, including lexical match, BERT-Score, GPT-3.5, GPT-4, and human (as a reference), give to various QA-models across both NQ and TQ subsets of EVOUNA. The scores illuminate the relative effectiveness of different QA models as evaluated by different evaluators. It's noteworthy that the relative ranks given by evaluators to different QA models vary, indicating evaluators are still not capable to judge the relative level of different models on Open-QA. For example, for NQ, human evaluators rank NQ-BingChat second highest while GPT-3.5 ranks it lowest. These discrepancies show only GPT-4 ranks QA model outputs on TQ in the same relative ranks as humans, revealing the complexity and nuance in open-domain QA model evaluation.

In summary, these findings highlight the challenges and complexities involved in evaluating answer quality in open-domain question answering systems. Despite these difficulties, the results also imply that GPT-4, while a significant advancement, may still be a less effective alternative compared to human evaluators when it comes to judging the outputs of LLMs on factual question-answering tasks.

## 6 Analysis

### 6.1 Distributions of QA-Eval Results

We explore the evaluation results of various evaluators, including Lexical Matching, Neural Evaluation (BERT-Score), and LLM-as-evaluator (GPT-3.5) and illustrate the results (gathering all subsets) of three evaluators on the EVOUNA-NQ dataset using pie charts in Figure 1. Notably, both Lexical Matching and GPT-3.5 exhibit low proportions of False Positives, indicating they rarely misclassify incorrect answers as correct. In contrast, BERT-Score evenly distributes its errors between the two types. Lexical Matching maintains consistent performance across all EVOUNA-NQ subsets, while GPT-3.5 struggles specifically with the BingChat subset. This discrepancy is likely due to BingChat answers containing extraneous information and unique formatting, which may disrupt the LLM's performance. Lexical Matching remains largely unaffected by these factors. Excluding BingChat data significantly improves GPT-3.5's performance.

In the following sections, we delve deeper into the specific errors made by each evaluator and explore the underlying reasons.

## 6.2 Error Analysis in QA-Eval

We begin by delving into the inherent limitations of the three evaluator types, considering both their intrinsic mechanisms and the observed error cases. Due to space limit, a comprehensive discussion of these limitations is provided in Section E.4.1 of the Appendix.

Based on those limitations, we have designed a set of fine-grained Evaluator Error categories. This includes two common errors found across all evaluators, namely Paraphrasing Error and Synonym Error, as well as specific errors unique to each type of evaluator. In detail, Lexical Matching has Partial Match Error, Structure Variation Error and Overall Misleading Error; Neural Evaluation has Contextual Misunderstanding Error, Threshold Sensitivity and Extended Answer Error; LLM evaluators have Literal Interpretation Error, Literal Interpretation Error, Overgeneralization Error, Misleading Emphasis Error and Unknowable Reasons Error. The detailed definition and examples of Error Categories can be found in Section E.4.2 in the Appendix.

Based on the Error Categories, we have manually classified the errors produced by Lexical Matching, BERT-Score, and GPT-3.5 on each subset of our EVOUNA-NQ. For each subset, we selected 100 errors. We present a unified result (gathering all subsets) in Figure 2. The more detailed result can be found in Table 11 in the Appendix.

## 6.3 Insights from QA-Eval Results

Drawing from each evaluator's limitations and error classifications results, we offer these insights:

**Lexical Matching:** While lexical matching remains a simple and effective method for Open-QA evaluation, it struggles with issues of limited contextual understanding, low recall, and structural variations. It often marks answers that humans consider correct as incorrect, but rarely does the opposite. This makes Lexical Matching a strict metric, suitable for environments requiring high error recall. When it does mark a human-considered correct answer as wrong, it's usually because the generated answer contains the golden answer, but the overall meaning doesn't support it. For instance, it might negate the golden answer or only use it as part of the response. Lexical Matching struggles with "Structure Variation" errors. For example, if the golden answer is "8 September 2010" and the generated answer is "Amnesia: The Dark Descent was released on September 8, 2010.", Lexical Matching can't recognize it. The other two evaluators rarely have this issue. Due to its inability to handle semantics, it can't manage Synonym or Contextual understanding situations.

**Neural Evaluation (BERT-Score and BLEURT in this analysis):** Overall, they aren't well-suited for this QA-Eval task, with the poorest performance among the three types. They can only measure the similarity between two text segments. So, they handle Synonym errors well. However, if the generated answer contains extra information (common with larger models), this can easily influence the BERT-score. BERT-Score isn't great at Contextual Understanding. If the generated answer explains the golden answer without including its entities, BERT-Score can easily get it wrong. Adapting BERT-Score to this QA-Eval task by adjusting the threshold is another issue. Many datasets are highly sensitive to threshold settings.

**LLM-evaluator (GPT-3.5 in this analysis):** Overall, the LLM-evaluator can serve as a complement to lexical matching and is valuable for assessing the accuracy of generated answers, it remains sensitive to prompts and the impact of additional contexts, especially for BingChat answers. Its most common error is the "Paraphrasing error", possibly because it's easily influenced by other contexts. It has its own issues, like the "Overgeneralization error", which doesn't appear in the other two evaluators, although they are minor concerns. Sometimes, the LLM-evaluator makes clear mistakes that humans wouldn't. For example, for the question, "Who was the first chief minister of West Bengal?" with the golden answer being "Prafulla Chandra Ghosh", the generated answer was "The first Chief Minister of West Bengal was Dr. Bidhan Chandra Roy." GPT-3.5 marked the generated answer as correct, even though Dr. Bidhan Chandra Roy is not Prafulla Chandra Ghosh. This might be because the evaluator uses its inherent knowledge, overlooking the golden answer, or for other undetermined reasons. Such issues don't appear with the other two evaluators.

Table 7: GPT-3.5 evaluator performance with different prompt strategies on the EVOUNA-NQ set. Each cell displays accuracy (left) and F1 score (right).

|  | NQ-FiD | NQ-GPT35 | NQ-ChatGPT35 | NQ-ChatGPT4 | NQ-BingChat |
|---|---|---|---|---|---|
| Original | 93.6/95.3 | 84.1/87.2 | 82.2/86.9 | 80.9/86.9 | 69.5/77.2 |
| Ignoring Background | 93.9/95.5 | 82.9/86.0 | 80.8/85.5 | 79.6/85.7 | 65.7/73.4 |
| Giving Reasons | 89.6/91.9 | 76.2/78.4 | 73.3/78.2 | 64.3/71.2 | 55.6/62.2 |
| Chain-of-Thoughts | 90.2/92.9 | 84.0/88.0 | **84.5/89.4** | **86.0/91.2** | **80.4/87.1** |
| In-Context-Learning | 93.1/94.9 | **84.5/88.3** | 83.4/88.1 | 83.2/88.6 | 75.3/82.5 |

In summary, while lexical matching and LLM-evaluators are relatively more effective than neural-evaluations, they still underperform compared to human evaluators, often misjudging correct samples. Each evaluator has its own strengths and weaknesses.

### 6.4 Enhancing QA-Eval through Prompt Engineering

We also examine strategies to improve LLM' (specifically, GPT-3.5) performance in QA-Eval via prompt engineering. Four distinct methods were explored: Ignoring Background Information; Providing Reasons for Judgments; Chain of Thoughts [Wei et al., 2022]; In-Context Learning [Dong et al., 2023].

Table 12 outlines the specific prompts used for each method with GPT-3.5 in QA-Eval. The prompts are designed to elicit different model behaviors or responses.

We adopt an approach from Auto-Cot [Zhang et al., 2023] using K-Means clustering [Hartigan and Wong, 1979] to select representative examples for in-context learning. To avoid data leakage, we employ cross-domain clustering; we cluster NQ sets for TQ experiments and vice versa. For example, we select representative examples from NQ-ChatGPT4 for experiments on TQ-ChatGPT4. Four representative examples are chosen for each dataset.

Table 7 presents the performance of GPT-3.5 evaluator with different prompts on the EVOUNA-NQ dataset. Here are the insights: Directing GPT-3.5 to ignore the background information degrades performance on four datasets with long answers (NQ-GPT35/ChatGPT35/ChatGPT4/BingChat). Requiring the model to reason its judgments negatively impacts performance across all datasets. The effects of Chain-of-Thoughts and In-Context-Learning vary. For instance, both methods significantly improve performance on four datasets with long answers, but Chain-of-Thoughts shows a substantial decline on the NQ-FiD. This variability suggests that the influence of these techniques depends on the data distribution.

## 7 Conclusion

In this study, we developed the EVOUNA dataset, crafted with the specific intention of evaluating Open-QA system outputs, with an emphasis on large language models. Our critical observation was the apparent deficiency in existing evaluators - from traditional lexical match metrics to neural-evaluation models and large language models - in providing reliable evaluations for these outputs. The EVOUNA dataset offers a robust tool for comprehensive scrutiny of Open-QA models. We've examined the strengths and weaknesses of each evaluator type within our QA-Eval task and have manually categorized the errors they produce on our dataset.

## Limitations

Our study comes with a few limitations. Firstly, our data, sourced via OpenAI's API or webpage, is subject to frequent model updates which precludes full reproducibility. Secondly, during our main research phase, we faced limitations due to restrictions on the OpenAI GPT-4 API. This meant we couldn't collect a large amount of open-QA data using GPT-4, nor could we apply GPT-4 in our QA-Eval experiments. It was only after our paper was accepted that we gained access to the GPT-4 API. Consequently, we conducted additional experiments for both Open-QA and QA-Eval using GPT-4. However, to maintain the original structure of our work, we limited our analysis of the GPT-4 results and did not extensively modify the paper based on these late-stage findings. As the gold

standard answers in the Natural Questions and TriviaQA datasets occasionally contain inaccuracies, our dataset also carries the risk of inadvertently disseminating misinformation since we are not able to completely get rid of them.

## Acknowledgement

This publication has emanated from research conducted with the financial support of the Pioneer and "Leading Goose" R&D Program of Zhejiang under Grant Number 2022SDXHDX0003.

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

## A  Appendix

1. Submission introducing new datasets must include the following in the supplementary materials:

    (a) Dataset documentation and intended uses. Recommended documentation frameworks include datasheets for datasets, dataset nutrition labels, data statements for NLP, and accountability frameworks.

    (b) URL to website/platform where the dataset/benchmark can be viewed and downloaded by the reviewers.

    (c) Author statement that they bear all responsibility in case of violation of rights, etc., and confirmation of the data license.

    (d) Hosting, licensing, and maintenance plan. The choice of hosting platform is yours, as long as you ensure access to the data (possibly through a curated interface) and will provide the necessary maintenance.

2. To ensure accessibility, the supplementary materials for datasets must include the following:

    (a) Links to access the dataset and its metadata. This can be hidden upon submission if the dataset is not yet publicly available but must be added in the camera-ready version. In select cases, e.g when the data can only be released at a later date, this can be added afterward. Simulation environments should link to (open source) code repositories.

    (b) The dataset itself should ideally use an open and widely used data format. Provide a detailed explanation on how the dataset can be read. For simulation environments, use existing frameworks or explain how they can be used.

    (c) Long-term preservation: It must be clear that the dataset will be available for a long time, either by uploading to a data repository or by explaining how the authors themselves will ensure this.

    (d) Explicit license: Authors must choose a license, ideally a CC license for datasets, or an open source license for code (e.g. RL environments).

    (e) Add structured metadata to a dataset's meta-data page using Web standards (like schema.org and DCAT): This allows it to be discovered and organized by anyone. If you use an existing data repository, this is often done automatically.

    (f) Highly recommended: a persistent dereferenceable identifier (e.g. a DOI minted by a data repository or a prefix on identifiers.org) for datasets, or a code repository (e.g. GitHub, GitLab,...) for code. If this is not possible or useful, please explain why.

3. For benchmarks, the supplementary materials must ensure that all results are easily reproducible. Where possible, use a reproducibility framework such as the ML reproducibility checklist, or otherwise guarantee that all results can be easily reproduced, i.e. all necessary datasets, code, and evaluation procedures must be accessible and documented.

4. For papers introducing best practices in creating or curating datasets and benchmarks, the above supplementary materials are not required.

## B  Data License and Maintenance Plan

The EVOUNA data we create is open sourced at `https://github.com/wangcunxiang/QA-Eval`. The test data is under Apache License 2.0. We plan to collect more data from various datasets, including Natural Questions, Trivia and WebQuestions, and open source it for future research.

## C  Data

### C.1  An example of Processing BingChat Answer

Raw Answer:

The revolution period of Venus around the sun is **224.7 Earth days** [1]. Is there anything else you would like to know about Venus?

Source: Conversation with Bing, 2023/3/31(1) Planet Venus: 20 interesting facts about the scorching world. https://www.space.com/15988-venus-planet-weird-facts.html Accessed

2023/3/31. (2) What Is The Revolution Period Of Venus In Earth Years? Here's The .... https://www.odysseymagazine.com/what-is-the-revolution-period-of-venus-in-earth-years/ Accessed 2023/3/31. (3) What Is Venus's Revolution Period in Earth Days? | Sciencing. https://sciencing.com/venuss-revolution-period-earth-days-2411.html Accessed 2023/3/31. (4) Top 7 what is the revolution period of venus in earth years 2022. https://entornoit.com/top-7-what-is-the-revolution-period-of-venus-in-earth-years-2022/ Accessed 2023/3/31. (5) The length of a day on Venus keeps changing | Astronomy.com. https://astronomy.com/news/2020/01/the-length-of-a-day-on-venus-keeps-changing Accessed 2023/3/31. (6) Your Age on Other Worlds | Exploratorium. https://www.exploratorium.edu/ronh/age/ Accessed 2023/3/31.

Processed Answer:

The revolution period of Venus around the sun is 224. 7 Earth days.

## C.2 Human Annotation Guidelines

Here is a question, a set of golden answers (split with /), an AI-generated answer. You are required to judge (1) whether the question have answers that change over time, simply annotate Yes or No; (2) whether the golden answer contain severe errors; (3) whether the AI-generated answer is correct according to the question and golden answers, simply annotate Yes or No.

Here is a set of guidelines for task (1) whether the question have answers that change over time:

- If the question is clearly time-sensitive, then it is Yes.
- If there are words closely related to the current time node such as "this year", "last year", "next time" and "last time" in this question, then it is Yes.
- If the question contains values that change over decades, such as "who is the player with the most goals in the World Cup so far", then it is Yes.
- If the question contains values that do not change in decades, such as "what is the tallest mountain in the world", then it is No.

If the answer to task (1) is Yes, skip to the next.

Here is a set of guidelines for task (2) whether the golden answer contain severe errors:

- If the golden answer has structure errors, then it is Yes.
  Example: Question: the south west wind blows across nigeria between? Golden: till September
- If the golden answer is obviously not what is asked, then it is Yes.
- If the golden answer has format errors, then it is Yes.
  Example: Question: what season does bart bass die in gossip girl? Golden: (
- If the golden answer has only factual errors, then it is No.

(We also present some examples shown in Section C.3.1.) If the answer to task (2) is Yes, skip to the next.

Here is a set of guidelines for task (3):

- If the question specifies a number (e.g., names of four people), and the response does not meet this requirement (e.g., provides only one name), the answer is deemed incorrect.
- Spelling errors in the responses are considered mistakes. For example, if "golden answer" is misspelled as "gloden answer," the response is marked as incorrect.
- For questions related to specific times, such as "When was the term social justice first used?" a response of "1840s" would be considered correct. However, if the answer needs to be precise to a specific day, month, and year, each time component needs to be factually accurate for the response to be marked as correct.
- For location-based queries, like "Where was Oak Island filmed?", a response of "Canada" would be deemed correct. But, if the answer requires specific details like state, city, or county, each geographical component must be accurate for the answer to be considered correct.

- If there is a direct answer and subsequent explanation in the response, then only focus on whether the direct answer is correct, not whether the subsequent explanation is correct

These guidelines were strictly followed to maintain the reliability and validity of the evaluation process.

### C.3 Supplements to the Annotation

**AI-generated answers.**

For local-deployed models (DPR+FiD) and models can be accessed with APIs (text-davinci-003 for GPT-3.5 and gpt-3.5-turbo for ChatGPT-3.5), we generate the answers locally. For models that can only be interacted within he webpage, including ChatGPT-4 (we do not have API permissions) and BingChat, we ask the annotators to get the answer by interacting in the webpage and make judgement for the three tasks.

**Data assignment.**

We ask one annotator to judge samples with answers generated by DPR+FiD, GPT-3.5 and ChatGPT-3.5; one for samples by ChatGPT-4; one for samples by Bing Chat, for convenience.

**Improper questions or goldens.** If a sample has an improper question or improper goldens, we mark the sample as improper. Since we have three different annotators to judge improper questions and goldens, if at least two annotators mark the improper result as True, we mark it as True, then we ask the left annotator (if there exists) to re-annotate the sample.

#### C.3.1 Golden Error Examples

Here some examples whose golden answer has obvious mistake. The first two have factual errors while the next one has the structure error and the last one has format error.

Question: was star wars a book or a movie first? Golden: film

Question: what is the democracy of the united states? Golden: federal republic

Question: the south west wind blows across nigeria between? Golden: till September

Question: what season does bart bass die in gossip girl? Golden: (

## D Methods

### D.1 DPR and FiD

The DPR model retrieves relevant documents from all given documents to answer a specific question. Given a question $q$ and a database $D$ with each document denoted as $d$, the DPR model comprises two main components: the question encoder $Q_{enc}$ and the document encoder $D_{enc}$. Both typically rely on neural networks, such as BERT [Devlin et al., 2019].

The question encoder $Q_{enc}$ maps a question $q$ to a dense vector representation $q_{emb} = Q_{enc}(q)$, and the document encoder $D_{enc}$ maps each document $d$ in the database $D$ to a dense vector representation $d_{emb} = D_{enc}(d)$.

We compute the similarity between the question embedding $q_{emb}$ and each document embedding $d_{emb}$ using the dot product:

$$s(d, q) = q_{emb} \cdot d_{emb} \tag{3}$$

Documents in the database $D$ are ranked based on their similarity scores, and the top $k$ most relevant documents $D_k$ are retrieved. These documents are then used as input for the reader model $\mathcal{M}_{reader}$ to generate an answer $\hat{a}$ to the question $q$:

$$\hat{a} = \mathcal{M}_{reader}(q, D_k) \tag{4}$$

Table 8: Performance of Eval-Models on EVOUNA. In each cell, the left is the precision while the right is the recall.

| | NQ-FiD | NQ-GPT35 | NQ-ChatGPT35 | NQ-ChatGPT4 | NQ-BingChat |
|---|---|---|---|---|---|
| Lexical Matching | 99.6/85.4 | 99.5/77.1 | 96.0/76.2 | 99.7/78.1 | 97.6/79.8 |
| BERT-Score | 76.7/91.7 | 74.7/80.8 | 81.9/80.6 | 86.8/82.0 | 86.6/70.2 |
| GPT-3.5 | 96.5/94.2 | 92.2/82.7 | 93.8/81.0 | 95.1/79.9 | 95.7/64.8 |
| GPT-4 | 96.6/95.5 | 92.0/94.4 | 91.9/95.5 | 92.7/97.6 | 92.8/96.8 |
| Another Human | 98.5/96.3 | 97.8/97.8 | 97.8/95.3 | 99.0/96.8 | 98.7/95.8 |
| | | | on EVOUNA-NaturalQuestions | | |
| | TQ-FiD | TQ-GPT35 | TQ-ChatGPT35 | TQ-ChatGPT4 | TQ-BingChat |
| Lexical Matching | 100/90.0 | 99.8/90.3 | 100/90.8 | 99.5/90.6 | 98.7/89.9 |
| BERT-Score | 91.5/63.7 | 86.6/81.6 | 89.7/87.2 | 93.6/97.6 | 94.9/82.6 |
| GPT-3.5 | 98.5/96.1 | 98.2/90.5 | 97.5/93.5 | 98.1/93.4 | 98.4/80.0 |
| GPT-4 | 98.5/98.1 | 98.7/98.1 | 98.5/98.5 | 98.5/99.1 | 98.3/97.9 |
| Another Human | 100/100 | 99.4/99.7 | 98.9/99.5 | 99.8/100 | 99.8/100 |
| | | | on EVOUNA-TriviaQA | | |

## D.2 BERT-Score

Given a reference $r = A$ and a hypothesis $h = \hat{a}$, we first obtain their contextualized word embeddings using a pre-trained BERT model:

$$
\begin{aligned}
E_r &= \text{BERT}(r), \\
E_h &= \text{BERT}(h)
\end{aligned}
\tag{5}
$$

Next, we compute the cosine similarity between each token in the reference and each token in the hypothesis:

$$
S_{i,j} = \frac{E_{r_i} \cdot E_{h_j}}{|E_{r_i}||E_{h_j}|}
\tag{6}
$$

We then find the optimal token matchings using the maximum cosine similarity:

$$
\begin{aligned}
P_r &= \frac{1}{|r|} \sum_{i=1}^{|r|} \max_{j=1}^{|h|} S_{i,j}, \\
P_h &= \frac{1}{|h|} \sum_{j=1}^{|h|} \max_{i=1}^{|r|} S_{i,j}
\end{aligned}
\tag{7}
$$

Finally, the BERT-score is calculated as the F1 score between the reference and hypothesis:

$$
\text{BERT-score} = \frac{2 \cdot P_r \cdot P_h}{P_r + P_h}
\tag{8}
$$

To decide whether the AI-generated answer is positive or not, we set a threshold $\tau$ and classify the prediction $\hat{y}$ as positive if the BERT-score is above the threshold and as negative otherwise:

$$
\hat{y} = \begin{cases} \text{Positive,} & \text{BERT-score} >= \tau \\ \text{Negative,} & \text{BERT-score} < \tau \end{cases}
\tag{9}
$$

## E  Analysis

### E.1  Additional Analysis for Open-QA

From the Table 4, we have several additional observations:

Table 9: The Proportions of Evaluation Outcomes Across Three Evaluators on the EVOUNA-NQ Dataset.

|  | True Positive | True Negative | False Positive | False Negative |
|---|---|---|---|---|
| Lexical Matching | 57.5 | 26.8 | 1.1 | 14.7 |
| BERT-Score | 57.7 | 13.6 | 14.8 | 14.0 |
| GPT-3.5 Evaluator | 57.8 | 24.5 | 3.3 | 14.3 |
| GPT-3.5 Evaluator without NQ-BingChat | 59.8 | 26.7 | 3.6 | 9.9 |
| GPT-4 Evaluator | 70.3 | 21.6 | 5.2 | 2.9 |

All models perform better on TriviaQA compared to Natural Questions. This might suggest that the TriviaQA dataset, which is known for its trivia-style questions, is more aligned with the kind of diverse and general knowledge these models have been trained on. In contrast, the Natural Questions dataset, which is derived from real Google search queries, might contain more complex or niche questions that are challenging for the models.

**GPT-3.5 vs ChatGPT-3.5** : These two models have comparable performance, both achieving approximately 65-70% accuracy on NQ and around 80% on TQ. This similarity is expected, as they are versions of the same base model, with the main difference being that ChatGPT is fine-tuned specifically for conversational contexts.

**GPT-4 vs GPT-3.5 and ChatGPT-3.5** : The newer model GPT-4 significantly outperforms both GPT-3.5 and ChatGPT-3.5 on both datasets. This suggests that the improvements incorporated into GPT-4, likely including a larger model size and potentially refined training techniques, have resulted in substantial gains in question answering performance.

**ChatGPT-4 vs BingChat** : These two models exhibit the highest performance on both datasets. Their performance is remarkably similar, with GPT-4 outperforming Bing Chat by only a small margin on both datasets. This suggests that the two models, despite potentially having quite different architectures and training procedures, have reached similar levels of proficiency in question answering.

**LLMs vs. Retrieval-based Methods** : The DPR+FiD model, a representative of traditional retrieval-based methods, performs comparably to the earlier language models (GPT-3.5 and ChatGPT-3.5), but falls behind the newer ones (ChatGPT-4 and Bing Chat). This indicates that while retrieval-based methods remain competitive, the newer generation of language models have surpassed them in terms of question answering capability. This could be due to the ability of these large models to better understand and generate natural language, enabling them to generate more accurate and contextually appropriate answers.

### E.2 Supplemental Analysis for QA-Eval

Table 8 showcases the performance of various evaluation models on EVOUNA-NaturalQuestions and EVOUNA-TriviaQA datasets. The reported metrics are precision and recall.

Looking at the EVOUNA-NaturalQuestions results, we observe that Lexical Matching and GPT-3.5 evaluation models achieve high precision across all QA models. However, the Lexical Matching model tends to have lower recall compared to GPT-3.5. BERT-Score has relatively lower precision but delivers better recall, indicating its ability to identify relevant answers but with a higher false positive rate. Human evaluation, as expected, provides near-perfect precision and recall scores.

For the EVOUNA-TriviaQA results, a similar pattern is observed. Lexical Matching, GPT-3.5, and human evaluation maintain high precision across all QA models. BERT-Score sees a drop in precision but has comparable recall, especially with the TQ-ChatGPT35 and TQ-ChatGPT4. Again, human evaluation shows nearly perfect performance.

The results underscore the different strengths of the evaluation models: Lexical Matching for precision, BERT-Score for recall, and GPT-3.5 and human evaluation for both. However, all models' performance varies with the dataset and QA model, emphasizing the importance of multiple evaluation methods for comprehensive assessment.

Table 10: Distribution of error types across different generative models on the NQ-test dataset. Each cell represents the proportion of the respective error type to *all responses* generated by the model.

| | InAcc | InCom | IrrA | OutInf | MisQs | Others |
|---|---|---|---|---|---|---|
| DPR + FiD | 25.0 | 3.0 | 0.9 | 1.2 | 1.7 | 0.0 |
| GPT-3.5 | 25.3 | 5.4 | 0.3 | 2.1 | 1.8 | 0.1 |
| ChatGPT-3.5 | 23.2 | 7.9 | 0.5 | 1.4 | 2.4 | 0.2 |
| GPT-4 | 13.3 | 2.8 | 0.3 | 1.2 | 1.3 | 0.0 |
| Bing Chat | 9.5 | 7.6 | 1.3 | 1.3 | 0.8 | 0.5 |

### E.3 Error Analysis in Open-QA

We classify the errors in the Open-QA scenario into several distinct categories:

- Inaccurate Information (InAcc): These errors occur when the model's response, while relevant to the question, contains inaccuracies.

- Incomplete Answer (InCom): This type of error is characterized by the model providing pertinent information but failing to fully address the question.

- Irrelevant Answer (IrrA): The model's response bears no relevance to the posed question.

- Outdated Information (OutInf): These errors occur when the model provides information that was correct at some point in the past but is no longer valid or applicable.

- Misinterpretation of the Question (MisQs): This category includes errors where the model misinterprets the question's intent or context.

- Other Errors: This catch-all category includes any errors that don't fit into the above classifications.

To perform this error classification, we initially used ChatGPT-4 to conduct a preliminary categorization of the Open-QA error data. Subsequently, human annotators were engaged to review and correct the classification results. The finalized results are represented in Table 10.

Analyzing the data reveals several interesting patterns. Notably, Bing Chat appears to have the highest rate of 'Incomplete Answer' errors, suggesting that while it generally understands the question, it often fails to provide a comprehensive answer. However, it also has the lowest rate of 'Inaccurate Information' errors, implying that the quality of the information it provides is usually high.

Conversely, DPR + FiD, GPT-3.5, and ChatGPT-3.5 all have similar rates of 'Inaccurate Information' errors, indicating a potential challenge in maintaining accuracy for these models. GPT-4 seems to outperform the other models in both 'Inaccurate Information' and 'Incomplete Answer' errors, suggesting an overall improvement in the quality and completeness of its responses.

It's also worth noting the relatively low incidence of 'Outdated Information' and 'Misinterpretation of the Question' errors across all models, suggesting that these areas are less problematic in current models.

This error analysis is helpful in identifying the strengths and weaknesses of different models and provides valuable insights into the areas that need further improvements.

### E.4 Error Analysis in QA-Eval

#### E.4.1 Limitations of Each Evaluator

Based on our theoretical analysis and observations of erroneous cases, we identified the following issues with each type of evaluator:

**Lexical Matching**:

- Lack of Semantic Understanding: The exact match metric doesn't take into account the semantic meaning of the answers. It only checks if the predicted answer is exactly the

same as the ground truth, even if the predicted answer is semantically correct but phrased differently.

- Inability to Handle Synonyms: The exact match metric cannot handle synonyms. If the predicted answer uses a different word that has the same meaning as the word in the ground truth answer, the exact match metric will consider it as a wrong answer.

- Inability to Handle Paraphrasing: Similar to the point above, the exact match metric cannot handle paraphrasing. If the predicted answer is a paraphrase of the ground truth answer, the exact match metric will consider it as a wrong answer.

- Inability to Handle Partially Correct Answers: The exact match metric cannot handle partially correct answers. If the predicted answer is partially correct, the exact match metric will consider it as a wrong answer.

- Inability to Handle Reordered Words: The exact match metric cannot handle reordered words. If the predicted answer has the same words as the ground truth answer but in a different order, the exact match metric will consider it as a wrong answer.

- Inability to Handle Different Levels of Detail: The exact match metric cannot handle different levels of detail. If the predicted answer provides more or less detail than the ground truth answer but is still correct, the exact match metric will consider it as a wrong answer.

- Inability to Handle Different Formats: The exact match metric cannot handle different formats. If the predicted answer is in a different format than the ground truth answer (for example, dates or numbers), the exact match metric will consider it as a wrong answer.

These limitations highlight the need for more sophisticated evaluation metrics that can understand the semantic meaning of the answers and handle synonyms, paraphrasing, partially correct answers, reordered words, different levels of detail, and different formats.

**Neural Evaluation**: The limitations of neural evaluation methods, such as BERT-Score and BLEURT, are evident. Most crucially, many neural evaluations are primarily designed to measure the similarity between two phrases or sentences. They are not tailored for binary tasks, especially those assessing the factual correctness of answers. Instead, they provide a continuous score that gauges the similarity between the generated text and the reference text, rendering them directly unsuitable for this particular task. In our study, we employed BERT-score and BLEURT for this task by setting a threshold. However, the performance of both BERT-score and BLEURT was suboptimal. The primary shortcoming of neural evaluations for this task is their misalignment with its requirements.

Furthermore, BERT-score has the following limitations:

- Sensitivity to Verbosity: BERT-score may penalize verbose answers even if they contain the correct information. If the AI-generated answer provides a detailed explanation while the golden answer is concise, the score might be lower than expected.

- Mismatched Focus: If the AI-generated answer is correct but emphasizes different aspects or details than the golden answer, BERT-score might not recognize the similarity, leading to a lower score.

- Lack of Contextual Understanding: BERT-score measures the similarity between embeddings but might not fully capture the contextual nuances of certain answers, especially when there are multiple valid ways to answer a question.

- Synonym and Paraphrasing Issues: BERT-score might not always recognize synonyms or paraphrased answers as being equivalent to the golden answer, leading to potential discrepancies in scoring.

- Threshold Limitations: Setting a fixed threshold (e.g., 0.5) for determining correctness can be arbitrary. Some answers might be just below the threshold but still be correct, while others might be just above but incorrect.

- Doesn't Account for Minor Details: BERT-score might not be sensitive enough to minor inaccuracies in the AI-generated answer, especially if the overall semantic content is similar to the golden answer.

- Lack of Absolute Truth Measure: BERT-score is a relative measure of similarity between two pieces of text. It doesn't provide an absolute measure of the truthfulness or correctness of an answer.

- Influence of Sentence Structure: The structure or order of sentences in the AI-generated answer compared to the golden answer might affect the score, even if the content is the same.
- Generalization Issues: BERT-score is based on pre-trained embeddings. It might not generalize well to niche topics or questions that require specialized knowledge outside of its training data.
- Over-reliance on Embeddings: While embeddings capture semantic information, they might not always capture the nuanced differences between two pieces of text, especially in a QA setting where precision is crucial.

In summary, while BERT-score is a powerful metric for evaluating text similarity, its application in a QA-eval task has limitations.

**GPT-3.5** has its own set of limitations:

- Literal Interpretation: One of the limitations is the model's tendency to interpret questions or golden answers too literally. This can lead to situations where the evaluator fails to recognize correct answers that provide a broader context or a different interpretation that still addresses the core of the question.
- Overgeneralization: Another challenge is the model's propensity to overgeneralize based on its vast training data. This can result in the evaluator deeming an answer as correct even if it doesn't align specifically with the nuances of the question at hand.
- Misleading Emphasis: The evaluator might sometimes be swayed by partial correctness in an answer. If an answer emphasizes certain correct elements, the evaluator might overlook primary claims that are factually incorrect, leading to a misleading evaluation.
- Unknowable Reasoning: There are instances where the evaluator's judgment is puzzling, even to human experts. The model might deem an answer as correct that has no discernible correlation with the golden answer. This limitation underscores the "black-box" nature of deep learning models, where their internal reasoning processes remain opaque.
- Lack of Feedback Mechanism: Especially with closed-source models, there's a lack of a feedback loop to correct or fine-tune the model based on its evaluation errors. This can lead to repeated mistakes or biases in evaluation.
- Sensitivity to Prompt Engineering: Both closed-source and open-source LLMs can be sensitive to the way questions are framed or prompts are constructed. This can introduce variability in the evaluation, where slight rephrasings might lead to different judgments.
- Potential Bias: All LLMs, whether closed or open source, can inherit biases from their training data. In the context of QA-Eval, this might manifest as favoring certain types of answers or being biased against certain topics or contexts.

### E.4.2 Error Categories

Based on the aforementioned limitations, we have designed a set of Evaluator Error categories. This includes two common errors found across all evaluators as well as specific errors unique to each type of evaluator.

**General Error Categories for All Evaluators**

- **Paraphrasing Error**: The evaluator fails to recognize answers that paraphrase the golden answer correctly but do not contain the exact substring.

  Example: Question: "What is the process by which plants convert sunlight into energy?" Golden Answer: "Photosynthesis" Generated Answer: "The mechanism plants use to transform light into energy is termed the photosynthetic process."

  Explanation: the generated answer is a paraphrase of the "Photosynthesis" but does not contain the word directly.

- **Synonym Error**: The evaluator fails to recognize answers that use synonyms or alternative phrasing to convey the same meaning as the golden answer.

Example: Question: "What's another term for a doctor?" Golden Answer: "Physician" Generated Answer: "A medical practitioner."

Explanation: "medical practitioner" is a synonym for "physician" but isn't a direct substring.

**Specific Error Categories for Lexical Matching**

- **Partial Match Error**: The evaluator fails to recognize answers that contain a part of the golden answer but not the entire substring.

  Example: Question: "Who painted the Mona Lisa?" Golden Answer: "Leonardo da Vinci" Generated Answer: "The Mona Lisa was painted by Leonardo."

  Explanation: only "Leonardo" is mentioned, not the full "Leonardo da Vinci".

- **Structure Variation Error**: The evaluator fails to recognize answers that essentially convey the same information as the golden answer but there's a variation in how it's structured.

  Example: Question: "When did 'Amnesia: The Dark Descent' come out?" Golden Answer: "8 September 2010" Generated Answer: "Amnesia: The Dark Descent was released on September 8, 2010."

  Explanation: the date format in the generated answer has an extra comma than the golden answer, even though the information is the same.

- **Overall Misleading Error**: The evaluator mistakenly recognizes the answer as correct because it contains a substring from the golden answer, even if the overall context of the answer is misleading.

  Example: Question: "Who wrote 'The Great Gatsby'?" Golden Answer: "F. Scott Fitzgerald" Generated Answer: "Ernest Hemingway and F. Scott Fitzgerald were close friends, but Hemingway wrote 'The Old Man and the Sea'."

  Explanation: The generated answer contains the substring "F. Scott Fitzgerald", which might lead the Lexical Matching Evaluator to judge it as correct. However, the overall context of the answer is misleading, suggesting a relationship between Hemingway and "The Great Gatsby", which is incorrect.

**Specific Error Categories for Neural Evaluation**

- **Contextual Misunderstanding Error**: The evaluator might misjudge answers based on word embeddings and fail to capture the context in which certain words or phrases are used.

  Example: Question: "Who wrote 'Romeo and Juliet'?" Golden Answer: "William Shakespeare" AI-generated Answer: "Shakespeare wrote many plays."

  Explanation: Even though the AI answer mentions Shakespeare, it doesn't directly answer the question.

- **Threshold Sensitivity**: Answers that are just below the threshold might be correct but are judged as incorrect, and vice versa.

  Example: Question: "What's the capital of France?" Golden Answer: "Paris" AI-generated Answer: "The capital city of France is Paris."

  Explanation: The AI answer is correct but might score just below the threshold due to added context.

- **Extended Answer Error**: The evaluator might penalize answers that provide more context or details than the golden answer, even if they are correct, because the BERT-score only considers the similarities of the candidates and references.

  Example: Question: "Who painted the Mona Lisa?" Golden Answer: "Leonardo da Vinci" AI-generated Answer: "Leonardo da Vinci, a renowned Italian artist, painted the Mona Lisa."

  Explanation: The AI answer provides more context but is still correct.

**Specific Error Categories for LLM-evaluator**

- **Literal Interpretation Error**: The evaluator might take the question or golden answer too literally and fail to recognize correct answers that provide a broader context or interpretation.

Table 11: The error results for Lexical Matching evaluator, BERT-Score evaluator and GPT-3.5 evaluator. Each kind evaluator has common error types and specific error types. General error rate indicates the error proportion of this evaluator on this subset.

| | NQ-FiD | NQ-GPT35 | NQ-ChatGPT35 | NQ-ChatGPT4 | NQ-BingChat |
|---|---|---|---|---|---|
| Paraphrasing Error | 29% | 37% | 29% | 60% | 49% |
| Synonym Error | 18% | 12% | 37% | 12% | 19% |
| Partial Match Error | 48% | 30% | 13% | 10% | 20% |
| Structure Variation Error | 4% | 16% | 15% | 12% | 7% |
| Overall Misleading Error | 1% | 5% | 6% | 6% | 5% |
| Lexical Matching: General error rate | 11.75 | 15.2 | 19.7 | 16.8 | 17.7 |
| | NQ-FiD | NQ-GPT35 | NQ-ChatGPT35 | NQ-ChatGPT4 | NQ-BingChat |
| Paraphrasing Error | 4% | 24% | 29% | 39% | 39% |
| Synonym Error | 4% | 7% | 4% | 5% | 5% |
| Contextual Misunderstanding Error | 63% | 22% | 23% | 20% | 15% |
| Threshold Sensitivity Error | 25% | 33% | 20% | 18% | 15% |
| Extended Answer Error | 4% | 14% | 24% | 18% | 26% |
| BERT-Score: General error rate | 25.0 | 30.5 | 27.2 | 23.2 | 32.4 |
| | NQ-FiD | NQ-GPT35 | NQ-ChatGPT35 | NQ-ChatGPT4 | NQ-BingChat |
| Paraphrasing Error | 16% | 52% | 36% | 52% | 47% |
| Synonym Error | 22% | 12% | 21% | 18% | 17% |
| Literal Interpretation Error | 21% | 4% | 11% | 6% | 13% |
| Overgeneralization Error | 17% | 13% | 8% | 8% | 6% |
| Misleading Emphasis Error | 7% | 2% | 5% | 3% | 6% |
| Unknowable Reasons Error | 17% | 8% | 19% | 13% | 11% |
| GPT3.5: General error rate | 6.4 | 16.0 | 17.8 | 16.6 | 30.5 |

Example: Question: "Which bird is known for its beautiful tail?" Golden Answer: "Peacock" Generated Answer: "Many birds have beautiful tails."

Explanation: The evaluator might take a literal approach and accept the general statement as correct without focusing on the specific bird in question.

- **Overgeneralization Error**: The evaluator might generalize based on its training data and judge an answer as correct even if it's not specific to the question.

Example: Question: "Who wrote 'Pride and Prejudice'?" Golden Answer: "Jane Austen" Generated Answer: "An English author."

Explanation: The evaluator might accept the general answer as it's not technically wrong, even though it lacks specificity.

- **Misleading Emphasis Error**: The evaluator might judge an answer as correct if it includes some correct information and put emphasis on it, and overlook the incorrect primary claim.

Example: Question: "What's the primary gas in Earth's atmosphere?" Golden Answer: "Nitrogen" Generated Answer: "Oxygen, which makes up about 78% of the atmosphere."

Explanation: GPT-3.5 might focus on the correct percentage and overlook incorrect mention of "Oxygen" as a primary gas.

- **Unknowable Reasons**: The evaluator makes an incorrect judgment for an unknowable reason. Even humans cannot figure out why the LLM thinks the generated answer is correct since it has no correlation with the golden answer.

Example: Question: "Who was the first chief minister of West Bengal?" Golden Answer: "Prafulla Chandra Ghosh" Generated Answer: "The first Chief Minister of West Bengal was Dr. Bidhan Chandra Roy."

Explanation: GPT-3.5 takes the generated answer as correct, but Dr. Bidhan Chandra Roy is apparently not Prafulla Chandra Ghosh.

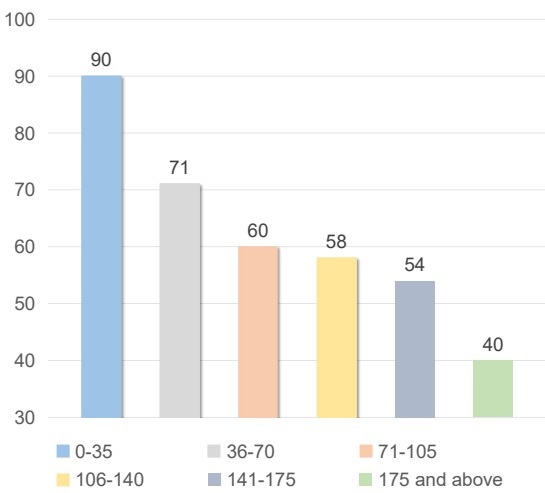

Figure 3: Correlation between the evaluation accuracy of GPT-3.5 and the answer length in tokens across all models.

### E.4.3 Length Analysis on QA-Eval

Figure 3 depicts the relationship between GPT-3.5's evaluation accuracy and the number of tokens present in the answers produced by all models. The token count is segmented into six distinct categories: 0-35, 36-70, 71-105, 106-140, 141-175, and 175 and above. The corresponding accuracy for these ranges are 90, 71, 60, 58, 54, and 40 respectively. Additionally, the average token counts for the answers by each model are as follows: FiD (4.8 tokens), GPT-3.5 (31.4 tokens), ChatGPT (41.9 tokens), GPT-4 (39.9 tokens), and BingChat (49.7 tokens).

We can draw several observations: 1. GPT-3.5's evaluation accuracy exhibits an inverse correlation with the length of the answer. As the number of tokens in the answer escalates, the evaluation accuracy diminishes. This could indicate that GPT-3.5 may struggle to accurately evaluate more extended responses, potentially due to challenges in retaining context or comprehending intricate or unfamiliar constructs in longer text spans. 2. Considering the average token counts, FiD, the model that generates the shortest responses on average (4.8 tokens), would predominantly fall into the 0-35 token range where GPT-3.5 has its peak accuracy (90). This observation could imply that GPT-3.5 would exhibit optimal evaluation performance with responses generated by the FiD model. 3. Conversely, models like Bing Chat, which on average yield longer responses (49.7 tokens), would generally fall into the token ranges where GPT-3.5's evaluation accuracy is lower. This can partially explain why GPT-3.5 performs worse than Lexical Matching in NQ-BingChat and TQ-BingChat.

### E.5 Enhancing QA-Eval through Prompt Engineering

We also examine strategies to improve LLM' (specifically, GPT-3.5) performance in QA-Eval via prompt engineering. Four distinct methods were explored: Ignoring Background Information; Providing Reasons for Judgments; Chain of Thoughts [Wei et al., 2022]; In-Context Learning [Dong et al., 2023].

Table 12 outlines the specific prompts used for each method with GPT-3.5 in QA-Eval. The prompts are designed to elicit different model behaviors or responses.

We adopt an approach from Auto-Cot [Zhang et al., 2023] using K-Means clustering [Hartigan and Wong, 1979] to select representative examples for in-context learning. To avoid data leakage, we employ cross-domain clustering; we cluster NQ sets for TQ experiments and vice versa. For example, we select representative examples from NQ-ChatGPT4 for experiments on TQ-ChatGPT4. Four representative examples are chosen for each dataset.

Table 7 presents the performance of GPT-3.5 evaluator with different prompts on the EVOUNA-NQ dataset. Here are the insights: Directing GPT-3.5 to ignore the background information degrades performance on four datasets with long answers (NQ-GPT35/ChatGPT35/ChatGPT4/BingChat).

Table 12: Specific prompts used in each method for GPT-3.5 on QA-Eval.

| Methods | Prompts |
|---|---|
| Original | Here is a question, a set of golden answers (split with /), an AI-generated answer. Can you judge whether the AI-generated answer is correct according to the question and golden answers, simply answer Yes or No |
| Ignoring Background | Here is a question, a set of golden answers (split with /), an AI-generated answer. Can you judge whether the AI-generated answer is correct according to the question and golden answers, please only consider the answer itself, ignore the background information. Simply answer Yes or No. |
| Giving Reasons | Here is a question, a set of golden answers (split with /), an AI-generated answer. Can you judge whether the AI-generated answer is correct according to the question and golden answers. Please make a judgment and give the reason. Your answer must be <Yes or No>\|<Reason> |
| Chain-of-Thoughts | Here is a question, a set of golden answers (split with /), an AI-generated answer. Can you judge whether the AI-generated answer is correct according to the question and golden answers. Please think step by step and make a judgment in the end. You must give your chain of thoughts. Your answer must be <your chain of thoughts>\|<Yes or No>. (chain of thoughts and final judgment must be split with '\|') |
| In-Context-Learning | Here is a question, a set of golden answers (split with /), an AI-generated answer. Can you judge whether the AI-generated answer is correct according to the question and golden answers, simply answer Yes or No. Here are some examples: Example 1: AAA; Example 2: BBB; Example 3: CCC; Example 4: DDD. |

Requiring the model to reason its judgments negatively impacts performance across all datasets. The effects of Chain-of-Thoughts and In-Context-Learning vary. For instance, both methods significantly improve performance on four datasets with long answers, but Chain-of-Thoughts shows a substantial decline on the NQ-FiD. This variability suggests that the influence of these techniques depends on the data distribution.

## E.6 Does retrieval Help in LLM?

In our quest to determine the impact of retrieval on Large Language Models (LLMs) in an Open-QA setting, we investigate two distinct scenarios. Firstly, we assess the performance of Bing Chat when retrieval is disabled. Secondly, we augment GPT-3.5 with a retrieval mechanism and gauge its effectiveness.

**Performance of Bing Chat Without Retrieval**    In this experiment, we modify the standard prompt fed to Bing Chat by preceding the question $q$ with the instruction "Please do not search, answer the following question directly:". We choose a sample of 500 questions from the NQ test dataset, filtering out those unsuitable for this setting. The results of this experiment are depicted in the left section of Figure 4.

The data suggests a significant decline in Bing Chat's performance when retrieval is disabled, dropping approximately 15 percentage points from 80.5 to 65.6. This is comparable to the performance of GPT-3.5 (65.0), which lacks a retrieval mechanism. This substantial decline implies that the retrieval

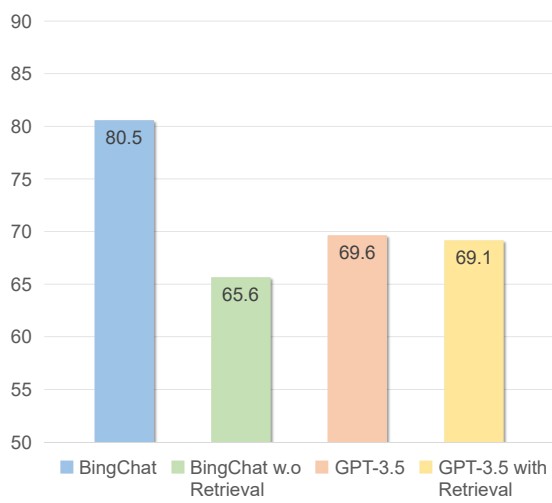

Figure 4: The performance of Bing Chat and GPT-3.5 on NQ set with or without retrieval.

Table 13: Performance of BERT-Score and BLEURT on the EVOUNA. In each cell, the left is the accuracy while the right is the Macro-F1.

| | NQ-FiD | NQ-GPT35 | NQ-ChatGPT35 | NQ-ChatGPT4 | NQ-BingChat |
|---|---|---|---|---|---|
| Lexical Matching | 89.7/92.0 | 84.8/86.9 | 80.3/84.9 | 82.5/87.6 | 82.3/87.8 |
| BERT-Score | 75.1/83.5 | 69.5/77.6 | 72.8/81.2 | 76.0/84.3 | 67.5/77.6 |
| BELURT | 84.4/89.4 | 74.1/83.1 | 78.0/86.3 | 82.2/89.6 | 82.8/90.0 |
| GPT-3.5 | 93.6/95.3 | 84.1/87.2 | 82.2/86.9 | 80.9/86.9 | 69.5/77.2 |
| GPT-4 | 94.5/96.0 | 91.0/93.2 | 90.6/93.7 | 92.0/95.1 | 91.4/94.7 |
| Another Human | **96.3/97.4** | **96.8/97.8** | **95.6/96.5** | **96.6/97.9** | **95.5/97.2** |
| | | on EVOUNA-NaturalQuestions | | | |
| | TQ-FiD | TQ-GPT35 | TQ-ChatGPT35 | TQ-ChatGPT4 | TQ-BingChat |
| Lexical Matching | 91.8/94.7 | 92.3/94.8 | 92.3/95.2 | 91.1/94.8 | 89.8/94.1 |
| BERT-Score | 65.5/75.1 | 75.7/84.1 | 80.8/88.4 | 93.5/90.5 | 80.4/88.3 |
| BELURT | 88.1/92.9 | 82.1/89.4 | 85.2/91.5 | 88.8/93.8 | 90.7/95.0 |
| GPT-3.5 | 95.7/97.3 | 91.2/94.2 | 92.5/95.5 | 92.4/95.7 | 80.9/88.2 |
| GPT-4 | 97.3/98.3 | 97.5/98.4 | 97.5/98.5 | 97.8/98.8 | 96.5/98.1 |
| Another Human | **100/100** | **99.4/99.6** | **98.8/99.2** | **99.8/99.2** | **99.8/99.9** |
| | | on EVOUNA-TriviaQA | | | |

component significantly boosts the performance of the LLM underpinning Bing Chat in an Open-QA context.

**Augmenting GPT-3.5 with a Retrieval Mechanism**   For the second scenario, we employ the same Dense Retriever used in the DPR+FiD model (referenced in Section3.2) to fetch relevant passages from the database for a given question. We then integrate these passages into the prompt supplied to GPT-3.5. The prompt reads: "We have a question here: QUESTION. Now, we have the following relevant passages: PASSAGE 1; PASSAGE 2; PASSAGE 3; PASSAGE 4; PASSAGE 5. Please answer the question referring to the above passages."

The results of this experiment, shown in the right section of Figure 4, reveal a slight decrease in performance with the addition of retrieval, falling from 69.6 to 69.1. This suggests that simply injecting retrieved passages into the prompts, without any form of thoughtful adaptation, does not contribute positively to the LLM's performance in an Open-QA setting.

### E.7   BLEURT Evaluator

Table 14: Open-QA and QA-Eval results of Atlas and Chat-Llama2 on 500 samples of NQ. In each cell, the left is the accuracy while the right is the Macro-F1.

| | NQ-Atlas | NQ-ChatLlama2 |
|---|---|---|
| Lexical Matching | 92.6/91.7 | 93.4/87.9 |
| BERT-Score | 67.1/72.5 | 73.4/61.9 |
| GPT-3.5 | 93.2/92.7 | 77.6/43.0 |
| GPT-4 | 93.2/92.6 | 91.3/85.1 |
| Human Score on NQ-Atlas: 47.9; Human Score on NQ-ChatLlama2: 29.7 | | |

Table 15: Error results of Eval-Models on the EVOUNA. In each cell, the left is the error rates while the right is the times compared with another human results.

| | NQ-FiD | NQ-GPT35 | NQ-ChatGPT35 | NQ-ChatGPT4 | NQ-BingChat |
|---|---|---|---|---|---|
| Lexical Matching | 10.3/2.8x | 15.2/4.8x | 19.7/4.5x | 17.5/5.1x | 17.7/3.9x |
| BERT-Score | 24.9/6.7x | 30.5/9.5x | 27.2/6.2x | 24.0/7.1x | 32.5/7.2x |
| GPT-3.5 | 6.4/1.7x | 15.9/5.0x | 17.8/4.0x | 19.1/ 5.6x | 30.5/ 6.8x |
| GPT-4 | 5.5/1.5x | 9.0/2.8x | 9.4/2.1x | 8.0/2.4x | 8.6/1.9x |
| Another Human | **3.7/1.0x** | **3.2/1.0x** | **4.4/1.0x** | **3.4/1.0x** | **4.5/1.0x** |
| on EVOUNA-NaturalQuestions | | | | | |

| | TQ-FiD | TQ-GPT35 | TQ-ChatGPT35 | TQ-ChatGPT4 | TQ-BingChat |
|---|---|---|---|---|---|
| Lexical Matching | 8.2/∞ | 7.7/6.4xx | 7.7/6.4x | 8.9/44.5x | 10.2/51.0x |
| BERT-Score | 34.5/∞ | 24.3/20.3x | 19.2/16.0x | 6.5/32.5x | 19.6/98.0x |
| GPT-3.5 | 4.3/∞ | 8.8/7.3x | 7.5/6.3x | 7.6/38.0x | 19.1/95.5x |
| GPT-4 | 2.7/∞ | 2.5/4.2x | 2.5/2.1x | 2.2/11x | 3.5/17.5x |
| Another Human | **0/∞** | **0.6/1.0x** | **1.2/1.0x** | **0.2/1.0x** | **0.2/1.0x** |
| on EVOUNA-TriviaQA | | | | | |

We also conducted a QA-Eval analysis on a more recent Neural-Evaluation model, BLEURT [Sellam et al., 2020]. Similar to BERT-Score, we applied a threshold to BLEURT to make it suitable for QA-Eval. In this work, we set the threshold at 0.2 based on observed distributions. The results are shown in the Table 13. Although BLEURT outperforms BERT-Score on most datasets, it still lags significantly behind the performance of Lexical Matching, GPT-3.5 and human, especially in terms of Macro-F1.

### E.8 Additional Open-QA Models

We have conducted experiments on more transparent Open-QA models, including Atlas [Izacard et al., 2022], Llama-2 [Touvron et al., 2023], Chat-Llama-2 [Touvron et al., 2023] on 500 samples on NQ test subset. During our experiments, we notice that the base version of LLaMa-2 occasionally deviated from our instructions. As a result, we chose to proceed with Chat-Llama-2 for a more consistent evaluation. The results are shown in Table 14.

It's evident from the results that the performance of ATLAS and Chat-Llama2 is somewhat below the models discussed in our paper. Moreover, the evaluators' performance on NQ-Atlas and NQ-ChatLlama2 is consistent with the trends observed for the models we initially discussed.

## F Additional Related Work

Hashimoto et al. [2019] have also studied the correlations between human evaluation and automated metrics in NLP. However, there are key differences that set our research apart. First, We only discuss the Open-QA task, underscoring the nuances and challenges specific to this domain, while their research casts a wider net, aiming to bridge the gap between human and automated evaluation methods across various natural language generation tasks. Second, there are different emphasis on Human Evaluation, We introduce the EVOUNA dataset, which is enriched with human-annotated results, providing a fresh perspective on evaluation in the Open-QA domain, while They advocate for a unified framework that correlates human judgments with statistical metrics, offering a holistic

approach to evaluation in NLP. Last, we present the QA-Eval task and the EVOUNA dataset, tailored specifically for evaluating Open-QA systems, while heir research offers a comprehensive framework designed for a broader spectrum of natural language generation tasks.

