| BERT-Score | 25.0/6.8x | 30.5/9.5x | 27.2/6.2x | 23.2/6.8x | 32.4/7.2x |
| GPT-3.5 | 6.4/1.7x | 16.0/5.0x | 17.8/4.0x | 16.6/4.9x | 30.5/6.8x |
| Another Human | **3.7/1.0x** | **3.2/1.0x** | **4.4/1.0x** | **3.4/1.0x** | **4.5/1.0x** |
| on EVOUNA-NaturalQuestions | | | | | |
|  | TQ-FiD | TQ-GPT35 | TQ-ChatGPT35 | TQ-ChatGPT4 | TQ-BingChat |
| Lexical Matching | 10.0/33.3x 8.2/27.3 | 7.7/12.8x | 7.7/6.4x | 8.9/44.5x | 10.2/51.0x |
| BERT-Score | 34.6/115.3x | 24.3/40.5x | 19.3/16.1x | 16.6/83.0x | 19.6/98.0x |
| GPT-3.5 | 4.3/14.3x | 8.8/14.7x | 7.3/6.1x | 7.5/37.5x | 18.8/94.0x |
| Another Human | **0.3/1.0x** | **0.6/1.0x** | **1.2/1.0x** | **0.2/1.0x** | **0.2/1.0x** |
| on EVOUNA-TriviaQA | | | | | |

deviated from our instructions. As a result, we chose to proceed with Chat-Llama-2 for a more consistent evaluation. The results are shown in Table 14.

It's evident from the results that the performance of ATLAS and Chat-Llama2 is somewhat below the models discussed in our paper. Moreover, the evaluators' performance on NQ-Atlas and NQ-ChatLlama2 is consistent with the trends observed for the models we initially discussed.

# F  Additional Related Work

Hashimoto et al. [2019] have also studied the correlations between human evaluation and automated metrics in NLP. However, there are key differences that set our research apart. First, We only discuss the Open-QA task, underscoring the nuances and challenges specific to this domain, while their research casts a wider net, aiming to bridge the gap between human and automated evaluation methods across various natural language generation tasks. Second, there are different emphasis on Human Evaluation, We introduce the EVOUNA dataset, which is enriched with human-annotated results, providing a fresh perspective on evaluation in the Open-QA domain, while They advocate for a unified framework that correlates human judgments with statistical metrics, offering a holistic approach to evaluation in NLP. Last, we present the QA-Eval task and the EVOUNA dataset, tailored specifically for evaluating Open-QA systems, while heir research offers a comprehensive framework designed for a broader spectrum of natural language generation tasks.