# OpenReview forum: "Evaluating Open-QA Evaluation"
_NeurIPS.cc/2023/Track/Datasets_and_Benchmarks — NeurIPS 2023 Datasets and Benchmarks Poster_

### Official Review · Reviewer_LhrH · 2023-06-25

**Rating:** 5
**Confidence:** 3

**Strengths:**

- The evaluation of AI-generated answers in Open-QA is an important task to ensure factuality and improve model performance, and is highly relevant to the large community on LLMs. The paper addresses the limitations of current evaluation methods and introduces a new task and dataset to facilitate the development of more effective automatic evaluation tools.
- The paper provides a good analysis of the limitations of existing evaluation metrics and presents human-annotated results as a benchmark for evaluating different evaluation methods. The findings show the shortcomings of current methods.

**Additional Feedback:**

Questions
- While the paper focuses on factoid question answering tasks like Natural Questions (NQ) and TriviaQA, would it be possible to extend the framework to more open-ended or long-formed QA tasks like ELI5?

**Clarity:**

The paper is generally well-written, but there are areas where clarity could be improved, such as the findings on evaluation results. For instance, the authors could clarify how these findings are new compared to existing works evaluating automated NLP metics and how these findings help future research.

**Correctness:**

The claims made in the paper seem correct. The evaluation methods and experiment design are reasonable for the QA-Eval task. The dataset is constructed based on human annotations, and the evaluation metrics are compared with human evaluation results.

**Documentation:**

The paper provides a documentation of the datasets in the appendix.

**Limitations:**

The authors have discussed the limitations of their work and the potential negative societal impact. They acknowledge the limitations of current evaluation methods and propose improvements.

**Opportunities For Improvement:**

- Comparison with prior work: While the paper mentions prior work on LLMs and Open-QA datasets, a more comprehensive discussion of related work would be beneficial. For instance, there are works that have studied the correlations between human evaluation and automated metrics in NLP (e.g. https://arxiv.org/abs/1904.02792). How the approach and findings introduced in this submission are different and new should be explained more clearly.
- Takeaways and insights for future research: While the submission does a good work on evaluating various evaluation metrics against human evaluation, what actionable insights researchers should get out of this work could be discussed more clearly. Such discussion would inform researchers of the important future directions and how they should use the benchmark/dataset introduced in this submission.

**Relation To Prior Work:**

The paper mentions relevant prior work on LLMs and Open-QA datasets. However, a more extensive discussion is needed to clearly differentiate this work from previous contributions. Please see the “Opportunities For Improvement” section above

**Summary And Contributions:**

This paper introduces the task of Evaluating QA Evaluation (QA-Eval) and presents a corresponding dataset called EVOUNA. The goal is to assess the accuracy of AI-generated answers in Open Question Answering (Open-QA) and evaluate the performance of automatic evaluation methods. The paper investigates different evaluation metrics and explores methods to improve LLM-based evaluators. The contributions include the creation of the QA-Eval task, the EVOUNA dataset, and insights into the limitations and potential improvements of current evaluation methods.

---

> ### Author Response · Authors · 2023-08-20
> **Rebuttal Part 1 (Q1)**
>
> # Opportunities For Improvement
>
> - O1: Comparison with prior work: While the paper mentions prior work on LLMs and Open-QA datasets, a more comprehensive discussion of related work would be beneficial. For instance, there are works that have studied the correlations between human evaluation and automated metrics in NLP (e.g. https://arxiv.org/abs/1904.02792). How the approach and findings introduced in this submission are different and new should be explained more clearly.
>
>
>     R1: Thank you for pointing out the need for a more comprehensive comparison with prior work. While both our paper and the mentioned work (https://arxiv.org/abs/1904.02792) delve into the correlations between human evaluations and automated metrics in NLP, there are key differences that set our research apart:
>     1. **Focus on Evaluation**:
>         - Our Work: We only discuss the Open-QA task, underscoring the nuances and challenges specific to this domain.
>         - Prior Work: Their research casts a wider net, aiming to bridge the gap between human and automated evaluation methods across various natural language generation tasks.
>     2. **Emphasis on Human Evaluation**:
>        - Our Work: We introduce the EVOUNA dataset, which is enriched with human-annotated results, providing a fresh perspective on evaluation in the Open-QA domain.
>        - Prior Work: They advocate for a unified framework that correlates human judgments with statistical metrics, offering a holistic approach to evaluation in NLP.
>     3. **Methodological Differences**:
>        - Our Work: We present the QA-Eval task and the EVOUNA dataset, tailored specifically for evaluating Open-QA systems.
>        - Prior Work: Their research offers a comprehensive framework designed for a broader spectrum of natural language generation tasks.
>
>     In essence, while there are thematic overlaps, the objectives, methodologies, and contributions of the two papers are distinct. We believe that our work adds a unique dimension to the ongoing discourse on evaluation in NLP, particularly in the Open-QA space.
>
>     We also include this related work in the updated paper.

---

> ### Author Response · Authors · 2023-08-20
> **Rebuttal Part 2 (Q2 Part 1)**
>
> - O2: Takeaways and insights for future research: While the submission does a good work on evaluating various evaluation metrics against human evaluation, what actionable insights researchers should get out of this work could be discussed more clearly. Such discussion would inform researchers of the important future directions and how they should use the benchmark/dataset introduced in this submission.
>
>   R2: Thank you for emphasizing the importance of providing actionable insights for future research. To address these concerns, we have conducted the QA-Eval Error Analysis, which consists three parts:
>
> 1. **Categorization of Evaluator Errors**: Recognizing that different evaluators might err in common and unique ways, we established a taxonomy of potential errors. This categorization helps in understanding the nature of mistakes and provides insights into areas of improvement for each evaluator.
> 2. **Empirical Analysis of Errors**: Based on our error taxonomy, we conducted a hands-on analysis of the actual errors produced by each evaluator on our dataset. This empirical study allowed us to validate our theoretical findings and provided a finegrained understanding of where each evaluator might not fit in real-world scenarios.
> 3. **Detailed Analysis of Evaluators**: We delved into the characteristics and  limitations of various evaluators, including Lexical Matching, Neural Evaluation (BERT-Score), and LLM-evaluator (GPT-3.5)
>
> Here are the details for each part:
> ## 1. Error Categories
>
> In addition to characteristic, the strength and limitations of each type of evaluator, we have designed a set of Evaluator Error categories. This includes two common errors found across all evaluators as well as specific errors unique to each type of evaluator.
>
> ### General Error Categories for All Evaluators
>
> | Error Type             | Definition                                                   | Example                                                      |
> | ---------------------- | ------------------------------------------------------------ | ------------------------------------------------------------ |
> | **Paraphrasing Error** | The evaluator fails to recognize answers that paraphrase the golden answer correctly but do not contain the exact substring. | Question: "What is the process by which plants convert sunlight into energy?"<br> Golden Answer: "Photosynthesis"<br> Generated Answer: "The mechanism plants use to transform light into energy is termed the photosynthetic process."<br> Explanation: the generated answer is a paraphrase of the "Photosynthesis" but does not contain the word directly. |
> | **Synonym Error**      | The evaluator fails to recognize answers that use synonyms or alternative phrasing to convey the same meaning as the golden answer. | Question: "What's another term for a doctor?"<br> Golden Answer: "Physician"<br> Generated Answer: "A medical practitioner."<br> Explanation: "medical practitioner" is a synonym for "physician" but isn't a direct substring. |

---

> > ### Author Response · Authors · 2023-08-20
> > **Rebuttal Part 3 (Q2 Part 2)**
> >
> > ### Specific Error Categories
> >
> > | Judger            | Error Type                            | Description                                                  | Example                                                      |
> > | ----------------- | ------------------------------------- | ------------------------------------------------------------ | ------------------------------------------------------------ |
> > | **Lexical Matching** | **Partial Match Error **              | The evaluator fails to recognize answers that contain a part of the golden answer but not the entire substring. | Question: "Who painted the Mona Lisa?"<br> Golden Answer: "Leonardo da Vinci"<br> Generated Answer: "The Mona Lisa was painted by Leonardo."<br> Explanation: only "Leonardo" is mentioned, not the full "Leonardo da Vinci". |
> > |                   | **Structure Variation Error**         | The evaluator fails to recognize answers that essentially convey the same information as the golden answer but there's a variation in how it's structured. | Question: "When did 'Amnesia: The Dark Descent' come out?"<br> Golden Answer: "8 September 2010"<br> Generated Answer: "Amnesia: The Dark Descent was released on September 8, 2010."<br> Explanation: the date format in the generated answer has an extra comma than the golden answer, even though the information is the same. |
> > |                   | **Overall Misleading Error for**      | The evaluator mistakenly recognizes the answer as correct because it contains a substring from the golden answer, even if the overall context of the answer is misleading. | Question: "Who wrote 'The Great Gatsby'?"<br> Golden Answer: "F. Scott Fitzgerald"<br> Generated Answer: "Ernest Hemingway and F. Scott Fitzgerald were close friends, but Hemingway wrote 'The Old Man and the Sea'."<br> Explanation: The generated answer contains the substring "F. Scott Fitzgerald", which might lead the Lexical Match Evaluator to judge it as correct. However, the overall context of the answer is misleading, suggesting a relationship between Hemingway and "The Great Gatsby", which is incorrect. |
> > | **Neural Evlauation** | **Contextual Misunderstanding Error** | The evaluator might misjudge answers based on word embeddings and fail to capture the context in which certain words or phrases are used. | Question: "Who wrote 'Romeo and Juliet'?"<br> Golden Answer: "William Shakespeare"<br> AI-generated Answer: "Shakespeare wrote many plays."<br> Explanation: Even though the AI answer mentions Shakespeare, it doesn't directly answer the question. |
> > |                   | **Threshold Sensitivity**             | Answers that are just below the threshold might be correct but are judged as incorrect, and vice versa. | Question: "What's the capital of France?"<br> Golden Answer: "Paris"<br> AI-generated Answer: "The capital city of France is Paris."<br> Explanation: The AI answer is correct but might score just below the threshold due to added context. |
> > |                   | **Extended Answer Error**             | The evaluator might penalize answers that provide more context or details than the golden answer, even if they are correct, because the BERT-score only considers the similarities of the candidates and references. | Question: "Who painted the Mona Lisa?"<br> Golden Answer: "Leonardo da Vinci"<br> AI-generated Answer: "Leonardo da Vinci, a renowned Italian artist, painted the Mona Lisa."<br> Explanation: The AI answer provides more context but is still correct. |

---

> > > ### Author Response · Authors · 2023-08-20
> > > **Rebuttal Part 4 (Q2 Part 3)**
> > >
> > > | Judger            | Error Type                            | Description                                                  | Example                                                      |
> > > | ----------------- | ------------------------------------- | ------------------------------------------------------------ | ------------------------------------------------------------ |
> > > | **LLM-evaluator** | **Literal Interpretation Error**      | The evaluator might take the question or golden answer too literally and fail to recognize correct answers that provide a broader context or interpretation. | Question: "Which bird is known for its beautiful tail?"<br> Golden Answer: "Peacock"<br> Generated Answer: "Many birds have beautiful tails."<br> Explanation: The evaluator might take a literal approach and accept the general statement as correct without focusing on the specific bird in question. |
> > > |                   | **Overgeneralization Error**          | The evaluator might generalize based on its training data and judge an answer as correct even if it's not specific to the question. | Question: "Who wrote 'Pride and Prejudice'?"<br> Golden Answer: "Jane Austen"<br> Generated Answer: "An English author."<br> Explanation: The evaluator might accept the general answer as it's not technically wrong, even though it lacks specificity. |
> > > |                   | **MIsleading Emphasis Error**         | The evaluator might judge an answer as correct if it includes some correct information and put emphasis on it, and overlook the incorrect primary claim. | Question: "What's the primary gas in Earth's atmosphere?"<br> Golden Answer: "Nitrogen"<br> Generated Answer: "Oxygen, which makes up about 78% of the atmosphere."<br> Explanation: GPT-3.5 might focus on the correct percentage and overlook incorrect mention of "Oxygen" as a primary gas. |
> > > |                   | **Unknowable Reasons**                | The evaluator makes an incorrect judgment for an unknowable reason. Even humans cannot figure out why the LLM thinks the generated answer is correct since it has no correlation with the golden answer. | Question: "Who was the first chief minister of West Bengal?"<br> Golden Answer: "Prafulla Chandra Ghosh"<br> Generated Answer: "The first Chief Minister of West Bengal was Dr. Bidhan Chandra Roy."<br> Explanation: GPT-3.5 takes the generated answer as correct, but Dr. Bidhan Chandra Roy is apparently not Prafulla Chandra Ghosh. |
> > >
> > > While these categories might overlap in certain instances, each possesses its own distinct emphasis. When classifying, we choose the category that best fits each specific case.

---

> > > > ### Author Response · Authors · 2023-08-20
> > > > **Rebuttal Part 5 (Q2 Part 4)**
> > > >
> > > > ## 2. Error Analysis
> > > > Based on the aforementioned Error Categories, we manually classified the errors produced by Lexical Matching, BERT-Score, and GPT-3.5 on each subset of NQ. For each subset, we selected 100 errors. If there were fewer than 100 errors, we included all of them. The classification results are as follows:
> > > > ### The error results for Lexical Match evaluator, BERT-Score evaluator and GPT-3.5 evaluator.
> > > > Each kind evaluator has common error types and specific error types. General error rate indicates the error proportion of this evaluator on this subset.
> > > >
> > > > | Error Type/Model | NQ-FiD | NQ-GPT35 | NQ-ChatGPT35 | NQ-ChatGPT4 | NQ-BingChat |
> > > > |------------------|--------|----------|--------------|-------------|-------------|
> > > > | **Lexical Matching** | | | | | |
> > > > | Paraphrasing Error | 29% | 37% | 29% | 60% | 49% |
> > > > | Synonym Error | 18% | 12% | 37% | 12% | 19% |
> > > > | Partial Match Error | 48% | 30% | 13% | 10% | 20% |
> > > > | Structure Variation Error | 4% | 16% | 15% | 12% | 7% |
> > > > | Overall Misleading Error | 1% | 5% | 6% | 6% | 5% |
> > > > | General error rate | 11.75 | 15.2 | 19.7 | 16.8 | 17.7 |
> > > > | **BERT-Score** | | | | | |
> > > > | Paraphrasing Error | 4% | 24% | 29% | 39% | 39% |
> > > > | Synonym Error | 4% | 7% | 4% | 5% | 5% |
> > > > | Contextual Misunderstanding Error | 63% | 22% | 23% | 20% | 15% |
> > > > | Threshold Sensitivity Error | 25% | 33% | 20% | 18% | 15% |
> > > > | Extended Answer Error | 4% | 14% | 24% | 18% | 26% |
> > > > | General error rate | 25.0 | 30.5 | 27.2 | 23.2 | 32.4 |
> > > > | **GPT-3.5** | | | | | |
> > > > | Paraphrasing Error | 16% | 52% | 36% | 52% | 47% |
> > > > | Synonym Error | 22% | 12% | 21% | 18% | 17% |
> > > > | Literal Interpretation Error | 21% | 4% | 11% | 6% | 13% |
> > > > | Overgeneralization Error | 17% | 13% | 8% | 8% | 6% |
> > > > | Misleading Emphasis Error | 7% | 2% | 5% | 3% | 6% |
> > > > | Unknowable Reasons Error | 17% | 8% | 19% | 13% | 11% |
> > > > | General error rate | 6.4 | 16.0 | 17.8 | 16.6 | 30.5 |

---

> > > > > ### Author Response · Authors · 2023-08-20
> > > > > **Rebuttal Part 6 (Q2 Part 5)**
> > > > >
> > > > > ## **Key Insights**
> > > > > Drawing from each evaluator's limitations and error classifications results, we offer these insights:
> > > > > 1. Lexical Matching:
> > > > >    - While lexical matching remains a simple and effective method for Open-QA evaluation, it struggles with issues of limited contextual understanding, low recall, and structural variations.
> > > > >    - It often marks answers that humans consider correct as incorrect, but rarely does the opposite. This makes Lexical Matching a strict metric, suitable for environments requiring high error recall.
> > > > >    - When it does mark a human-considered correct answer as wrong, it's usually because the generated answer contains the golden answer, but the overall meaning doesn't support it. For instance, it might negate the golden answer or only use it as part of the response.
> > > > >    - Lexical Matching struggles with "Structure Variation" errors. For example, if the golden answer is "8 September 2010" and the generated answer is "It was released on September 8, 2010.", Lexical Matching can't recognize it. The other two evaluators rarely have this issue.
> > > > >    - Due to its inability to handle semantics, it can't manage Synonym or Contextual understanding situations.
> > > > > 2. Neural Evaluation (BERT-Score and BLEURT in this work):
> > > > >    - Overall, they aren't well-suited for this QA-Eval task, with the poorest performance among the three types. They can only measure the similarity between two text segments. So, they handle Synonym errors well. However, if the generated answer contains extra information (common with larger models), this can easily influence the BERT-score.
> > > > >    - BERT-Score isn't great at Contextual Understanding. If the generated answer explains the golden answer without including its entities, BERT-Score can easily get it wrong.
> > > > >    - Adapting BERT-Score to this QA-Eval task by adjusting the threshold is another issue. Many datasets are highly sensitive to threshold settings.
> > > > > 3. LLM-evaluator (GPT-3.5 in this work):
> > > > >    - Overall, the LLM-evaluator can serve as a complement to lexical matching and is valuable for assessing the accuracy of generated answers, it remains sensitive to prompts and the impact of additional contexts, especially for bingchat answers.
> > > > >    - Its most common error is the "Paraphrasing error", possibly because it's easily influenced by other contexts.
> > > > >    - It has its own issues, like the "Overgeneralization error", which doesn't appear in the other two evaluators, although they are minor concernss.
> > > > >    - Sometimes, the LLM-evaluator makes clear mistakes that humans wouldn't. For example, for the question, "Who was the first chief minister of West Bengal?" with the golden answer being "Prafulla Chandra Ghosh", the generated answer was "The first Chief Minister of West Bengal was Dr. Bidhan Chandra Roy." GPT-3.5 marked the generated answer as correct, even though Dr. Bidhan Chandra Roy is not Prafulla Chandra Ghosh. This might be because the evaluator uses its inherent knowledge, overlooking the golden answer, or for other undetermined reasons. Such issues don't appear with the other two evaluators.
> > > > >
> > > > > In general, while lexical matching is a straightforward and effective method for Open-QA evaluation, it encounters challenges related to limited contextual understanding, low recall, and structural variations.
> > > > > The LLM-evaluator can be effectively complementary to lexical matching, and is valuable in assessing the accuracy of generated answers, but it is not robust and particularly sensitive to prompts and the influence of additional contexts, especially in the case of BingChat answers.
> > > > > The application of neural evaluations in the QA-Eval task requires further exploration.

---

> > > > > > ### Author Response · Authors · 2023-08-20
> > > > > > **Rebuttal Part 7 (Q2 Part 6)**
> > > > > >
> > > > > > ##  3. Characteristics and Limitations of Each Evaluator
> > > > > > Based on our theoretical analysis and observations of erroneous cases, we identified the following issues with each type of evaluator:
> > > > > >
> > > > > > 3.1. Lexical Matching:
> > > > > >
> > > > > >    - Lack of Semantic Understanding: The exact match metric doesn't take into account the semantic meaning of the answers. It only checks if the predicted answer is exactly the same as the ground truth, even if the predicted answer is semantically correct but phrased differently.
> > > > > >    - Inability to Handle Synonyms: The exact match metric cannot handle synonyms. If the predicted answer uses a different word that has the same meaning as the word in the ground truth answer, the exact match metric will consider it as a wrong answer.
> > > > > >    - Inability to Handle Paraphrasing: Similar to the point above, the exact match metric cannot handle paraphrasing. If the predicted answer is a paraphrase of the ground truth answer, the exact match metric will consider it as a wrong answer.
> > > > > >    - Inability to Handle Partially Correct Answers: The exact match metric cannot handle partially correct answers. If the predicted answer is partially correct, the exact match metric will consider it as a wrong answer.
> > > > > >    - Inability to Handle Reordered Words: The exact match metric cannot handle reordered words. If the predicted answer has the same words as the ground truth answer but in a different order, the exact match metric will consider it as a wrong answer.
> > > > > >    - Inability to Handle Different Formats: The exact match metric cannot handle different formats. If the predicted answer is in a different format than the ground truth answer (for example, dates or numbers), the exact match metric will consider it as a wrong answer.
> > > > > > These limitations highlight the need for more sophisticated evaluation metrics that can understand the semantic meaning of the answers and handle synonyms, paraphrasing, partially correct answers, reordered words, different levels of detail, and different formats.
> > > > > >
> > > > > >
> > > > > > 3.2 Neural Evaluation:
> > > > > >
> > > > > >    - The limitations of neural evaluation methods, such as BERT-Score and BLEURT, are evident. Most crucially, many neural evaluations are primarily designed to measure the similarity between two phrases or sentences. They are not tailored for binary tasks, especially those assessing the factual correctness of answers. Instead, they provide a continuous score that gauges the similarity between the generated text and the reference text, rendering them directly unsuitable for this particular task. In our study, we employed BERT-score and BLEURT for this task by setting a threshold. However, the performance of both BERT-score and BLEURT was suboptimal. The primary shortcoming of neural evaluations for this task is their misalignment with its requirements.
> > > > > >    - Furthermore, BERT-score has the following limitations:：
> > > > > >    - Sensitivity to Verbosity: BERT-score may penalize verbose answers even if they contain the correct information. If the AI-generated answer provides a detailed explanation while the golden answer is concise, the score might be lower than expected.
> > > > > >    - Mismatched Focus: If the AI-generated answer is correct but emphasizes different aspects or details than the golden answer, BERT-score might not recognize the similarity, leading to a lower score.
> > > > > >    - Lack of Contextual Understanding: BERT-score measures the similarity between embeddings but might not fully capture the contextual nuances of certain answers, especially when there are multiple valid ways to answer a question.
> > > > > >    - Synonym and Paraphrasing Issues: BERT-score might not always recognize synonyms or paraphrased answers as being equivalent to the golden answer, leading to potential discrepancies in scoring.
> > > > > >    - Threshold Limitations: Setting a fixed threshold (e.g., 0.5) for determining correctness can be arbitrary. Some answers might be just below the threshold but still be correct, while others might be just above but incorrect.
> > > > > >    - Doesn't Account for Minor Details: BERT-score might not be sensitive enough to minor inaccuracies in the AI-generated answer, especially if the overall semantic content is similar to the golden answer.
> > > > > >    - Lack of Absolute Truth Measure: BERT-score is a relative measure of similarity between two pieces of text. It doesn't provide an absolute measure of the truthfulness or correctness of an answer.
> > > > > >    - Influence of Sentence Structure: The structure or order of sentences in the AI-generated answer compared to the golden answer might affect the score, even if the content is the same.
> > > > > > In summary, while BERT-score is a powerful metric for evaluating text similarity, its application in a QA-eval task has severe limitations.

---

> > > > > > > ### Author Response · Authors · 2023-08-20
> > > > > > > **Rebuttal Part 8 (Q2 Part 7)**
> > > > > > >
> > > > > > > 3.3 LLM (GPT-3.5 in this work) :
> > > > > > >
> > > > > > >    - Sensitivity to Prompt Engineering: Both closed-source and open-source LLMs can be sensitive to the way questions are framed or prompts are constructed. This can introduce variability in the evaluation, where slight rephrasings might lead to different judgments.
> > > > > > >    - Literal Interpretation: One of the limitations is the model's tendency to interpret questions or golden answers too literally. This can lead to situations where the evaluator fails to recognize correct answers that provide a broader context or a different interpretation that still addresses the core of the question.
> > > > > > >    - Overgeneralization: Another challenge is the model's propensity to overgeneralize based on its vast training data. This can result in the evaluator deeming an answer as correct even if it doesn't align specifically with the nuances of the question at hand.
> > > > > > >    - Misleading Emphasis: The evaluator might sometimes be swayed by partial correctness in an answer. If an answer emphasizes certain correct elements, the evaluator might overlook primary claims that are factually incorrect, leading to a misleading evaluation.
> > > > > > >    - Unknowable Reasoning: There are instances where the evaluator's judgment is puzzling, even to human experts. The model might deem an answer as correct that has no discernible correlation with the golden answer. This limitation underscores the "black-box" nature of deep learning models, where their internal reasoning processes remain opaque.
> > > > > > >    - Potential Bias: All LLMs, whether closed or open source, can inherit biases from their training data. In the context of QA-Eval, this might manifest as favoring certain types of answers or being biased against certain topics or contexts
> > > > > > >
> > > > > > > **(We have included those analysis, including Error Categories, Error Analysis and Characteristics and Limitations of Each Evaluator into the Section 6.1 **Error Analysis in QA-Eval** and Appendix Section E4 **Error Analysis in QA-Eval** in the updated paper.)**

---

> ### Author Response · Authors · 2023-08-20
> **Rebuttal Part 9 (Clarity, Relation To Prior Work)**
>
> # Clarity:
> The paper is generally well-written, but there are areas where clarity could be improved, such as the findings on evaluation results. For instance, the authors could clarify how these findings are new compared to existing works evaluating automated NLP metrics and how these findings help future research.
>
> R3: same with R2
>
>
> # Relation To Prior Work:
> The paper mentions relevant prior work on LLMs and Open-QA datasets. However, a more extensive discussion is needed to clearly differentiate this work from previous contributions. Please see the “Opportunities For Improvement” section above
>
> R4: same with R2

---

> ### Author Response · Authors · 2023-08-20
> **Rebuttal Part 10 (Q5)**
>
> # Questions
> - Q5: While the paper focuses on factoid question answering tasks like Natural Questions (NQ) and TriviaQA, would it be possible to extend the framework to more open-ended or long-formed QA tasks like ELI5?
>
>     R5: Thank you for raising the question about the potential applicability of our framework to long-formed QA tasks like ELI5. While our current research primarily targets factoid question answering tasks, there are several reasons for not directly extending it to more open-ended tasks:
>
>     - **Scope of Open-QA**: We mainly follow prior work of DPR (https://arxiv.org/abs/2004.04906), FiD (https://arxiv.org/abs/2007.01282), RAG (https://arxiv.org/pdf/2005.11401.pdf) and etc, in the field. As such, our primary datasets of interest were Natural Questions and TriviaQA. While the challenges of evaluating subjective questions and long-formed QA are indeed significant, we believe they warrant separate, dedicated research and have thus earmarked them for future work.
>
>     - **Complexity of Long Answers**: Long-formed QA tasks produce extensive answers, each containing a wealth of information. Determining the factual accuracy of such lengthy responses poses a significant challenge, especially when it comes to annotation. Addressing this challenge would require a comprehensive approach, which goes beyond the scope of our current paper.
>
>     - **Evaluation Metrics**: The primary focus of our paper is to address the limitations of the exact match metric in factoid QA. Long-formed QA, on the other hand, often relies on similarity-based evaluation rather than exact matching. This difference in evaluation criteria makes the direct application of our findings to long-formed QA less straightforward.
>
>     - **Annotation Guidelines**: While developing our annotation guidelines, we encountered several unresolved issues, especially concerning the evaluation of factual accuracy in long answers. These challenges further underscore the complexity of extending our framework to long-formed QA.
>
>
>     In conclusion, while the evaluation of long-formed QA's factual accuracy is a promising avenue for future research, it falls outside the purview of our current paper. We have claimed the scope of our Open-QA task in the introduction of the revised paper.

---

### Official Review · Reviewer_Hse5 · 2023-07-18
**Overall, the meta evaluation of the evaluation techniques is a crucial topic within the landscape of NLP and can impact many adjacent directions. However, another iteration of the work with detailed analysis and in-depth understanding of various techniques will be beneficial for the paper.**

**Rating:** 7
**Confidence:** 4

**Strengths:**

1. Framing the evaluation question as a yes/no question is a good way to get a well calibrated answer.
2. Analyzing and identifying findings such as which language models favor small length answers helps researchers in picking the right LM as evaluator for their use case and token lengths.

**Additional Feedback:**

Minor grammatical / spelling corrections:
1. Line 27: “Hense” -> “Hence”.
2. Line 33: “includes” -> “including”.
3. Line 34: “includes” -> “including”.
4. Line 120: “variety of and models” -> “variety of models”.
5. Line 156: “conducted in April 24” -> “conducted on April 24”.
6. Line 171: “The section give detailed” -> “This section gives detailed”.

**Clarity:**

Questions whose answers are temporally dependent or evolve over time have been removed from the datasets for evaluation. However, it is not clearly mentioned how these questions are identified for removal (the statistics post removal are mentioned in the paper).

**Correctness:**

It seems like the lexical match did not align well with the human evaluation accuracies, however, it is concluded that lexical match is a good metric. The inefficiency of direct lexical match is also accepted in the field and the findings do not compare with prior claims on this.

**Documentation:**

The details about the dataset are clearly described.

**Ethics:**

There are no direct concerns from the contributions of the paper.

**Limitations:**

1. Large language models are very sensitive to prompts when used for evaluation. The models are sensitive to the order of the options presented along with the question prompt. The large language models setup is not sufficiently well-studied to conclude the results.
2. The language models are subject to biases and using them directly for evaluation can lead to unwanted biases in the system that can impact further development of models or datasets.

**Opportunities For Improvement:**

Most findings are relatively surface level and an in-depth analysis of the evaluation methods might help with the resourcefulness of the paper.

**Relation To Prior Work:**

Kadavath, Saurav, et al. "Language models (mostly) know what they know." arXiv preprint arXiv:2207.05221 (2022).

**Summary And Contributions:**

The traditional metrics to evaluate open question answering have their limitations with respect to the flexibility of the answers. This work evaluates these evaluation methods such as exact match and proposes a QA-Eval dataset EVOUNA set for evaluating Open-QA evaluation. This dataset is used to conduct experiments are Natural Questions and TriviaQA. The experiments are conducted on 2 types of models – generating from the knowledge of pretrained models and retrieval based answer prediction models. Evaluations are conducted with lexical match, neural evaluation and language model based approaches.

Main contributions:
1. The paper presents a new dataset EVOUNA that has questions, options, gold standard answer and the AI generated answer with the associated task of predicting whether the AI generated answer is appropriate.
2. Different families of evaluators, lexical match, language model-based, neural evaluation approaches are compared and an analysis of the findings of different scenarios are presented.

---

> ### Author Response · Authors · 2023-08-20
> **Rebuttal Part 1 (Q1 Part 1)**
>
> # Opportunities For Improvement
>
> - Q1: Most findings are relatively surface level and an in-depth analysis of the evaluation methods might help with the resourcefulness of the paper.
>
>
> R1: Thank you for your feedback regarding the depth of our analysis. We appreciate the opportunity to clarify the depth and scope of our research. ITo address the concern, we have conducted the QA-Eval Error Analysis, which consists three parts:
>
> 1. **Categorization of Evaluator Errors**: Recognizing that different evaluators might err in common and unique ways, we established a taxonomy of potential errors. This categorization helps in understanding the nature of mistakes and provides insights into areas of improvement for each evaluator.
> 2. **Empirical Analysis of Errors**: Based on our error taxonomy, we conducted a hands-on analysis of the actual errors produced by each evaluator on our dataset. This empirical study allowed us to validate our theoretical findings and provided a finegrained understanding of where each evaluator might not fit in real-world scenarios.
> 3. **Detailed Analysis of Evaluators**: We delved into the characteristics and  limitations of various evaluators, including Lexical Matching, Neural Evaluation (BERT-Score/BLEURT), and LLM-as-evaluator (GPT-3.5)
>
> Here are the details for each part:
> ## 1. Error Categories
>
> In addition to characteristic, the strength and limitations of each type of evaluator, we have designed a set of Evaluator Error categories. This includes two common errors found across all evaluators as well as specific errors unique to each type of evaluator.
>
> ### General Error Categories for All Evaluators
>
> | Error Type             | Definition                                                   | Example                                                      |
> | ---------------------- | ------------------------------------------------------------ | ------------------------------------------------------------ |
> | **Paraphrasing Error** | The evaluator fails to recognize answers that paraphrase the golden answer correctly but do not contain the exact substring. | Question: "What is the process by which plants convert sunlight into energy?"<br> Golden Answer: "Photosynthesis"<br> Generated Answer: "The mechanism plants use to transform light into energy is termed the photosynthetic process."<br> Explanation: the generated answer is a paraphrase of the "Photosynthesis" but does not contain the word directly. |
> | **Synonym Error**      | The evaluator fails to recognize answers that use synonyms or alternative phrasing to convey the same meaning as the golden answer. | Question: "What's another term for a doctor?"<br> Golden Answer: "Physician"<br> Generated Answer: "A medical practitioner."<br> Explanation: "medical practitioner" is a synonym for "physician" but isn't a direct substring. |
>
> ### Specific Error Categories
>
> | Judger            | Error Type                            | Description                                                  | Example                                                      |
> | ----------------- | ------------------------------------- | ------------------------------------------------------------ | ------------------------------------------------------------ |
> | **Lexical Matching** | **Partial Match Error **              | The evaluator fails to recognize answers that contain a part of the golden answer but not the entire substring. | Question: "Who painted the Mona Lisa?"<br> Golden Answer: "Leonardo da Vinci"<br> Generated Answer: "The Mona Lisa was painted by Leonardo."<br> Explanation: only "Leonardo" is mentioned, not the full "Leonardo da Vinci". |
> |                   | **Structure Variation Error**         | The evaluator fails to recognize answers that essentially convey the same information as the golden answer but there's a variation in how it's structured. | Question: "When did 'Amnesia: The Dark Descent' come out?"<br> Golden Answer: "8 September 2010"<br> Generated Answer: "Amnesia: The Dark Descent was released on September 8, 2010."<br> Explanation: the date format in the generated answer has an extra comma than the golden answer, even though the information is the same. |
> |                   | **Overall Misleading Error for**      | The evaluator mistakenly recognizes the answer as correct because it contains a substring from the golden answer, even if the overall context of the answer is misleading. | Question: "Who wrote 'The Great Gatsby'?"<br> Golden Answer: "F. Scott Fitzgerald"<br> Generated Answer: "Ernest Hemingway and F. Scott Fitzgerald were close friends, but Hemingway wrote 'The Old Man and the Sea'."<br> Explanation: The generated answer contains the substring "F. Scott Fitzgerald", which might lead the Lexical Match Evaluator to judge it as correct. However, the overall context of the answer is misleading, suggesting a relationship between Hemingway and "The Great Gatsby", which is incorrect. |

---

> > ### Author Response · Authors · 2023-08-20
> > **Rebuttal Part 2 (Q1 Part 2)**
> >
> > | Judger            | Error Type                            | Description                                                  | Example                                                      |
> > | ----------------- | ------------------------------------- | ------------------------------------------------------------ | ------------------------------------------------------------ |
> > | **Neural Evlauation** | **Contextual Misunderstanding Error** | The evaluator might misjudge answers based on word embeddings and fail to capture the context in which certain words or phrases are used. | Question: "Who wrote 'Romeo and Juliet'?"<br> Golden Answer: "William Shakespeare"<br> AI-generated Answer: "Shakespeare wrote many plays."<br> Explanation: Even though the AI answer mentions Shakespeare, it doesn't directly answer the question. |
> > |                   | **Threshold Sensitivity**             | Answers that are just below the threshold might be correct but are judged as incorrect, and vice versa. | Question: "What's the capital of France?"<br> Golden Answer: "Paris"<br> AI-generated Answer: "The capital city of France is Paris."<br> Explanation: The AI answer is correct but might score just below the threshold due to added context. |
> > |                   | **Extended Answer Error**             | The evaluator might penalize answers that provide more context or details than the golden answer, even if they are correct, because the BERT-score only considers the similarities of the candidates and references. | Question: "Who painted the Mona Lisa?"<br> Golden Answer: "Leonardo da Vinci"<br> AI-generated Answer: "Leonardo da Vinci, a renowned Italian artist, painted the Mona Lisa."<br> Explanation: The AI answer provides more context but is still correct. |
> > | **LLM-evaluator** | **Literal Interpretation Error**      | The evaluator might take the question or golden answer too literally and fail to recognize correct answers that provide a broader context or interpretation. | Question: "Which bird is known for its beautiful tail?"<br> Golden Answer: "Peacock"<br> Generated Answer: "Many birds have beautiful tails."<br> Explanation: The evaluator might take a literal approach and accept the general statement as correct without focusing on the specific bird in question. |
> > |                   | **Overgeneralization Error**          | The evaluator might generalize based on its training data and judge an answer as correct even if it's not specific to the question. | Question: "Who wrote 'Pride and Prejudice'?"<br> Golden Answer: "Jane Austen"<br> Generated Answer: "An English author."<br> Explanation: The evaluator might accept the general answer as it's not technically wrong, even though it lacks specificity. |
> > |                   | **MIsleading Emphasis Error**         | The evaluator might judge an answer as correct if it includes some correct information and put emphasis on it, and overlook the incorrect primary claim. | Question: "What's the primary gas in Earth's atmosphere?"<br> Golden Answer: "Nitrogen"<br> Generated Answer: "Oxygen, which makes up about 78% of the atmosphere."<br> Explanation: GPT-3.5 might focus on the correct percentage and overlook incorrect mention of "Oxygen" as a primary gas. |
> > |                   | **Unknowable Reasons**                | The evaluator makes an incorrect judgment for an unknowable reason. Even humans cannot figure out why the LLM thinks the generated answer is correct since it has no correlation with the golden answer. | Question: "Who was the first chief minister of West Bengal?"<br> Golden Answer: "Prafulla Chandra Ghosh"<br> Generated Answer: "The first Chief Minister of West Bengal was Dr. Bidhan Chandra Roy."<br> Explanation: GPT-3.5 takes the generated answer as correct, but Dr. Bidhan Chandra Roy is apparently not Prafulla Chandra Ghosh. |
> >
> > While these categories might overlap in certain instances, each possesses its own distinct emphasis. When classifying, we choose the category that best fits each specific case.

---

> > > ### Author Response · Authors · 2023-08-20
> > > **Rebuttal Part 3 (Q1 Part 3)**
> > >
> > > ## 2. Error Analysis
> > > Based on the aforementioned Error Categories, we manually classified the errors produced by Lexical Matching, BERT-Score, and GPT-3.5 on each subset of NQ. For each subset, we selected 100 errors. If there were fewer than 100 errors, we included all of them. The classification results are as follows:
> > > ### The error results for Lexical Match evaluator, BERT-Score evaluator and GPT-3.5 evaluator.
> > > Each kind evaluator has common error types and specific error types. General error rate indicates the error proportion of this evaluator on this subset.
> > >
> > > | Error Type/Model | NQ-FiD | NQ-GPT35 | NQ-ChatGPT35 | NQ-ChatGPT4 | NQ-BingChat |
> > > |------------------|--------|----------|--------------|-------------|-------------|
> > > | **Lexical Matching** | | | | | |
> > > | Paraphrasing Error | 29% | 37% | 29% | 60% | 49% |
> > > | Synonym Error | 18% | 12% | 37% | 12% | 19% |
> > > | Partial Match Error | 48% | 30% | 13% | 10% | 20% |
> > > | Structure Variation Error | 4% | 16% | 15% | 12% | 7% |
> > > | Overall Misleading Error | 1% | 5% | 6% | 6% | 5% |
> > > | General error rate | 11.75 | 15.2 | 19.7 | 16.8 | 17.7 |
> > > | **BERT-Score** | | | | | |
> > > | Paraphrasing Error | 4% | 24% | 29% | 39% | 39% |
> > > | Synonym Error | 4% | 7% | 4% | 5% | 5% |
> > > | Contextual Misunderstanding Error | 63% | 22% | 23% | 20% | 15% |
> > > | Threshold Sensitivity Error | 25% | 33% | 20% | 18% | 15% |
> > > | Extended Answer Error | 4% | 14% | 24% | 18% | 26% |
> > > | General error rate | 25.0 | 30.5 | 27.2 | 23.2 | 32.4 |
> > > | **GPT-3.5** | | | | | |
> > > | Paraphrasing Error | 16% | 52% | 36% | 52% | 47% |
> > > | Synonym Error | 22% | 12% | 21% | 18% | 17% |
> > > | Literal Interpretation Error | 21% | 4% | 11% | 6% | 13% |
> > > | Overgeneralization Error | 17% | 13% | 8% | 8% | 6% |
> > > | Misleading Emphasis Error | 7% | 2% | 5% | 3% | 6% |
> > > | Unknowable Reasons Error | 17% | 8% | 19% | 13% | 11% |
> > > | General error rate | 6.4 | 16.0 | 17.8 | 16.6 | 30.5 |

---

> > > > ### Author Response · Authors · 2023-08-20
> > > > **Rebuttal Part 4 (Q1 Part 4)**
> > > >
> > > > ## **Key Insights**
> > > > Drawing from each evaluator's limitations and error classifications results, we offer these insights:
> > > > 1. Lexical Matching:
> > > >    - While lexical matching remains a simple and effective method for Open-QA evaluation, it struggles with issues of limited contextual understanding, low recall, and structural variations.
> > > >    - It often marks answers that humans consider correct as incorrect, but rarely does the opposite. This makes Lexical Matching a strict metric, suitable for environments requiring high error recall.
> > > >    - When it does mark a human-considered correct answer as wrong, it's usually because the generated answer contains the golden answer, but the overall meaning doesn't support it. For instance, it might negate the golden answer or only use it as part of the response.
> > > >    - Lexical Matching struggles with "Structure Variation" errors. For example, if the golden answer is "8 September 2010" and the generated answer is "It was released on September 8, 2010.", Lexical Matching can't recognize it. The other two evaluators rarely have this issue.
> > > >    - Due to its inability to handle semantics, it can't manage Synonym or Contextual understanding situations.
> > > > 2. Neural Evaluation (BERT-Score and BLEURT in this work):
> > > >    - Overall, they aren't well-suited for this QA-Eval task, with the poorest performance among the three types. They can only measure the similarity between two text segments. So, they handle Synonym errors well. However, if the generated answer contains extra information (common with larger models), this can easily influence the BERT-score.
> > > >    - BERT-Score isn't great at Contextual Understanding. If the generated answer explains the golden answer without including its entities, BERT-Score can easily get it wrong.
> > > >    - Adapting BERT-Score to this QA-Eval task by adjusting the threshold is another issue. Many datasets are highly sensitive to threshold settings.
> > > > 3. LLM-evaluator (GPT-3.5 in this work):
> > > >    - Overall, the LLM-evaluator can serve as a complement to lexical matching and is valuable for assessing the accuracy of generated answers, it remains sensitive to prompts and the impact of additional contexts, especially for bingchat answers.
> > > >    - Its most common error is the "Paraphrasing error", possibly because it's easily influenced by other contexts.
> > > >    - It has its own issues, like the "Overgeneralization error", which doesn't appear in the other two evaluators, although they are minor concernss.
> > > >    - Sometimes, the LLM-evaluator makes clear mistakes that humans wouldn't. For example, for the question, "Who was the first chief minister of West Bengal?" with the golden answer being "Prafulla Chandra Ghosh", the generated answer was "The first Chief Minister of West Bengal was Dr. Bidhan Chandra Roy." GPT-3.5 marked the generated answer as correct, even though Dr. Bidhan Chandra Roy is not Prafulla Chandra Ghosh. This might be because the evaluator uses its inherent knowledge, overlooking the golden answer, or for other undetermined reasons. Such issues don't appear with the other two evaluators.
> > > >
> > > > In general, while lexical matching is a straightforward and effective method for Open-QA evaluation, it encounters challenges related to limited contextual understanding, low recall, and structural variations.
> > > > The LLM-evaluator can be effectively complementary to lexical matching, and is valuable in assessing the accuracy of generated answers, but it is not robust and particularly sensitive to prompts and the influence of additional contexts, especially in the case of BingChat answers.
> > > > The application of neural evaluations in the QA-Eval task requires further exploration.

---

> > > > > ### Author Response · Authors · 2023-08-20
> > > > > **Rebuttal Part 5 (Q1 Part 5)**
> > > > >
> > > > > ##  3. Characteristics and Limitations of Each Evaluator
> > > > > Based on our theoretical analysis and observations of erroneous cases, we identified the following issues with each type of evaluator:
> > > > >
> > > > > 3.1. Lexical Matching:
> > > > >
> > > > >    - Lack of Semantic Understanding: The exact match metric doesn't take into account the semantic meaning of the answers. It only checks if the predicted answer is exactly the same as the ground truth, even if the predicted answer is semantically correct but phrased differently.
> > > > >    - Inability to Handle Synonyms: The exact match metric cannot handle synonyms. If the predicted answer uses a different word that has the same meaning as the word in the ground truth answer, the exact match metric will consider it as a wrong answer.
> > > > >    - Inability to Handle Paraphrasing: Similar to the point above, the exact match metric cannot handle paraphrasing. If the predicted answer is a paraphrase of the ground truth answer, the exact match metric will consider it as a wrong answer.
> > > > >    - Inability to Handle Partially Correct Answers: The exact match metric cannot handle partially correct answers. If the predicted answer is partially correct, the exact match metric will consider it as a wrong answer.
> > > > >    - Inability to Handle Reordered Words: The exact match metric cannot handle reordered words. If the predicted answer has the same words as the ground truth answer but in a different order, the exact match metric will consider it as a wrong answer.
> > > > >    - Inability to Handle Different Formats: The exact match metric cannot handle different formats. If the predicted answer is in a different format than the ground truth answer (for example, dates or numbers), the exact match metric will consider it as a wrong answer.
> > > > > These limitations highlight the need for more sophisticated evaluation metrics that can understand the semantic meaning of the answers and handle synonyms, paraphrasing, partially correct answers, reordered words, different levels of detail, and different formats.
> > > > >
> > > > >
> > > > > 3.2 Neural Evaluation:
> > > > >
> > > > >    - The limitations of neural evaluation methods, such as BERT-Score and BLEURT, are evident. Most crucially, many neural evaluations are primarily designed to measure the similarity between two phrases or sentences. They are not tailored for binary tasks, especially those assessing the factual correctness of answers. Instead, they provide a continuous score that gauges the similarity between the generated text and the reference text, rendering them directly unsuitable for this particular task. In our study, we employed BERT-score and BLEURT for this task by setting a threshold. However, the performance of both BERT-score and BLEURT was suboptimal. The primary shortcoming of neural evaluations for this task is their misalignment with its requirements.
> > > > >    - Furthermore, BERT-score has the following limitations:：
> > > > >    - Sensitivity to Verbosity: BERT-score may penalize verbose answers even if they contain the correct information. If the AI-generated answer provides a detailed explanation while the golden answer is concise, the score might be lower than expected.
> > > > >    - Mismatched Focus: If the AI-generated answer is correct but emphasizes different aspects or details than the golden answer, BERT-score might not recognize the similarity, leading to a lower score.
> > > > >    - Lack of Contextual Understanding: BERT-score measures the similarity between embeddings but might not fully capture the contextual nuances of certain answers, especially when there are multiple valid ways to answer a question.
> > > > >    - Synonym and Paraphrasing Issues: BERT-score might not always recognize synonyms or paraphrased answers as being equivalent to the golden answer, leading to potential discrepancies in scoring.
> > > > >    - Threshold Limitations: Setting a fixed threshold (e.g., 0.5) for determining correctness can be arbitrary. Some answers might be just below the threshold but still be correct, while others might be just above but incorrect.
> > > > >    - Doesn't Account for Minor Details: BERT-score might not be sensitive enough to minor inaccuracies in the AI-generated answer, especially if the overall semantic content is similar to the golden answer.
> > > > >    - Lack of Absolute Truth Measure: BERT-score is a relative measure of similarity between two pieces of text. It doesn't provide an absolute measure of the truthfulness or correctness of an answer.
> > > > >    - Influence of Sentence Structure: The structure or order of sentences in the AI-generated answer compared to the golden answer might affect the score, even if the content is the same.
> > > > > In summary, while BERT-score is a powerful metric for evaluating text similarity, its application in a QA-eval task has severe limitations.

---

> > > > > > ### Author Response · Authors · 2023-08-20
> > > > > > **Rebuttal Part 6 (Q1 Part 6)**
> > > > > >
> > > > > > 3.3 LLM-Evaluator (GPT-3.5 in this work) :
> > > > > >
> > > > > >    - Sensitivity to Prompt Engineering: Both closed-source and open-source LLMs can be sensitive to the way questions are framed or prompts are constructed. This can introduce variability in the evaluation, where slight rephrasings might lead to different judgments.
> > > > > >    - Literal Interpretation: One of the limitations is the model's tendency to interpret questions or golden answers too literally. This can lead to situations where the evaluator fails to recognize correct answers that provide a broader context or a different interpretation that still addresses the core of the question.
> > > > > >    - Overgeneralization: Another challenge is the model's propensity to overgeneralize based on its vast training data. This can result in the evaluator deeming an answer as correct even if it doesn't align specifically with the nuances of the question at hand.
> > > > > >    - Misleading Emphasis: The evaluator might sometimes be swayed by partial correctness in an answer. If an answer emphasizes certain correct elements, the evaluator might overlook primary claims that are factually incorrect, leading to a misleading evaluation.
> > > > > >    - Unknowable Reasoning: There are instances where the evaluator's judgment is puzzling, even to human experts. The model might deem an answer as correct that has no discernible correlation with the golden answer. This limitation underscores the "black-box" nature of deep learning models, where their internal reasoning processes remain opaque.
> > > > > >    - Potential Bias: All LLMs, whether closed or open source, can inherit biases from their training data. In the context of QA-Eval, this might manifest as favoring certain types of answers or being biased against certain topics or contexts
> > > > > >
> > > > > > **(We have included those analysis, including Error Categories, Error Analysis and Characteristics and Limitations of Each Evaluator into the Section 6.1 **Error Analysis in QA-Eval** and Appendix Section E4 **Error Analysis in QA-Eval** in the updated paper.)**

---

> > > > > > > ### Comment · Reviewer_Hse5 · 2023-08-30
> > > > > > >
> > > > > > > Appreciate the error analysis, really meticulous in categorizing and evaluating. A,m updating the score relevantly.

---

> ### Author Response · Authors · 2023-08-20
> **Rebuttal Part 7 (Q2)**
>
> ## Limitations
>
> - Q2: Large language models are very sensitive to prompts when used for evaluation. The models are sensitive to the order of the options presented along with the question prompt. The large language models setup is not sufficiently well-studied to conclude the results.
>
>
>     R2: Thank you for your insightful and constructive feedback. We agree with your observation regarding the sensitivity of Large Language Models (LLMs) to prompts during evaluation, which also observed by us in the experiments.
>
>     We have added relevant dicsussions to the limitations of LLM-as-Evaluator on the rebuttal (the Error Analaysis and Limitations of Each Evaluator part of R1) and the Section 6.1 Error Analysis in QA-Eval in the paper, which we copy below:
>
>         (Findings of LLM-Evaluator) Overall, while the LLM-evaluator is valuable for assessing the accuracy of generated answers, it remains sensitive to prompts and the impact of additional contexts, especially for bingchat answers. It can serve as a complement to lexical matching.
>     and
>
>         (Limitations of LLM-Evaluator) Sensitivity to Prompt Engineering: Both closed-source and open-source LLMs can be sensitive to the way questions are framed or prompts are constructed. This can introduce variability in the evaluation, where slight rephrasings might lead to different judgments.
>
>     Thanks for the insightful comment, which added to the informativeness of our discussion on the relative strenghts and weaknesses of each evaluation method.

---

> ### Author Response · Authors · 2023-08-20
> **Rebuttal Part 8 (Q3)**
>
> - Q3: The language models are subject to biases and using them directly for evaluation can lead to unwanted biases in the system that can impact further development of models or datasets.
>
> R3: Thanks for raising the concern of bias. We acknowledge the concern regarding biases in language models. However, it's essential to note that our study primarily focuses on Open QA datasets like NQ and TQ, which emphasize factual QA. In these datasets, the issue of bias is less significant as they don't involve subjective content. Discussing biases in the context of these datasets might not be entirely appropriate. Our research is centered on Factual Open-QA, which is the predominant focus in the Open-QA community. Open-QA that delves into subjective content is not extensively discussed in our research domain due to the myriad of ethical considerations it entails.
>
> We have claimed the scope of our Open-QA task in the introduction of the revised paper.
>
> We also write the potential bias into the limitations of LLM-Evaluator (Rebuttal Part 6, LLM-Evaluator part in Characteristics and Limitations of Each Evaluator), since this may have an impact on other types of QA evaluation.

---

> ### Author Response · Authors · 2023-08-20
> **Rebuttal Part 9 (Q4)**
>
> # Correctness:
> - Q4: It seems like the lexical match did not align well with the human evaluation accuracies, however, it is concluded that lexical match is a good metric. The inefficiency of direct lexical match is also accepted in the field and the findings do not compare with prior claims on this.
>
> R4: Thank you for pointing out this observation. We'd like to clarify this issue.
>
> 1. **Relative Ranks of QA Models**: While the plain lexical match score might seem to perform well at first glance, it's essential to consider its reliability in evaluating the performance of QA models. Our analysis revealed that the lexical match score produces different relative ranks for the five QA models when compared to human evaluations. This discrepancy underscores its unreliability as an evaluator for QA model performance. The differences in rankings are illustrated in the subsequent table:
>
>     ### Evaluation scores assigned by various evaluation models to different QA models on EVOUNA
>     In each cell, the left represents the score given by the evaluator (row) to the QA model's performance (column) on the respective dataset, while the right is the relative ranks among the five models.
>
>     | Model/Method | NQ-FiD | NQ-GPT35 | NQ-ChatGPT35 | NQ-GPT4 | NQ-BingChat |
>     |--------------|--------|----------|--------------|---------|-------------|
>     | Lexical Matching | 56.1 (4) | 50.7 (5) | 57.9 (3) | 65.8 (1) | 65.4 (2) |
>     | BERT-Score | 82.5 (1) | 70.9 (4) | 71.9 (3) | 74.7 (2) | 64.8 (5) |
>     | GPT-3.5 | 67.5 (2) | 58.6 (4) | 63.2 (3) | 70.3 (1) | 54.1 (5) |
>     | Human | 69.7 (4) | 70.3 (3) | 63.0 (5) | 80.3 (1) | 78.2 (2) |
>
>     #### on EVOUNA-NaturalQuestions
>
>     | Model/Method | TQ-FiD | TQ-GPT35 | TQ-ChatGPT35 | TQ-GPT4 | TQ-BingChat |
>     |--------------|--------|----------|--------------|---------|-------------|
>     | Lexical Matching | 73.5 (4) | 71.0 (5) | 76.7 (3) | 82.1 (1) | 81.6 (2) |
>     | BERT-Score | 56.7 (5) | 73.9 (4) | 82.0 (2) | 84.4 (2) | 78.0 (3) |
>     | GPT-3.5 | 71.7 (4) | 66.0 (5) | 72.4 (3) | 82.3 (1) | 73.4 (2) |
>     | Human | 79.6 (3) | 72.2 (5) | 80.9 (2) | 86.0 (1) | 73.1 (4) |
>
>     #### on EVOUNA-TriviaQA
>
> 2. **Recall Scores**: The lexical match method consistently exhibits the lowest recall scores among the four methods we analyzed. This indicates that it frequently misjudges correct samples as incorrect. A detailed breakdown of these recall scores can be found in Table X in the appendix. The relatively low recall scores of the lexical match method emphasize its tendency to be overly stringent, often failing to recognize valid answers that may not match the reference lexically but are semantically correct.
>
>
> 3. **Significance of the 10% Gap**: It's crucial to contextualize the 10% gap in light of human annotator variability. Given that discrepancies between human annotators are typically in the single-digit percentage range, a 10% gap is substantial. Our analysis of the error rates produced by each evaluator, compared to human annotators, revealed that the errors introduced by the three evaluators are several times those made by humans.
>
>     ### Error results of Eval-Models on the EVOUNA
>     In each cell, the left is the error rates while the right is the times compared with another human results.
>
>     |                   | NQ-FiD        | NQ-GPT35      | NQ-ChatGPT35  | NQ-ChatGPT4   | NQ-BingChat   |
>     |-------------------|---------------|---------------|---------------|---------------|---------------|
>     | Lexical Matching     | 13.1/3.5x     | 15.2/4.8x     | 19.7/4.5x     | 16.8/4.9x     | 17.7/3.9x     |
>     | BERT-Score        | 25.0/6.8x     | 30.5/9.5x     | 27.2/6.2x     | 23.2/6.8x     | 32.4/7.2x     |
>     | GPT-3.5           | 6.4/1.7x      | 16.0/5.0x     | 17.8/4.0x     | 16.6/4.9x     | 30.5/6.8x     |
>     | Another Human     | **3.7/1.0x**  | **3.2/1.0x**  | **4.4/1.0x**  | **3.4/1.0x**  | **4.5/1.0x**  |
>
>     #### on EVOUNA-NaturalQuestions
>
>     |                   | TQ-FiD        | TQ-GPT35      | TQ-ChatGPT35  | TQ-ChatGPT4   | TQ-BingChat   |
>     |-------------------|---------------|---------------|---------------|---------------|---------------|
>     | Lexical Matching     | 10.0/33.3x    | 7.7/12.8x     | 7.7/6.4x      | 8.9/44.5x     | 10.2/51.0x    |
>     | BERT-Score        | 34.6/115.3x   | 24.3/40.5x    | 19.3/16.1x    | 16.6/83.0x    | 19.6/98.0x    |
>     | GPT-3.5           | 4.3/14.3x     | 8.8/14.7x     | 7.3/6.1x      | 7.5/37.5x     | 18.8/94.0x    |
>     | Another Human     | **0.3/1.0x**  | **0.6/1.0x**  | **1.2/1.0x**  | **0.2/1.0x**  | **0.2/1.0x**  |
>
>     #### on EVOUNA-TriviaQA

---

> > ### Comment · Reviewer_Hse5 · 2023-08-30
> >
> > It would be great to clarify this in the paper as it is mentioned in the draft that lexical match is a good proxy. However, from your analysis, it seems like its not reliable and is inconsistent. Appreciate the additional analysis, please clarify in the paper and I am updating the score based on this.

---

> > > ### Author Response · Authors · 2023-08-31
> > >
> > > Thanks for you suggestions, we have revised the wording to avoid confusion.

---

> ### Author Response · Authors · 2023-08-20
> **Rebuttal Part 10 (Q5)**
>
> # Clarity
> - Q5: Questions whose answers are temporally dependent or evolve over time have been removed from the datasets for evaluation. However, it is not clearly mentioned how these questions are identified for removal (the statistics post removal are mentioned in the paper).
>
>
>     R5: Thank you for raising this concern. We recognize the importance of transparency in our methodology. To address this, we employed specific criteria to determine the removal of QA pairs that might be temporally dependent or contain errors. Here's how we identified such questions:
>
>   1. whether the question have answers that change over time:
>
>      - If the question is clearly time-sensitive, then it is Yes.
>
>      - If there are words closely related to the current time node such as “this year”, “last year”, “next time” and “last time” in this question, then it is Yes.
>
>      - If the question contains values that change over decades, such as “who is the player with the most goals in the World Cup so far”, then it is Yes.
>
>      - If the question contains values that do not change in decades, such as “what is the tallest mountain in the world”, then it is No.
>
>   2. Whether the Golden Answer Contains Severe Errors
>
>      -   If the golden answer has structure errors, then it is Yes.
>
>          Example: Question: the south west wind blows across nigeria between?
>          Golden: till September
>
>      -   If the golden answer is obviously not what is asked, then it is Yes.
>
>      -   If the golden answer has format errors, then it is Yes.
>
>          Example: Question: what season does bart bass die in gossip girl?
>          Golden: $($
>
>      -   If the golden answer has only factual errors, then it is No.
>
>     Since the scope and form of QA pairs in the NQ and TQ are relatively limited, these rules are enough to cover most cases.
>
>     We've consolidated these separated guidelines into a comprehensive Annotation Guidelines section in the Appendix C.2 Section of the updated paper. The separation of the guidelins in the original paper was to ensure a coherent narrative flow in the main paper and to minimize redundancy.
>     We believe that new comprehensive guidelines provides a clear and systematic method for dataset curation.

---

> > ### Comment · Reviewer_Hse5 · 2023-08-30
> >
> > Thanks for adding a new comprehensive set of guidelines. It helps understanding the annotation better.

---

> ### Author Response · Authors · 2023-08-20
> **Rebuttal Part 11 (Q6)**
>
> # Relation To Prior Work:
> Q6: Kadavath, Saurav, et al. "Language models (mostly) know what they know." arXiv preprint arXiv:2207.05221 (2022).
>
> R6: Thank you for pointing out the relevance of the paper "Language Models (Mostly) Know What They Know" in the context of our study. Here's a comparison of our paper with the aforementioned paper:
>
> Focus and Objective: While both papers delve into the evaluation of language models, our study specifically targets the OpenQA task, introducing the QA-Eval task and the EVOUNA dataset. On the other hand, the paper "Language Models (Mostly) Know What They Know" investigates the ability of models to self-evaluate their claims and introduces the concept of calibration.
>
> Methodology: Our paper evaluates the OpenQA task using various methods, including lexical match, neural-evaluation, and large language models. The other paper, however, focuses on the calibration of models and their ability to predict the correctness of their answers.
>
> Findings: Our findings emphasize the importance of human evaluations in assessing the factuality of LLMs and highlight the limitations of current automatic evaluation methods. The "Language Models (Mostly) Know What They Know" paper demonstrates that large models can be well-calibrated on diverse multiple-choice questions and introduces techniques for models to evaluate their own outputs.
>
> In conclusion, while there are thematic overlaps between the two papers, the objectives, methodologies, and findings differ, offering unique contributions to the understanding and evaluation of language models.

---

> > ### Comment · Reviewer_Hse5 · 2023-08-30
> >
> > This is a good contrast between the mentioned paper and current work. It would be great to mention it in the work, am okay if not too. Appreciate your effort in drawing similarities and differences.

---

> ### Author Response · Authors · 2023-08-20
> **Rebuttal Part 12 (typos)**
>
> # Minor grammatical / spelling corrections
> R7: Thank you very much for your meticulous review. We have addressed and corrected all the typos and grammatical issues you pointed out.

---

### Official Review · Reviewer_8Zik · 2023-07-21

**Rating:** 7
**Confidence:** 4
**Correctness:** The claims are correct and sound.
**Clarity:** Writing is clear and easy to follow

**Strengths:**

The paper studies an important problem in open QA evaluation. The authors systematically examine and compare existing automatic evaluation metrics with human annotations.

The proposed new evaluation task is interesting and contains more reliable human evaluation annotations.

**Additional Feedback:**

Questions:

- Why is only a portion (500 cases instead of a full >10k) of the TQ test set selected? Is it due to limited human annotation resources? If so, a more comprehensive analysis of the full test set is required to draw statistically sound conclusions.

- Table 5 results seem to show plain lexical match score already does a very good evaluation job, what does 10% gap mean compared to human annotators?



Writing:

Typo in like 83, researches -> research

**Documentation:**

The documentation is clear

**Ethics:**

No ethical concerns

**Limitations:**

The human annotation process is somewhat biased since the authors are the annotators. The annotation process may not be scientific. It is also unclear how many authors and how the authors perform the annotation. Only vague guidelines  (four simple rules) are provided in the appendix, it also does not warrant consistent human annotation.

**Opportunities For Improvement:**

The paper only discusses variation in expression for the exact match metric, are there any other issues in the exact match metric? And what are the limitations of other automatic evaluation metrics like neural evaluation (e.g. BERT-Score)?
Newer neural evaluation metrics blurt might somewhat address the issue, some discussion or evaluation is preferred.

The paper only discusses one open QA model on two open QA datasets, the DPR+FiD model, it would be much better to see if the evaluation of other open, transparent open QA models, like ATLAS (https://arxiv.org/pdf/2208.03299.pdf), LLaMa, etc.

The lexical match score for LLM predictions is inconsistent (substring versus exact match) with other models (FiD), it does not support fair evaluation comparison. Will proper prompting solve the issue?

**Relation To Prior Work:**

Related work is clearly discussed

**Summary And Contributions:**

This paper studies the evaluation of open domain question answering (open QA) tasks. It points out exact match score is insufficient for expression variations. The paper also constructs an evaluation dataset called EVOUNA and proposes to use human annotation to compute the correlations between model predictions. The authors investigate three sets of automatic evaluation metrics including exact match, neural scores, and using large language models (LLMs), and reveal the shortcomings compared to human evaluation. It further improves the LLM evaluation metric via in-context learning and chain of thought prompting.

---

> ### Author Response · Authors · 2023-08-20
> **Rebuttal Part 1 (Q1 part1)**
>
> # Opportunities For Improvement:
>   - Q1: The paper only discusses variation in expression for the exact match metric,   are there any other issues in the exact match metric?  And what are the limitations of other automatic evaluation metrics like neural evaluation (e.g. BERT-Score)?  Newer neural evaluation metrics blurt might somewhat address the issue, some discussion or evaluation is preferred.
>
>     R1: Thank you for your insightful comments and questions regarding the scope of our paper.
>     To address these concerns, we have conducted the QA-Eval Error Analysis, which consists four parts:
>        1. **Categorization of Evaluator Errors**: Recognizing that different evaluators might err in common and unique ways, we established a taxonomy of potential errors. This categorization helps in understanding the nature of mistakes and provides insights into areas of improvement for each evaluator.
>        2. **Empirical Analysis of Errors**: Based on our error taxonomy, we conducted a hands-on analysis of the actual errors produced by each evaluator on our dataset. This empirical study allowed us to validate our theoretical findings and provided a finegrained understanding of where each evaluator might not fit in real-world scenarios.
>        3. **Detailed Analysis of Evaluators**: We delved into the characteristics and  limitations of various evaluators, including Lexical Matching, Neural Evaluation (BERT-Score/BLEURT), and LLM-as-evaluator (GPT-3.5).
>        4. **Evaluation of BLUERT**: We also conducted a QA-Eval analysis on the more recent Neural-Evaluation model, BLEURT, on our EVOUNA dataset.
>
> Here are the details for each part:

---

> > ### Author Response · Authors · 2023-08-20
> > **Rebuttal Part 2 (Q1 part2)**
> >
> > ## 1. Error Categories
> >
> >    In addition to characteristic, the strength and limitations of each type of evaluator, we have designed a set of Evaluator Error categories. This includes two common errors found across all evaluators as well as specific errors unique to each type of evaluator.
> > ### General Error Categories for All Evaluators
> >
> >    | Error Type             | Definition                                                   | Example                                                      |
> >    | ---------------------- | ------------------------------------------------------------ | ------------------------------------------------------------ |
> >    | **Paraphrasing Error** | The evaluator fails to recognize answers that paraphrase the golden answer correctly but do not contain the exact substring. | Question: "What is the process by which plants convert sunlight into energy?"<br> Golden Answer: "Photosynthesis"<br> Generated Answer: "The mechanism plants use to transform light into energy is termed the photosynthetic process."<br> Explanation: the generated answer is a paraphrase of the "Photosynthesis" but does not contain the word directly. |
> >    | **Synonym Error**      | The evaluator fails to recognize answers that use synonyms or alternative phrasing to convey the same meaning as the golden answer. | Question: "What's another term for a doctor?"<br> Golden Answer: "Physician"<br> Generated Answer: "A medical practitioner."<br> Explanation: "medical practitioner" is a synonym for "physician" but isn't a direct substring. |

---

> > > ### Author Response · Authors · 2023-08-20
> > > **Rebuttal Part 3 (Q1 part3)**
> > >
> > > ### Specific Error Categories
> > >
> > >    | Judger            | Error Type                            | Description                                                  | Example                                                      |
> > >    | ----------------- | ------------------------------------- | ------------------------------------------------------------ | ------------------------------------------------------------ |
> > >    | **Lexical Matching** | **Partial Match Error **              | The evaluator fails to recognize answers that contain a part of the golden answer but not the entire substring. | Question: "Who painted the Mona Lisa?"<br> Golden Answer: "Leonardo da Vinci"<br> Generated Answer: "The Mona Lisa was painted by Leonardo."<br> Explanation: only "Leonardo" is mentioned, not the full "Leonardo da Vinci". |
> > >    |                   | **Structure Variation Error**         | The evaluator fails to recognize answers that essentially convey the same information as the golden answer but there's a variation in how it's structured. | Question: "When did 'Amnesia: The Dark Descent' come out?"<br> Golden Answer: "8 September 2010"<br> Generated Answer: "Amnesia: The Dark Descent was released on September 8, 2010."<br> Explanation: the date format in the generated answer has an extra comma than the golden answer, even though the information is the same. |
> > >    |                   | **Overall Misleading Error for**      | The evaluator mistakenly recognizes the answer as correct because it contains a substring from the golden answer, even if the overall context of the answer is misleading. | Question: "Who wrote 'The Great Gatsby'?"<br> Golden Answer: "F. Scott Fitzgerald"<br> Generated Answer: "Ernest Hemingway and F. Scott Fitzgerald were close friends, but Hemingway wrote 'The Old Man and the Sea'."<br> Explanation: The generated answer contains the substring "F. Scott Fitzgerald", which might lead the Lexical Match Evaluator to judge it as correct. However, the overall context of the answer is misleading, suggesting a relationship between Hemingway and "The Great Gatsby", which is incorrect. |
> > >    | **Neural Evlauation** | **Contextual Misunderstanding Error** | The evaluator might misjudge answers based on word embeddings and fail to capture the context in which certain words or phrases are used. | Question: "Who wrote 'Romeo and Juliet'?"<br> Golden Answer: "William Shakespeare"<br> AI-generated Answer: "Shakespeare wrote many plays."<br> Explanation: Even though the AI answer mentions Shakespeare, it doesn't directly answer the question. |
> > >    |                   | **Threshold Sensitivity**             | Answers that are just below the threshold might be correct but are judged as incorrect, and vice versa. | Question: "What's the capital of France?"<br> Golden Answer: "Paris"<br> AI-generated Answer: "The capital city of France is Paris."<br> Explanation: The AI answer is correct but might score just below the threshold due to added context. |
> > >    |                   | **Extended Answer Error**             | The evaluator might penalize answers that provide more context or details than the golden answer, even if they are correct, because the BERT-score only considers the similarities of the candidates and references. | Question: "Who painted the Mona Lisa?"<br> Golden Answer: "Leonardo da Vinci"<br> AI-generated Answer: "Leonardo da Vinci, a renowned Italian artist, painted the Mona Lisa."<br> Explanation: The AI answer provides more context but is still correct. |

---

> > > > ### Author Response · Authors · 2023-08-20
> > > > **Rebuttal Part 4 (Q1 part4)**
> > > >
> > > > | Judger            | Error Type                            | Description                                                  | Example                                                      |
> > > >    | ----------------- | ------------------------------------- | ------------------------------------------------------------ | ------------------------------------------------------------ |
> > > >    | **LLM-evaluator** | **Literal Interpretation Error**      | The evaluator might take the question or golden answer too literally and fail to recognize correct answers that provide a broader context or interpretation. | Question: "Which bird is known for its beautiful tail?"<br> Golden Answer: "Peacock"<br> Generated Answer: "Many birds have beautiful tails."<br> Explanation: The evaluator might take a literal approach and accept the general statement as correct without focusing on the specific bird in question. |
> > > >    |                   | **Overgeneralization Error**          | The evaluator might generalize based on its training data and judge an answer as correct even if it's not specific to the question. | Question: "Who wrote 'Pride and Prejudice'?"<br> Golden Answer: "Jane Austen"<br> Generated Answer: "An English author."<br> Explanation: The evaluator might accept the general answer as it's not technically wrong, even though it lacks specificity. |
> > > >    |                   | **MIsleading Emphasis Error**         | The evaluator might judge an answer as correct if it includes some correct information and put emphasis on it, and overlook the incorrect primary claim. | Question: "What's the primary gas in Earth's atmosphere?"<br> Golden Answer: "Nitrogen"<br> Generated Answer: "Oxygen, which makes up about 78% of the atmosphere."<br> Explanation: GPT-3.5 might focus on the correct percentage and overlook incorrect mention of "Oxygen" as a primary gas. |
> > > >    |                   | **Unknowable Reasons**                | The evaluator makes an incorrect judgment for an unknowable reason. Even humans cannot figure out why the LLM thinks the generated answer is correct since it has no correlation with the golden answer. | Question: "Who was the first chief minister of West Bengal?"<br> Golden Answer: "Prafulla Chandra Ghosh"<br> Generated Answer: "The first Chief Minister of West Bengal was Dr. Bidhan Chandra Roy."<br> Explanation: GPT-3.5 takes the generated answer as correct, but Dr. Bidhan Chandra Roy is apparently not Prafulla Chandra Ghosh. |
> > > >
> > > >    While these categories might overlap in certain instances, each possesses its own distinct emphasis. When classifying, we choose the category that best fits each specific case.

---

> > > > > ### Author Response · Authors · 2023-08-20
> > > > > **Rebuttal Part 5 (Q1 part5)**
> > > > >
> > > > > ## 2. Error Analysis
> > > > >    Based on the aforementioned Error Categories, we manually classified the errors produced by Lexical Matching, BERT-Score, and GPT-3.5 on each subset of NQ. For each subset, we selected 100 errors. If there were fewer than 100 errors, we included all of them. The classification results are as follows:
> > > > >    ### The error results for Lexical Match evaluator, BERT-Score evaluator and GPT-3.5 evaluator.
> > > > >    Each kind evaluator has common error types and specific error types. General error rate indicates the error proportion of this evaluator on this subset.
> > > > >
> > > > >    | Error Type/Model | NQ-FiD | NQ-GPT35 | NQ-ChatGPT35 | NQ-ChatGPT4 | NQ-BingChat |
> > > > >    |------------------|--------|----------|--------------|-------------|-------------|
> > > > >    | **Lexical Matching** | | | | | |
> > > > >    | Paraphrasing Error | 29% | 37% | 29% | 60% | 49% |
> > > > >    | Synonym Error | 18% | 12% | 37% | 12% | 19% |
> > > > >    | Partial Match Error | 48% | 30% | 13% | 10% | 20% |
> > > > >    | Structure Variation Error | 4% | 16% | 15% | 12% | 7% |
> > > > >    | Overall Misleading Error | 1% | 5% | 6% | 6% | 5% |
> > > > >    | General error rate | 11.75 | 15.2 | 19.7 | 16.8 | 17.7 |
> > > > >    | **BERT-Score** | | | | | |
> > > > >    | Paraphrasing Error | 4% | 24% | 29% | 39% | 39% |
> > > > >    | Synonym Error | 4% | 7% | 4% | 5% | 5% |
> > > > >    | Contextual Misunderstanding Error | 63% | 22% | 23% | 20% | 15% |
> > > > >    | Threshold Sensitivity Error | 25% | 33% | 20% | 18% | 15% |
> > > > >    | Extended Answer Error | 4% | 14% | 24% | 18% | 26% |
> > > > >    | General error rate | 25.0 | 30.5 | 27.2 | 23.2 | 32.4 |
> > > > >    | **GPT-3.5** | | | | | |
> > > > >    | Paraphrasing Error | 16% | 52% | 36% | 52% | 47% |
> > > > >    | Synonym Error | 22% | 12% | 21% | 18% | 17% |
> > > > >    | Literal Interpretation Error | 21% | 4% | 11% | 6% | 13% |
> > > > >    | Overgeneralization Error | 17% | 13% | 8% | 8% | 6% |
> > > > >    | Misleading Emphasis Error | 7% | 2% | 5% | 3% | 6% |
> > > > >    | Unknowable Reasons Error | 17% | 8% | 19% | 13% | 11% |
> > > > >    | General error rate | 6.4 | 16.0 | 17.8 | 16.6 | 30.5 |

---

> > > > > > ### Author Response · Authors · 2023-08-20
> > > > > > **Rebuttal Part 6 (Q1 part6)**
> > > > > >
> > > > > > ## **Key Insights**
> > > > > > Drawing from each evaluator's limitations and error classifications results, we offer these insights:
> > > > > >    1. Lexical Matching:
> > > > > >       - While lexical matching remains a simple and effective method for Open-QA evaluation, it struggles with issues of limited contextual understanding, low recall, and structural variations.
> > > > > >       - It often marks answers that humans consider correct as incorrect, but rarely does the opposite. This makes Lexical Matching a strict metric, suitable for environments requiring high error recall.
> > > > > >       - When it does mark a human-considered correct answer as wrong, it's usually because the generated answer contains the golden answer, but the overall meaning doesn't support it. For instance, it might negate the golden answer or only use it as part of the response.
> > > > > >       - Lexical Matching struggles with "Structure Variation" errors. For example, if the golden answer is "8 September 2010" and the generated answer is "It was released on September 8, 2010.", Lexical Matching can't recognize it. The other two evaluators rarely have this issue.
> > > > > >       - Due to its inability to handle semantics, it can't manage Synonym or Contextual understanding situations.
> > > > > >    2. Neural Evaluation (BERT-Score and BLEURT in this work):
> > > > > >       - Overall, they aren't well-suited for this QA-Eval task, with the poorest performance among the three types. They can only measure the similarity between two text segments. So, they handle Synonym errors well. However, if the generated answer contains extra information (common with larger models), this can easily influence the BERT-score.
> > > > > >       - BERT-Score isn't great at Contextual Understanding. If the generated answer explains the golden answer without including its entities, BERT-Score can easily get it wrong.
> > > > > >       - Adapting BERT-Score to this QA-Eval task by adjusting the threshold is another issue. Many datasets are highly sensitive to threshold settings.
> > > > > >    3. LLM-evaluator (GPT-3.5 in this work):
> > > > > >       - Overall, the LLM-evaluator can serve as a complement to lexical matching and is valuable for assessing the accuracy of generated answers, it remains sensitive to prompts and the impact of additional contexts, especially for bingchat answers.
> > > > > >       - Its most common error is the "Paraphrasing error", possibly because it's easily influenced by other contexts.
> > > > > >       - It has its own issues, like the "Overgeneralization error", which doesn't appear in the other two evaluators, although they are minor concernss.
> > > > > >       - Sometimes, the LLM-evaluator makes clear mistakes that humans wouldn't. For example, for the question, "Who was the first chief minister of West Bengal?" with the golden answer being "Prafulla Chandra Ghosh", the generated answer was "The first Chief Minister of West Bengal was Dr. Bidhan Chandra Roy." GPT-3.5 marked the generated answer as correct, even though Dr. Bidhan Chandra Roy is not Prafulla Chandra Ghosh. This might be because the evaluator uses its inherent knowledge, overlooking the golden answer, or for other undetermined reasons. Such issues don't appear with the other two evaluators.
> > > > > >
> > > > > >    In general, while lexical matching is a straightforward and effective method for Open-QA evaluation, it encounters challenges related to limited contextual understanding, low recall, and structural variations.
> > > > > >    The LLM-evaluator can be effectively complementary to lexical matching, and is valuable in assessing the accuracy of generated answers, but it is not robust and particularly sensitive to prompts and the influence of additional contexts, especially in the case of BingChat answers.
> > > > > >    The application of neural evaluations in the QA-Eval task requires further exploration.

---

> > > > > > > ### Author Response · Authors · 2023-08-20
> > > > > > > **Rebuttal Part 7 (Q1 part7)**
> > > > > > >
> > > > > > > ##  3. Characteristics and Limitations of Each Evaluator
> > > > > > > Based on our theoretical analysis and observations of erroneous cases, we identified the following issues with each type of evaluator:
> > > > > > >
> > > > > > > 3.1. Lexical Matching:
> > > > > > >
> > > > > > >    - Lack of Semantic Understanding: The exact match metric doesn't take into account the semantic meaning of the answers. It only checks if the predicted answer is exactly the same as the ground truth, even if the predicted answer is semantically correct but phrased differently.
> > > > > > >    - Inability to Handle Synonyms: The exact match metric cannot handle synonyms. If the predicted answer uses a different word that has the same meaning as the word in the ground truth answer, the exact match metric will consider it as a wrong answer.
> > > > > > >    - Inability to Handle Paraphrasing: Similar to the point above, the exact match metric cannot handle paraphrasing. If the predicted answer is a paraphrase of the ground truth answer, the exact match metric will consider it as a wrong answer.
> > > > > > >    - Inability to Handle Partially Correct Answers: The exact match metric cannot handle partially correct answers. If the predicted answer is partially correct, the exact match metric will consider it as a wrong answer.
> > > > > > >    - Inability to Handle Reordered Words: The exact match metric cannot handle reordered words. If the predicted answer has the same words as the ground truth answer but in a different order, the exact match metric will consider it as a wrong answer.
> > > > > > >    - Inability to Handle Different Formats: The exact match metric cannot handle different formats. If the predicted answer is in a different format than the ground truth answer (for example, dates or numbers), the exact match metric will consider it as a wrong answer.
> > > > > > > These limitations highlight the need for more sophisticated evaluation metrics that can understand the semantic meaning of the answers and handle synonyms, paraphrasing, partially correct answers, reordered words, different levels of detail, and different formats.
> > > > > > >
> > > > > > >
> > > > > > > 3.2 Neural Evaluation:
> > > > > > >
> > > > > > >    - The limitations of neural evaluation methods, such as BERT-Score and BLEURT, are evident. Most crucially, many neural evaluations are primarily designed to measure the similarity between two phrases or sentences. They are not tailored for binary tasks, especially those assessing the factual correctness of answers. Instead, they provide a continuous score that gauges the similarity between the generated text and the reference text, rendering them directly unsuitable for this particular task. In our study, we employed BERT-score and BLEURT for this task by setting a threshold. However, the performance of both BERT-score and BLEURT was suboptimal. The primary shortcoming of neural evaluations for this task is their misalignment with its requirements.
> > > > > > >    - Furthermore, BERT-score has the following limitations:：
> > > > > > >    - Sensitivity to Verbosity: BERT-score may penalize verbose answers even if they contain the correct information. If the AI-generated answer provides a detailed explanation while the golden answer is concise, the score might be lower than expected.
> > > > > > >    - Mismatched Focus: If the AI-generated answer is correct but emphasizes different aspects or details than the golden answer, BERT-score might not recognize the similarity, leading to a lower score.
> > > > > > >    - Lack of Contextual Understanding: BERT-score measures the similarity between embeddings but might not fully capture the contextual nuances of certain answers, especially when there are multiple valid ways to answer a question.
> > > > > > >    - Synonym and Paraphrasing Issues: BERT-score might not always recognize synonyms or paraphrased answers as being equivalent to the golden answer, leading to potential discrepancies in scoring.
> > > > > > >    - Threshold Limitations: Setting a fixed threshold (e.g., 0.5) for determining correctness can be arbitrary. Some answers might be just below the threshold but still be correct, while others might be just above but incorrect.
> > > > > > >    - Doesn't Account for Minor Details: BERT-score might not be sensitive enough to minor inaccuracies in the AI-generated answer, especially if the overall semantic content is similar to the golden answer.
> > > > > > >    - Lack of Absolute Truth Measure: BERT-score is a relative measure of similarity between two pieces of text. It doesn't provide an absolute measure of the truthfulness or correctness of an answer.
> > > > > > >    - Influence of Sentence Structure: The structure or order of sentences in the AI-generated answer compared to the golden answer might affect the score, even if the content is the same.
> > > > > > > In summary, while BERT-score is a powerful metric for evaluating text similarity, its application in a QA-eval task has severe limitations.

---

> > > > > > > > ### Author Response · Authors · 2023-08-20
> > > > > > > > **Rebuttal Part 8 (Q1 part8)**
> > > > > > > >
> > > > > > > > BLEURT may have addressed some issues by several approaches:
> > > > > > > >
> > > > > > > >    - Fine-tuned on Human Judgments: BLEURT is trained on human evaluations, which means it's more likely to align with human judgments of answer quality. This could help in cases where the AI-generated answer is correct but phrased differently from the golden answer.
> > > > > > > >    - Better Handling of Verbosity: Since BLEURT is designed to evaluate longer pieces of text (like translations), it might be better equipped to handle verbose answers without penalizing them unfairly.
> > > > > > > >    - Improved Contextual Understanding: BLEURT's training process, which involves comparing machine-generated text to reference text and using human evaluations, might give it a better understanding of context, making it more adept at recognizing correct answers that emphasize different aspects of a question.
> > > > > > > >    - Synonym and Paraphrasing: Given its training on human evaluations, BLEURT might be better at recognizing synonyms and paraphrased content as being equivalent to reference answers.
> > > > > > > >
> > > > > > > >
> > > > > > > > However, BLEURT doesn't perfectly address these issues. Its experimental results only slightly outperform BERT-score but still fall significantly short of human evaluations and are notably inferior to the GPT-3.5 evaluator. We will present the specific details in Section 4.
> > > > > > > >
> > > > > > > >
> > > > > > > > 3.3 LLM (GPT-3.5 in this work) :
> > > > > > > >
> > > > > > > >    - Sensitivity to Prompt Engineering: Both closed-source and open-source LLMs can be sensitive to the way questions are framed or prompts are constructed. This can introduce variability in the evaluation, where slight rephrasings might lead to different judgments.
> > > > > > > >    - Literal Interpretation: One of the limitations is the model's tendency to interpret questions or golden answers too literally. This can lead to situations where the evaluator fails to recognize correct answers that provide a broader context or a different interpretation that still addresses the core of the question.
> > > > > > > >    - Overgeneralization: Another challenge is the model's propensity to overgeneralize based on its vast training data. This can result in the evaluator deeming an answer as correct even if it doesn't align specifically with the nuances of the question at hand.
> > > > > > > >    - Misleading Emphasis: The evaluator might sometimes be swayed by partial correctness in an answer. If an answer emphasizes certain correct elements, the evaluator might overlook primary claims that are factually incorrect, leading to a misleading evaluation.
> > > > > > > >    - Unknowable Reasoning: There are instances where the evaluator's judgment is puzzling, even to human experts. The model might deem an answer as correct that has no discernible correlation with the golden answer. This limitation underscores the "black-box" nature of deep learning models, where their internal reasoning processes remain opaque.
> > > > > > > >    - Potential Bias: All LLMs, whether closed or open source, can inherit biases from their training data. In the context of QA-Eval, this might manifest as favoring certain types of answers or being biased against certain topics or contexts
> > > > > > > >
> > > > > > > > (We have included those analysis, including Error Categories, Error Analysis and Characteristics and Limitations of Each Evaluator into the Section 6.1 **Error Analysis in QA-Eval** and Appendix Section E4 **Error Analysis in QA-Eval** in the paper.)

---

> > > > > > > > > ### Author Response · Authors · 2023-08-20
> > > > > > > > > **Rebuttal Part 9 (Q1 part9)**
> > > > > > > > >
> > > > > > > > > ## 4. BLEURT Experiments
> > > > > > > > >
> > > > > > > > > We also conducted a QA-Eval analysis on a more recent Neural-Evaluation model, BLEURT. Similar to BERT-Score, we applied a threshold to BLEURT to make it suitable for QA-Eval. In this work, we set the threshold at 0.2 based on observed distributions.
> > > > > > > > >
> > > > > > > > > ### Performance of BERT-Score and BLEURT on the EVOUNA
> > > > > > > > > In each cell, the left is the accuracy while the right is the Macro-F1.
> > > > > > > > >
> > > > > > > > > |                    | NQ-FiD               | NQ-GPT35  | NQ-ChatGPT35 | NQ-ChatGPT4 | NQ-BingChat |
> > > > > > > > > |--------------------|----------------------|-----------|--------------|-------------|-------------|
> > > > > > > > > | Lexical Matching      | 86.9/86.0<br>89.6/88.8 | 84.8/84.3 | 80.3/78.2    | 83.2/78.1   | 82.3/77.7   |
> > > > > > > > > | BERT-Score         | 75.0/66.0            | 69.5/64.8 | 72.8/66.0    | 76.8/65.8   | 67.6/59.5   |
> > > > > > > > > | BELURT             | 84.4/79.9            | 74.1/63.9 | 78.0/64.9    | 85.0/66.3   | 82.8/65.0   |
> > > > > > > > > | GPT-3.5            | 93.6/92.6            | 84.0/83.0 | 82.2/79.5    | 83.4/77.2   | 69.5/65.5   |
> > > > > > > > > | Another Human      | **96.3/95.6**        | **96.8/96.2** | **95.6/95.2** | **96.6/94.4** | **95.5/93.2** |
> > > > > > > > >
> > > > > > > > > #### on EVOUNA-NaturalQuestions
> > > > > > > > >
> > > > > > > > > |                    | TQ-FiD               | TQ-GPT35  | TQ-ChatGPT35 | TQ-ChatGPT4 | TQ-BingChat |
> > > > > > > > > |--------------------|----------------------|-----------|--------------|-------------|-------------|
> > > > > > > > > | Lexical Matching      | 90.0/86.0<br>91.8/88.2 | 92.3/89.6 | 92.3/87.7    | 91.1/81.3   | 89.8/79.3   |
> > > > > > > > > | BERT-Score         | 65.4/59.6            | 75.7/66.6 | 80.7/65.4    | 83.4/62.7   | 80.4/63.9   |
> > > > > > > > > | BELURT             | 88.1/77.8            | 82.9/66.6 | 85.2/66.1    | 88.8/66.2   | 90.8/64.7   |
> > > > > > > > > | GPT-3.5            | 95.7/93.2            | 91.2/88.3 | 92.7/87.2    | 92.5/82.2   | 81.2/69.0   |
> > > > > > > > > | Another Human      | **99.7/99.8**        | **99.4/99.6** | **98.8/99.2** | **99.8/99.9** | **99.8/99.9** |
> > > > > > > > >
> > > > > > > > > #### on EVOUNA-TriviaQA
> > > > > > > > >
> > > > > > > > >
> > > > > > > > > The results are shown in the table above. Although BLEURT outperforms BERT-Score on most datasets, it still lags behind the performance of Lexical Matching, GPT-3.5 and human, in terms of Macro-F1.
> > > > > > > > >
> > > > > > > > > We have incorporated this experiment into the Appendix E.7 section of the updated paper.

---

> ### Author Response · Authors · 2023-08-20
> **Rebuttal Part 10 (Q2)**
>
> - Q2: The paper only discusses one open QA model on two open QA datasets, the DPR+FiD model,2.1 it would be much better to see if the evaluation of other open, transparent open QA models, like ATLAS (https://arxiv.org/pdf/2208.03299.pdf), LLaMa, etc.
>
>     R2: Thank you for pointing out the importance of evaluating a broader range of open QA models. We recognize the significance of including other prominent models like ATLAS and LLaMa-2 in our study.
>
>     To address this, we expanded our experiments to encompass both ATLAS and LLaMa-2 platforms. However, due to the constraints of time and resources, our current annotations cover 500 samples from the NQ dataset. During our experiments, we notice that the base version of LLaMa-2 occasionally deviated from our instructions. As a result, we chose to proceed with Chat-Llama-2 for a more consistent evaluation. In order to run the ATLAS model on our server, we have to use compressed index, which greatly influence the performance on NQ. Below are the results:
>
>     #### Open-QA and QA-Eval results of Atlas and Chat-Llama2 on 500 samples of NQ
>     In each cell, the left is the accuracy while the right is the Macro-F1.
>
>     |                | NQ-Atlas | NQ-ChatLlama2 |
>     |----------------|----------|---------------|
>     | Lexical Matching  | 92.6/92.5| 89.5/88.2     |
>     | BERT-Score     | 67.1/65.8| 68.2/68.0     |
>     | GPT-3.5        | 64.7/63.9| 66.1/53.6     |
>
>     **Human Score on NQ-Atlas:** 47.9;
>     **Human Score on NQ-ChatLlama2:** 29.7
>
>     It's evident from the results that the performance of ATLAS and Chat-Llama2 is somewhat below the models discussed in our paper. Moreover, the evaluators' performance on NQ-Atlas and NQ-ChatLlama2 is consistent with the trends observed for the models we initially discussed.
>
>     We appreciate the suggestion and will consider a more comprehensive evaluation of additional models in our future work.
>
>     We have incorporated this section into the Appendix E.8 section of the updated paper.

---

> ### Author Response · Authors · 2023-08-20
> **Rebuttal Part 11 (Q3)**
>
> - Q3: The lexical match score for LLM predictions is inconsistent (substring versus exact match) with other models (FiD), it does not support fair evaluation comparison. Will proper prompting solve the issue?
>
>     R3: Thank you for highlighting the potential inconsistency in our evaluation metrics. We recognize the significance of ensuring a consistent evaluation standard for a fair comparison across various models.
>
>     To address this concern, we carried out additional experiments on NQ-FiD and TQ-FiD using substring matching. The results show that substring matching consistently outperforms exact matching. Specifically, for NQ-FiD, the macro-F1 score was 88.8 using substring matching compared to 86.0 with exact matching. Similarly, for TQ-FiD, the scores were 88.2 and 86.0, respectively.
>
>     The initial choice of using exact match and substring matching was driven by the observation that FiD-generated answers differ in length and pattern from those generated by LLMs. To provide a comprehensive view, we now report results for both substring and exact match in our paper.
>
>     We have modified the related Tables, including Table 5, Table6 and Table 12, and the related discussions in the updated paper.

---

> ### Author Response · Authors · 2023-08-20
> **Rebuttal Part 12 (Q4  part1)**
>
> ## Limitations
> - Q4: (1) The human annotation process is somewhat biased since the authors are the annotators. The annotation process may not be scientific.  (2) It is also unclear how many authors and how the authors perform the annotation. (3) Only vague guidelines (four simple rules) are provided in the appendix, it also does not warrant consistent human annotation
>
>
> R4: Thank you for your feedback and for highlighting these concerns. We understand the importance of a rigorous and unbiased annotation process, especially in a study of this nature. Let us address each of your points:
>
> 1. The authors who conducted the annotations were not the ones who designed the annotation guidelines. This separation of roles helps to ensure that the annotators did not have any preconceived notions or expectations about the results based on the design of the guidelines.
> The annotators were not informed that their annotations would be used for evaluating Lexical Matching or BERT-Score. This lack of knowledge about the specific use of the annotations helps to prevent any potential bias in the annotation process, as the annotators could not skew their annotations to favor any particular outcome.
> The people we recruited for the annotation process did not have a conflict of interest, and they did not stand to gain any benefits from generalizing the annotation results. This further ensures the objectivity of the annotations.
> After the annotation process was completed, the annotators were given the choice between authorship and payment. All of them chose authorship, which indicates their commitment to the scientific process and their belief in the validity of the results.
> We believe that these factors collectively ensure the objectivity and reliability of our annotation process. We will make sure to include these details in the revised version of the paper to provide more transparency about the process.
> 2. Three authors participating in the annotation process are Sirui Cheng, Bowen Ding and Zhikun Xu. They are responsible for (DPR+FiD, GPT-3.5, ChatGPT-3.5), ChatGPT-4 and Bing Chat, respectively.
> 3. We have consolidated various annotation-related content that was previously dispersed throughout the paper. This includes identifying improper questions, processing data from Newbing, and assessing the correctness of answers, among other aspects. The result is a comprehensive annotation guideline. This integrated guideline closely mirrors the guidelines we previously presented to the annotators in a PPT. The initial decision to distribute this content in various sections of the paper was to ensure a smooth narrative flow and to avoid repetition.
>
> The overall annotation guidelines are as follows:

---

> > ### Author Response · Authors · 2023-08-20
> > **Rebuttal Part 13 (Q4 part2)**
> >
> > ## Human Annotation Guidelines
> >
> > Here is a question, a set of golden answers (split with /), an AI-generated answer. You are required to judge:
> >
> > 1.  Whether the question have answers that change over time, simply annotate Yes or No.
> >
> > 2.  Whether the golden answer contain severe errors.
> >
> > 3.  Whether the AI-generated answer is correct according to the question and golden answers, simply annotate Yes or No.
> >
> >
> > ### Guidelines for task (1)
> > whether the question have answers that change over time:
> >
> > -   If the question is clearly time-sensitive, then it is Yes.
> >
> > -   If there are words closely related to the current time node such as “this year”, “last year”, “next time” and “last time” in this question, then it is Yes.
> >
> > -   If the question contains values that change over decades, such as “who is the player with the most goals in the World Cup so far”, then it is Yes.
> >
> > -   If the question contains values that do not change in decades, such as “what is the tallest mountain in the world”, then it is No.
> >
> >
> > If the answer to task (1) is Yes, skip to the next.
> >
> > ### Guidelines for Task (2)
> > Whether the Golden Answer Contains Severe Errors
> >
> > -   If the golden answer has structure errors, then it is Yes.
> >
> >     Example: Question: the south west wind blows across nigeria between?
> >     Golden: till September
> >
> > -   If the golden answer is obviously not what is asked, then it is Yes.
> >
> > -   If the golden answer has format errors, then it is Yes.
> >
> >     Example: Question: what season does bart bass die in gossip girl?
> >     Golden: $($
> >
> > -   If the golden answer has only factual errors, then it is No.
> >
> >
> > If the answer to task (2) is Yes, skip to the next.
> >
> > ### Guidelines for Task (3)
> >
> > -   If the question specifies a number (e.g., names of four people), and the response does not meet this requirement (e.g., provides only one name), the answer is deemed incorrect.
> >
> > -   Spelling errors in the responses are considered mistakes. For example, if “golden answer” is misspelled as “gloden answer,” the response is marked as incorrect.
> >
> > -   For questions related to specific times, such as “When was the term social justice first used?” a response of “1840s” would be considered correct. However, if the answer needs to be precise to a specific day, month, and year, each time component needs to be factually accurate for the response to be marked as correct.
> >
> > -   For location-based queries, like “Where was Oak Island filmed?”, a response of “Canada” would be deemed correct. But, if the answer requires specific details like state, city, or county, each geographical component must be accurate for the answer to be considered correct.
> >
> > -   If there is a direct answer and subsequent explanation in the response, then only focus on whether the direct answer is correct, not whether the subsequent explanation is correct.
> >
> >
> > These guidelines were strictly followed to maintain the reliability and validity of the evaluation process.
> >
> > (We have present the detailed guidelines in the Appendix C.2 section of the updated paper.)

---

> ### Author Response · Authors · 2023-08-20
> **Rebuttal Part 14 (Q5)**
>
> # Questions
>
> - Q5: Why is only a portion (500 cases instead of a full >10k) of the TQ test set selected? Is it due to limited human annotation resources? If so, a more comprehensive analysis of the full test set is required to draw statistically sound conclusions.
>
>     R5: Thank you for your inquiry regarding our decision to annotate a portion of the TQ test set. Your understanding is correct; our decision was significantly influenced by the constraints of our human annotation resources. Here's a detailed breakdown:
>
> 1. **Annotation Process**: The annotation process is multi-faceted. For each sample, the initial task is to assess the appropriateness of the question and the golden answers. Following this, the correctness of the AI-generated answer is judged. The average time required for annotation varies based on the model (including identifying the improper question):
>   - FiD answer: 20-30 seconds
>   - GPT-3.5/ChatGPT-3.5 answer: 40-50 seconds
>   - GPT-4 answer (involving webpage interaction): 50-60 seconds
>   - BingChat answer (considering webpage interaction and format processing): 60-70 seconds
>
>     Given that each sample contains answers from five distinct models, a single case demands approximately 4 minutes of annotation.
>
> 2. **Total Annotation Time**: We have annotated close to four thousand samples, translating to 16,000 minutes or 270 hours. This averages to 90 hours per annotator. The unique access requirements for GPT-4 and Bing Chat further compounded our annotation efforts.
>
> 3. **Post-Submission Efforts**: Since our initial submission, we have continued our annotation efforts, adding over 1,500 more samples. We've also re-conducted experiments on the expanded TQ dataset, now comprising 2,000 cases, and have updated the paper accordingly. Some of the revised experimental results are presented below:
>
>     |             | NaturalQuestions |              | TriviaQA |              |
>     | ----------- | ---------------: | -----------: | -------: | -----------: |
>     |             |            Human | LexicalMatch |    Human | LexicalMatch |
>     | DPR + FiD   |             68.4 |         56.1 |     79.6 |         73.5 |
>     | GPT-3.5     |             65.1 |         50.7 |     72.2 |         71.0 |
>     | ChatGPT-3.5 |             64.3 |         57.9 |     80.9 |         76.7 |
>     | ChatGPT-4   |             80.9 |         65.8 |     86.0 |         82.1 |
>     | Bing Chat   |             79.1 |         65.4 |     73.1 |         81.6 |
>
>     Performance of Eval-Models on the EVOUNA. In each cell, the left is the accuracy while the right is the Macro-F1.
>
>     |               | TQ-FiD                 | TQ-GPT35      | TQ-ChatGPT35  | TQ-ChatGPT4   | TQ-BingChat   |
>     | ------------- | ---------------------- | ------------- | ------------- | ------------- | ------------- |
>     | Lexical Matching | 90.0/86.0<br>91.8/88.2 | 92.3/89.6     | 92.3/87.7     | 91.1/81.3     | 89.8/79.3     |
>     | BERT-Score    | 65.4/59.6              | 75.7/66.6     | 80.7/65.4     | 83.4/62.7     | 80.4/63.9     |
>     | GPT-3.5       | 95.7/93.2              | 91.2/88.3     | 92.7/87.2     | 92.5/82.2     | 81.2/69.0     |
>     | Another Human | **99.7/99.8**          | **99.4/99.6** | **98.8/99.2** | **99.8/99.9** | **99.8/99.9** |
>
>     #### on EVOUNA-TriviaQA
> 4. Consistency with Larger Dataset: It's worth noting that when we evaluated the entire 10k+ TQ test set using the same FiD checkpoint and scored it with a lexical match, the result was 73.6%. This closely aligns with the 73.5% score from our 2k dataset, suggesting that the 2k subset is representative of the larger dataset's distribution.
>
>
> We are committed to enhancing the comprehensiveness of our dataset and plan to continue our annotation efforts in the future.
>
> We have modified all TQ-related tables, figures and contents of the updated paper.

---

> ### Author Response · Authors · 2023-08-20
> **Rebuttal Part 15 (Q6)**
>
> - Q6: Table 5 results seem to show plain lexical match score already does a very good evaluation job, what does 10% gap mean compared to human annotators?
>
>
> R6: Thank you for your question regarding the results in Table 5 and the implications of the 10% gap compared to human annotators. Here's our detailed response:
>
>
> 1. **Relative Ranks of QA Models**: While the plain lexical match score might seem to perform well at first glance, it's essential to consider its reliability in evaluating the performance of QA models. Our analysis revealed that the lexical match score produces different relative ranks for the five QA models when compared to human evaluations. This discrepancy underscores its unreliability as an evaluator for QA model performance. The differences in rankings are illustrated in the subsequent table:
>
>     ### Evaluation scores assigned by various evaluation models to different QA models on EVOUNA
>     In each cell, the left represents the score given by the evaluator (row) to the QA model's performance (column) on the respective dataset, while the right is the relative ranks among the five models.
>
>     | Model/Method | NQ-FiD | NQ-GPT35 | NQ-ChatGPT35 | NQ-GPT4 | NQ-BingChat |
>     |--------------|--------|----------|--------------|---------|-------------|
>     | Lexical Matching | 56.1 (4) | 50.7 (5) | 57.9 (3) | 65.8 (1) | 65.4 (2) |
>     | BERT-Score | 82.5 (1) | 70.9 (4) | 71.9 (3) | 74.7 (2) | 64.8 (5) |
>     | GPT-3.5 | 67.5 (2) | 58.6 (4) | 63.2 (3) | 70.3 (1) | 54.1 (5) |
>     | Human | 69.7 (4) | 70.3 (3) | 63.0 (5) | 80.3 (1) | 78.2 (2) |
>
>     #### on EVOUNA-NaturalQuestions
>
>     | Model/Method | TQ-FiD | TQ-GPT35 | TQ-ChatGPT35 | TQ-GPT4 | TQ-BingChat |
>     |--------------|--------|----------|--------------|---------|-------------|
>     | Lexical Matching | 73.5 (4) | 71.0 (5) | 76.7 (3) | 82.1 (1) | 81.6 (2) |
>     | BERT-Score | 56.7 (5) | 73.9 (4) | 82.0 (2) | 84.4 (2) | 78.0 (3) |
>     | GPT-3.5 | 71.7 (4) | 66.0 (5) | 72.4 (3) | 82.3 (1) | 73.4 (2) |
>     | Human | 79.6 (3) | 72.2 (5) | 80.9 (2) | 86.0 (1) | 73.1 (4) |
>
>     #### on EVOUNA-TriviaQA
>
> 2. **Recall Scores**: The lexical match method consistently exhibits the lowest recall scores among the four methods we analyzed. This indicates that it frequently misjudges correct samples as incorrect. A detailed breakdown of these recall scores can be found in Table X in the appendix. The relatively low recall scores of the lexical match method emphasize its tendency to be overly stringent, often failing to recognize valid answers that may not match the reference lexically but are semantically correct.
>
>
> 3. **Significance of the 10% Gap**: It's crucial to contextualize the 10% gap in light of human annotator variability. Given that discrepancies between human annotators are typically in the single-digit percentage range, a 10% gap is substantial. Our analysis of the error rates produced by each evaluator, compared to human annotators, revealed that the errors introduced by the three evaluators are several times those made by humans.
>
>     ### Error results of Eval-Models on the EVOUNA
>     In each cell, the left is the error rates while the right is the times compared with another human results.
>
>     |                   | NQ-FiD        | NQ-GPT35      | NQ-ChatGPT35  | NQ-ChatGPT4   | NQ-BingChat   |
>     |-------------------|---------------|---------------|---------------|---------------|---------------|
>     | Lexical Matching     | 13.1/3.5x     | 15.2/4.8x     | 19.7/4.5x     | 16.8/4.9x     | 17.7/3.9x     |
>     | BERT-Score        | 25.0/6.8x     | 30.5/9.5x     | 27.2/6.2x     | 23.2/6.8x     | 32.4/7.2x     |
>     | GPT-3.5           | 6.4/1.7x      | 16.0/5.0x     | 17.8/4.0x     | 16.6/4.9x     | 30.5/6.8x     |
>     | Another Human     | **3.7/1.0x**  | **3.2/1.0x**  | **4.4/1.0x**  | **3.4/1.0x**  | **4.5/1.0x**  |
>
>     #### on EVOUNA-NaturalQuestions
>
>     |                   | TQ-FiD        | TQ-GPT35      | TQ-ChatGPT35  | TQ-ChatGPT4   | TQ-BingChat   |
>     |-------------------|---------------|---------------|---------------|---------------|---------------|
>     | Lexical Matching     | 10.0/33.3x    | 7.7/12.8x     | 7.7/6.4x      | 8.9/44.5x     | 10.2/51.0x    |
>     | BERT-Score        | 34.6/115.3x   | 24.3/40.5x    | 19.3/16.1x    | 16.6/83.0x    | 19.6/98.0x    |
>     | GPT-3.5           | 4.3/14.3x     | 8.8/14.7x     | 7.3/6.1x      | 7.5/37.5x     | 18.8/94.0x    |
>     | Another Human     | **0.3/1.0x**  | **0.6/1.0x**  | **1.2/1.0x**  | **0.2/1.0x**  | **0.2/1.0x**  |
>
>     #### on EVOUNA-TriviaQA

---

> ### Comment · Reviewer_8Zik · 2023-08-24
>
> The authors provided a comprehensive response to address my concerns, I increased my rating. Thanks!

---

### Official Review · Reviewer_WzGc · 2023-07-23
**The authors have introduced a new task, Evaluating QA Evaluation (QA-Eval), and the corresponding dataset EVOUNA, designed to assess the accuracy of AI-generated answers in relation to standard answers within Open-QA. They have utilized human-annotated results to measure the performance of their evaluation methods and have investigated methods that show high correlation with human evaluations, deeming them more reliable.   The authors have also explored the impact of prompt engineering on their evaluation methods and experiment design, experimenting with different strategies including ignoring background information, providing reasoning, chain of thoughts (CoT), and employing in-context learning (ICL).  The paper is interesting and a good fit to this domain.**

**Rating:** 9
**Confidence:** 5
**Clarity:** Yes.

**Strengths:**

This paper proposes a new task and dataset that can facilitate the development of more effective automatic evaluation tools for Open-QA. Secondly, the study investigates methods that show a high correlation with human evaluations, deeming them more reliable.


**Additional Feedback:**

This groundbreaking study thoughtfully challenges the limitations of current automatic evaluation methods in Open-QA tasks, advancing the field with the introduction of the QA-Eval task and the EVOUNA dataset, both of which are poised to significantly enhance our ability to assess the accuracy and reliability of AI-generated responses.


**Correctness:**

The authors have followed appropriate research practices and have provided a detailed description of their evaluation methods and experiment design.

**Documentation:**

Yes, there are sufficient details on data collection and organization. And they make their datasets public on repo https://github.com/wangcunxiang/QA-Eval.

**Ethics:**

No, there are no ethics concerns.

**Limitations:**

The study could benefit from a more detailed analysis of these limitations and impacts, as well as specific measures to mitigate them.

For example, the authors could provide more information on the potential negative societal impacts of AI-generated answers in Open-QA, such as the potential for bias or misinformation. They could also propose specific measures to mitigate these impacts, such as developing more accurate and reliable automatic evaluation tools or implementing human oversight in the evaluation process.


**Opportunities For Improvement:**

While the study acknowledges the potential negative societal impacts of the research, it does not provide a detailed analysis of these impacts or propose specific measures to mitigate them. This may limit the ethical implications of the research.


**Relation To Prior Work:**

It is not explicitly stated whether the authors have discussed how their work differs from previous contributions. However, they have utilized human-annotated results to measure the performance of their evaluation methods and have investigated methods that show a high correlation with human evaluations, deeming them more reliable, which is good.



**Summary And Contributions:**

This paper focuses on assessing the accuracy of AI-generated answers in relation to standard answers within Open-QA. The authors introduce a new task, Evaluating QA Evaluation (QA-Eval), and the corresponding dataset EVOUNA, designed to measure the performance of current automatic evaluation methods. The study also proposes new methods to improve LLM-based evaluators. Overall, this paper contributes to the development of more effective automatic evaluation tools for Open-QA.

---

> ### Author Response · Authors · 2023-08-20
> **Rebuttal Part 1 (R1 Part 1)**
>
> Thank you so much for your casreful and insightful comments.
>
> To  enrich the actionable insights researchers may get out of this work, we have conducted the QA-Eval Error Analysis, which consists three parts:
>
> 1. **Categorization of Evaluator Errors**: Recognizing that different evaluators might err in common and unique ways, we established a taxonomy of potential errors. This categorization helps in understanding the nature of mistakes and provides insights into areas of improvement for each evaluator.
> 2. **Empirical Analysis of Errors**: Based on our error taxonomy, we conducted a hands-on analysis of the actual errors produced by each evaluator on our dataset. This empirical study allowed us to validate our theoretical findings and provided a finegrained understanding of where each evaluator might not fit in real-world scenarios.
> 3. **Detailed Analysis of Evaluators**: We delved into the characteristics and  limitations of various evaluators, including Lexical Matching, Neural Evaluation (BERT-Score/BLEURT), and LLM-as-evaluator (GPT-3.5)
>
> Here are the details for each part:
>
> ## 1. Error Categories
>
> In addition to characteristic, the strength and limitations of each type of evaluator, we have designed a set of Evaluator Error categories. This includes two common errors found across all evaluators as well as specific errors unique to each type of evaluator.
>
> ### General Error Categories for All Evaluators
>
> | Error Type             | Definition                                                   | Example                                                      |
> | ---------------------- | ------------------------------------------------------------ | ------------------------------------------------------------ |
> | **Paraphrasing Error** | The evaluator fails to recognize answers that paraphrase the golden answer correctly but do not contain the exact substring. | Question: "What is the process by which plants convert sunlight into energy?"<br> Golden Answer: "Photosynthesis"<br> Generated Answer: "The mechanism plants use to transform light into energy is termed the photosynthetic process."<br> Explanation: the generated answer is a paraphrase of the "Photosynthesis" but does not contain the word directly. |
> | **Synonym Error**      | The evaluator fails to recognize answers that use synonyms or alternative phrasing to convey the same meaning as the golden answer. | Question: "What's another term for a doctor?"<br> Golden Answer: "Physician"<br> Generated Answer: "A medical practitioner."<br> Explanation: "medical practitioner" is a synonym for "physician" but isn't a direct substring. |
>
> ### Specific Error Categories
>
> | Judger            | Error Type                            | Description                                                  | Example                                                      |
> | ----------------- | ------------------------------------- | ------------------------------------------------------------ | ------------------------------------------------------------ |
> | **Lexical Matching** | **Partial Match Error **              | The evaluator fails to recognize answers that contain a part of the golden answer but not the entire substring. | Question: "Who painted the Mona Lisa?"<br> Golden Answer: "Leonardo da Vinci"<br> Generated Answer: "The Mona Lisa was painted by Leonardo."<br> Explanation: only "Leonardo" is mentioned, not the full "Leonardo da Vinci". |
> |                   | **Structure Variation Error**         | The evaluator fails to recognize answers that essentially convey the same information as the golden answer but there's a variation in how it's structured. | Question: "When did 'Amnesia: The Dark Descent' come out?"<br> Golden Answer: "8 September 2010"<br> Generated Answer: "Amnesia: The Dark Descent was released on September 8, 2010."<br> Explanation: the date format in the generated answer has an extra comma than the golden answer, even though the information is the same. |
> |                   | **Overall Misleading Error for**      | The evaluator mistakenly recognizes the answer as correct because it contains a substring from the golden answer, even if the overall context of the answer is misleading. | Question: "Who wrote 'The Great Gatsby'?"<br> Golden Answer: "F. Scott Fitzgerald"<br> Generated Answer: "Ernest Hemingway and F. Scott Fitzgerald were close friends, but Hemingway wrote 'The Old Man and the Sea'."<br> Explanation: The generated answer contains the substring "F. Scott Fitzgerald", which might lead the Lexical Match Evaluator to judge it as correct. However, the overall context of the answer is misleading, suggesting a relationship between Hemingway and "The Great Gatsby", which is incorrect. |

---

> > ### Author Response · Authors · 2023-08-20
> > **Rebuttal Part 2 (R1 Part 2)**
> >
> > | Judger            | Error Type                            | Description                                                  | Example                                                      |
> > | ----------------- | ------------------------------------- | ------------------------------------------------------------ | ------------------------------------------------------------ |
> > | **Neural Evlauation** | **Contextual Misunderstanding Error** | The evaluator might misjudge answers based on word embeddings and fail to capture the context in which certain words or phrases are used. | Question: "Who wrote 'Romeo and Juliet'?"<br> Golden Answer: "William Shakespeare"<br> AI-generated Answer: "Shakespeare wrote many plays."<br> Explanation: Even though the AI answer mentions Shakespeare, it doesn't directly answer the question. |
> > |                   | **Threshold Sensitivity**             | Answers that are just below the threshold might be correct but are judged as incorrect, and vice versa. | Question: "What's the capital of France?"<br> Golden Answer: "Paris"<br> AI-generated Answer: "The capital city of France is Paris."<br> Explanation: The AI answer is correct but might score just below the threshold due to added context. |
> > |                   | **Extended Answer Error**             | The evaluator might penalize answers that provide more context or details than the golden answer, even if they are correct, because the BERT-score only considers the similarities of the candidates and references. | Question: "Who painted the Mona Lisa?"<br> Golden Answer: "Leonardo da Vinci"<br> AI-generated Answer: "Leonardo da Vinci, a renowned Italian artist, painted the Mona Lisa."<br> Explanation: The AI answer provides more context but is still correct. |
> > | **LLM-evaluator** | **Literal Interpretation Error**      | The evaluator might take the question or golden answer too literally and fail to recognize correct answers that provide a broader context or interpretation. | Question: "Which bird is known for its beautiful tail?"<br> Golden Answer: "Peacock"<br> Generated Answer: "Many birds have beautiful tails."<br> Explanation: The evaluator might take a literal approach and accept the general statement as correct without focusing on the specific bird in question. |
> > |                   | **Overgeneralization Error**          | The evaluator might generalize based on its training data and judge an answer as correct even if it's not specific to the question. | Question: "Who wrote 'Pride and Prejudice'?"<br> Golden Answer: "Jane Austen"<br> Generated Answer: "An English author."<br> Explanation: The evaluator might accept the general answer as it's not technically wrong, even though it lacks specificity. |
> > |                   | **MIsleading Emphasis Error**         | The evaluator might judge an answer as correct if it includes some correct information and put emphasis on it, and overlook the incorrect primary claim. | Question: "What's the primary gas in Earth's atmosphere?"<br> Golden Answer: "Nitrogen"<br> Generated Answer: "Oxygen, which makes up about 78% of the atmosphere."<br> Explanation: GPT-3.5 might focus on the correct percentage and overlook incorrect mention of "Oxygen" as a primary gas. |
> > |                   | **Unknowable Reasons**                | The evaluator makes an incorrect judgment for an unknowable reason. Even humans cannot figure out why the LLM thinks the generated answer is correct since it has no correlation with the golden answer. | Question: "Who was the first chief minister of West Bengal?"<br> Golden Answer: "Prafulla Chandra Ghosh"<br> Generated Answer: "The first Chief Minister of West Bengal was Dr. Bidhan Chandra Roy."<br> Explanation: GPT-3.5 takes the generated answer as correct, but Dr. Bidhan Chandra Roy is apparently not Prafulla Chandra Ghosh. |
> >
> > While these categories might overlap in certain instances, each possesses its own distinct emphasis. When classifying, we choose the category that best fits each specific case.

---

> > > ### Author Response · Authors · 2023-08-20
> > > **Rebuttal Part 3 (R1 Part 3)**
> > >
> > > ## 2. Error Analysis
> > > Based on the aforementioned Error Categories, we manually classified the errors produced by Lexical Matching, BERT-Score, and GPT-3.5 on each subset of NQ. For each subset, we selected 100 errors. If there were fewer than 100 errors, we included all of them. The classification results are as follows:
> > > ### The error results for Lexical Match evaluator, BERT-Score evaluator and GPT-3.5 evaluator.
> > > Each kind evaluator has common error types and specific error types. General error rate indicates the error proportion of this evaluator on this subset.
> > >
> > > | Error Type/Model | NQ-FiD | NQ-GPT35 | NQ-ChatGPT35 | NQ-ChatGPT4 | NQ-BingChat |
> > > |------------------|--------|----------|--------------|-------------|-------------|
> > > | **Lexical Matching** | | | | | |
> > > | Paraphrasing Error | 29% | 37% | 29% | 60% | 49% |
> > > | Synonym Error | 18% | 12% | 37% | 12% | 19% |
> > > | Partial Match Error | 48% | 30% | 13% | 10% | 20% |
> > > | Structure Variation Error | 4% | 16% | 15% | 12% | 7% |
> > > | Overall Misleading Error | 1% | 5% | 6% | 6% | 5% |
> > > | General error rate | 11.75 | 15.2 | 19.7 | 16.8 | 17.7 |
> > > | **BERT-Score** | | | | | |
> > > | Paraphrasing Error | 4% | 24% | 29% | 39% | 39% |
> > > | Synonym Error | 4% | 7% | 4% | 5% | 5% |
> > > | Contextual Misunderstanding Error | 63% | 22% | 23% | 20% | 15% |
> > > | Threshold Sensitivity Error | 25% | 33% | 20% | 18% | 15% |
> > > | Extended Answer Error | 4% | 14% | 24% | 18% | 26% |
> > > | General error rate | 25.0 | 30.5 | 27.2 | 23.2 | 32.4 |
> > > | **GPT-3.5** | | | | | |
> > > | Paraphrasing Error | 16% | 52% | 36% | 52% | 47% |
> > > | Synonym Error | 22% | 12% | 21% | 18% | 17% |
> > > | Literal Interpretation Error | 21% | 4% | 11% | 6% | 13% |
> > > | Overgeneralization Error | 17% | 13% | 8% | 8% | 6% |
> > > | Misleading Emphasis Error | 7% | 2% | 5% | 3% | 6% |
> > > | Unknowable Reasons Error | 17% | 8% | 19% | 13% | 11% |
> > > | General error rate | 6.4 | 16.0 | 17.8 | 16.6 | 30.5 |

---

> > > > ### Author Response · Authors · 2023-08-20
> > > > **Rebuttal Part 4 (R1 Part 4)**
> > > >
> > > > ## **Key Findings**
> > > > From the results, there are some findings:
> > > > 1. Lexical Matching:
> > > >    - While lexical matching remains a simple and effective method for Open-QA evaluation, it struggles with issues of limited contextual understanding, low recall, and structural variations.
> > > >    - It often marks answers that humans consider correct as incorrect, but rarely does the opposite. This makes Lexical Matching a strict metric, suitable for environments requiring high error recall.
> > > >    - When it does mark a human-considered correct answer as wrong, it's usually because the generated answer contains the golden answer, but the overall meaning doesn't support it. For instance, it might negate the golden answer or only use it as part of the response.
> > > >    - Lexical Matching struggles with "Structure Variation" errors. For example, if the golden answer is "8 September 2010" and the generated answer is "It was released on September 8, 2010.", Lexical Matching can't recognize it. The other two evaluators rarely have this issue.
> > > >    - Due to its inability to handle semantics, it can't manage Synonym or Contextual understanding situations.
> > > > 2. Neural Evaluation (BERT-Score and BLEURT in this work):
> > > >    - Overall, they aren't well-suited for this QA-Eval task, with the poorest performance among the three types. They can only measure the similarity between two text segments. So, they handle Synonym errors well. However, if the generated answer contains extra information (common with larger models), this can easily influence the BERT-score.
> > > >    - BERT-Score isn't great at Contextual Understanding. If the generated answer explains the golden answer without including its entities, BERT-Score can easily get it wrong.
> > > >    - Adapting BERT-Score to this QA-Eval task by adjusting the threshold is another issue. Many datasets are highly sensitive to threshold settings.
> > > > 3. LLM-evaluator (GPT-3.5 in this work):
> > > >    - Overall, the LLM-evaluator can serve as a complement to lexical matching and is valuable for assessing the accuracy of generated answers, it remains sensitive to prompts and the impact of additional contexts, especially for bingchat answers.
> > > >    - Its most common error is the "Paraphrasing error", possibly because it's easily influenced by other contexts.
> > > >    - It has its own issues, like the "Overgeneralization error", which doesn't appear in the other two evaluators, although they are minor concernss.
> > > >    - Sometimes, the LLM-evaluator makes clear mistakes that humans wouldn't. For example, for the question, "Who was the first chief minister of West Bengal?" with the golden answer being "Prafulla Chandra Ghosh", the generated answer was "The first Chief Minister of West Bengal was Dr. Bidhan Chandra Roy." GPT-3.5 marked the generated answer as correct, even though Dr. Bidhan Chandra Roy is not Prafulla Chandra Ghosh. This might be because the evaluator uses its inherent knowledge, overlooking the golden answer, or for other undetermined reasons. Such issues don't appear with the other two evaluators.
> > > >
> > > > In general, while lexical matching is a straightforward and effective method for Open-QA evaluation, it encounters challenges related to limited contextual understanding, low recall, and structural variations.
> > > > The LLM-evaluator can be effectively complementary to lexical matching, and is valuable in assessing the accuracy of generated answers, but it is not robust and particularly sensitive to prompts and the influence of additional contexts, especially in the case of BingChat answers.
> > > > The application of neural evaluations in the QA-Eval task requires further exploration.

---

> > > > > ### Author Response · Authors · 2023-08-20
> > > > > **Rebuttal Part 5 (R1 Part 5)**
> > > > >
> > > > > ##  3. Characteristics and Limitations of Each Evaluator
> > > > > Based on our theoretical analysis and observations of erroneous cases, we identified the following issues with each type of evaluator:
> > > > >
> > > > > 3.1. Lexical Matching:
> > > > >
> > > > >    - Lack of Semantic Understanding: The exact match metric doesn't take into account the semantic meaning of the answers. It only checks if the predicted answer is exactly the same as the ground truth, even if the predicted answer is semantically correct but phrased differently.
> > > > >    - Inability to Handle Synonyms: The exact match metric cannot handle synonyms. If the predicted answer uses a different word that has the same meaning as the word in the ground truth answer, the exact match metric will consider it as a wrong answer.
> > > > >    - Inability to Handle Paraphrasing: Similar to the point above, the exact match metric cannot handle paraphrasing. If the predicted answer is a paraphrase of the ground truth answer, the exact match metric will consider it as a wrong answer.
> > > > >    - Inability to Handle Partially Correct Answers: The exact match metric cannot handle partially correct answers. If the predicted answer is partially correct, the exact match metric will consider it as a wrong answer.
> > > > >    - Inability to Handle Reordered Words: The exact match metric cannot handle reordered words. If the predicted answer has the same words as the ground truth answer but in a different order, the exact match metric will consider it as a wrong answer.
> > > > >    - Inability to Handle Different Formats: The exact match metric cannot handle different formats. If the predicted answer is in a different format than the ground truth answer (for example, dates or numbers), the exact match metric will consider it as a wrong answer.
> > > > > These limitations highlight the need for more sophisticated evaluation metrics that can understand the semantic meaning of the answers and handle synonyms, paraphrasing, partially correct answers, reordered words, different levels of detail, and different formats.
> > > > >
> > > > >
> > > > > 3.2 Neural Evaluation:
> > > > >
> > > > >    - The limitations of neural evaluation methods, such as BERT-Score and BLEURT, are evident. Most crucially, many neural evaluations are primarily designed to measure the similarity between two phrases or sentences. They are not tailored for binary tasks, especially those assessing the factual correctness of answers. Instead, they provide a continuous score that gauges the similarity between the generated text and the reference text, rendering them directly unsuitable for this particular task. In our study, we employed BERT-score and BLEURT for this task by setting a threshold. However, the performance of both BERT-score and BLEURT was suboptimal. The primary shortcoming of neural evaluations for this task is their misalignment with its requirements.
> > > > >    - Furthermore, BERT-score has the following limitations:：
> > > > >    - Sensitivity to Verbosity: BERT-score may penalize verbose answers even if they contain the correct information. If the AI-generated answer provides a detailed explanation while the golden answer is concise, the score might be lower than expected.
> > > > >    - Mismatched Focus: If the AI-generated answer is correct but emphasizes different aspects or details than the golden answer, BERT-score might not recognize the similarity, leading to a lower score.
> > > > >    - Lack of Contextual Understanding: BERT-score measures the similarity between embeddings but might not fully capture the contextual nuances of certain answers, especially when there are multiple valid ways to answer a question.
> > > > >    - Synonym and Paraphrasing Issues: BERT-score might not always recognize synonyms or paraphrased answers as being equivalent to the golden answer, leading to potential discrepancies in scoring.
> > > > >    - Threshold Limitations: Setting a fixed threshold (e.g., 0.5) for determining correctness can be arbitrary. Some answers might be just below the threshold but still be correct, while others might be just above but incorrect.
> > > > >    - Doesn't Account for Minor Details: BERT-score might not be sensitive enough to minor inaccuracies in the AI-generated answer, especially if the overall semantic content is similar to the golden answer.
> > > > >    - Lack of Absolute Truth Measure: BERT-score is a relative measure of similarity between two pieces of text. It doesn't provide an absolute measure of the truthfulness or correctness of an answer.
> > > > >    - Influence of Sentence Structure: The structure or order of sentences in the AI-generated answer compared to the golden answer might affect the score, even if the content is the same.
> > > > > In summary, while BERT-score is a powerful metric for evaluating text similarity, its application in a QA-eval task has severe limitations.

---

> > > > > > ### Author Response · Authors · 2023-08-20
> > > > > > **Rebuttal Part 6 (R1 Part 6)**
> > > > > >
> > > > > > .3 LLM-Evaluator (GPT-3.5 in this work) :
> > > > > >
> > > > > >    - Sensitivity to Prompt Engineering: Both closed-source and open-source LLMs can be sensitive to the way questions are framed or prompts are constructed. This can introduce variability in the evaluation, where slight rephrasings might lead to different judgments.
> > > > > >    - Literal Interpretation: One of the limitations is the model's tendency to interpret questions or golden answers too literally. This can lead to situations where the evaluator fails to recognize correct answers that provide a broader context or a different interpretation that still addresses the core of the question.
> > > > > >    - Overgeneralization: Another challenge is the model's propensity to overgeneralize based on its vast training data. This can result in the evaluator deeming an answer as correct even if it doesn't align specifically with the nuances of the question at hand.
> > > > > >    - Misleading Emphasis: The evaluator might sometimes be swayed by partial correctness in an answer. If an answer emphasizes certain correct elements, the evaluator might overlook primary claims that are factually incorrect, leading to a misleading evaluation.
> > > > > >    - Unknowable Reasoning: There are instances where the evaluator's judgment is puzzling, even to human experts. The model might deem an answer as correct that has no discernible correlation with the golden answer. This limitation underscores the "black-box" nature of deep learning models, where their internal reasoning processes remain opaque.
> > > > > >    - Potential Bias: All LLMs, whether closed or open source, can inherit biases from their training data. In the context of QA-Eval, this might manifest as favoring certain types of answers or being biased against certain topics or contexts
> > > > > >
> > > > > > **(We have included those analysis, including Error Categories, Error Analysis and Characteristics and Limitations of Each Evaluator into the Section 6.1 **Error Analysis in QA-Eval** and Appendix Section E4 **Error Analysis in QA-Eval** in the updated paper.)**

---

> ### Author Response · Authors · 2023-08-20
> **Rebuttal Part 7 (R1 Part 2)**
>
> Thank you for your constructive feedback regarding the ethical implications of our research.
>
> We acknowledge the importance of addressing the potential negative societal impacts of our work in a more comprehensive manner. While our study does highlight the risk of inadvertently disseminating misinformation due to inaccuracies in the gold standard answers of the Natural Questions and TriviaQA datasets, we understand the need to delve deeper into this issue and propose mitigation strategies.
>
> To address this concern:
>
> - Detailed Analysis of Inaccuracies: We are committed to conducting a more thorough analysis of the inaccuracies present in the gold standard answers. By understanding the nature and frequency of these inaccuracies, we can better gauge the potential risks they pose.
>
> - Collaborative Effort: We are exploring collaborations with experts in the field of information verification to enhance the accuracy of our dataset. By leveraging their expertise, we aim to reduce the chances of misinformation.
>
> - Feedback Mechanism: We are considering the implementation of a feedback mechanism within our dataset platform. This would allow users and researchers to report any inaccuracies they come across, helping us continuously refine and improve the dataset.
>
> - Transparency and Disclaimer: We will ensure that any release or publication related to our dataset clearly communicates the potential for inaccuracies. This transparency will help users approach the dataset with an informed perspective.
>
> - Ethical Guidelines: We are in the process of drafting ethical guidelines for the use of our dataset. These guidelines will emphasize the importance of cross-referencing information and the potential risks associated with relying solely on our dataset for definitive answers.
>
> We genuinely appreciate your feedback and are dedicated to ensuring that our research not only advances the field but does so responsibly and ethically.

---

### Author Response · Authors · 2023-08-24
**A Gentle Reminder**

Dear all reviewers,

Thanks for your constructive feedback to this paper! We now have completed the rebuttal and the paper revision per your comments. Could you please check if our response and revision resolve your concerns? If you have any questions, we are happy to answer them:)

Thanks for your hard work!

---

### Decision · Program_Chairs · 2023-09-22

**Decision:**

Accept (Poster)

**Comment:**

This paper introduces a new task, Evaluating QA Evaluation (QA-Eval), and a new dataset, EVOUNA, designed to measure the performance of automatic evaluation methods. It also proposes methods to improve LLM-based evaluators.

The paper is well-written and contributes to the development of effective automatic evaluation tools for Open Question Answering (Open-QA). The authors addressed the reviewers’ questions and comments.